# STATISTICAL ADVANTAGES OF PERTURBING COSINE ROUTER IN MIXTURE OF EXPERTS

**Huy Nguyen**[1]    **Pedram Akbarian**[*,1]    **Trang Pham**[*,2]    **Trang Nguyen**[*,2]
**Shuijan Zhang**[1]        **Nhat Ho**[1]

[1] The University of Texas at Austin    [2] Qualcomm AI Research[†]

## ABSTRACT

The cosine router in Mixture of Experts (MoE) has recently emerged as an attractive alternative to the conventional linear router. Indeed, the cosine router demonstrates favorable performance in image and language tasks and exhibits better ability to mitigate the representation collapse issue, which often leads to parameter redundancy and limited representation potentials. Despite its empirical success, a comprehensive analysis of the cosine router in MoE has been lacking. Considering the least square estimation of the cosine routing MoE, we demonstrate that due to the intrinsic interaction of the model parameters in the cosine router via some partial differential equations, regardless of the structures of the experts, the estimation rates of experts and model parameters can be as slow as $\mathcal{O}(1/\log^{\tau}(n))$ where $\tau > 0$ is some constant and $n$ is the sample size. Surprisingly, these pessimistic non-polynomial convergence rates can be circumvented by the widely used technique in practice to stabilize the cosine router — simply adding noises to the $\ell^2$-norms in the cosine router, which we refer to as *perturbed cosine router*. Under the strongly identifiable settings of the expert functions, we prove that the estimation rates for both the experts and model parameters under the perturbed cosine routing MoE are significantly improved to polynomial rates. Finally, we conduct extensive simulation studies in both synthetic and real data settings to empirically validate our theoretical results.

## 1 INTRODUCTION

Proposed by Jacobs et al. (1991) and Jordan & Jacobs (1994), Mixture of Experts (MoE) has been known as an effective statistical method to incorporate the capabilities of various specialized models called experts. Different from conventional mixture models (Lindsay, 1995) in which the mixture weights are scalars, the MoE rather utilizes a routing mechanism to determine a set of weights depending on an input token. In particular, the router first computes the similarity scores between each token and experts and then assigns more weights to the more relevant experts determined based on those scores. To further improve the scalability of the MoE, Shazeer et al. (2017) has recently introduced a sparse variant of this model, which routes each input to only a subset of experts. This sparse MoE model allows us to increase the number of learnable parameters with nearly constant computational overhead. As a consequence, the sparse MoE has been leveraged in several applications, including large language models (Jiang et al., 2024; Puigcerver et al., 2024; Zhou et al., 2023b; Dai et al., 2024; Pham et al., 2024), computer vision (Riquelme et al., 2021; Liang et al., 2022), speech recognition (You et al., 2021; Gulati et al., 2020; Peng et al., 1996), continual learning (Le et al., 2024; Li et al., 2025), multi-task learning (Hazimeh et al., 2021), and other applications (Han et al., 2024; Chow et al., 2023; Le et al., 2025).

In the above applications, practitioners often use a linear router which calculates the similarity score by taking the inner product of a token hidden representation and an expert embedding. Nevertheless, Chi et al. (2022) discovered that utilizing the linear router might induce the representation collapse issue. This phenomenon occurs when a fraction of experts govern the decision-making process, leading to the redundancy of other experts. In response, Chi et al. (2022) proposed an alternative known as a

---

[*]Equal Contribution
[†]Qualcomm Vietnam Company Limited

cosine router. In particular, this router begins with projecting the token hidden representation into a low-dimensional space, followed by applying $L^2$ normalization to both the token representations and expert embeddings. By doing so, the similarity scores become more stable, circumventing the dominance of certain experts. The efficacy of the cosine routing MoE has been experimentally demonstrated in language modeling (Chi et al., 2022), and domain generalization (Li et al., 2023). On the other hand, a comprehensive theoretical study of the cosine router has remained lacking.

In the literature, there have been some attempts to understand the MoE models with different types of gating functions whose outputs are the composition of some functions and the routing scores. First of all, considering the classification problem with cluster structures, Chen et al. (2022) demonstrated that the router operated by a neural network could learn the cluster-center features, which helped divide a complex problem into simpler classification sub-problems that individual expert could handle. Next, Ho et al. (2022) studied the expert estimation under an input-free gating Gaussian MoE model, showing that the rates for estimating experts depend on the algebraic structures among experts. Subsequently, the Gaussian MoE model with softmax gating was explored in (Nguyen et al., 2023; 2024a) which pointed out that interactions among model parameters via some partial differential equations (PDE) did harm the expert estimation rates. Saying that the setting of Gaussian MoE was far from practice, Nguyen et al. (2024c) rather took into account a regression framework with the regression function being a linear router MoE model. They verified the benefits of formulating experts as feed-forward networks with popular activation functions like ReLU and sigmoid from the perspective of the expert estimation problem.

It is worth noting that the expert estimation problem allows us to capture how fast an expert specializes in a specific task, which is one of the most important problems in the MoE literature known as expert specialization (Dai et al., 2024; Krishnamurthy et al., 2023). Furthermore, from the convergence analysis of expert estimation, we can gain several insights for designing the router and expert networks (see Section 4). Therefore, we will investigate the effects of the cosine router on the convergence of expert estimation in this paper. For that sake, let us now present the problem setting formally.

**Problem setting.** We assume that $(X_1, Y_1), (X_2, Y_2), \ldots, (X_n, Y_n) \in \mathbb{R}^{d_1} \times \mathbb{R}$ is an i.i.d sample of size $n$ generated according to the following model

$$Y_i = f_{G_*}(X_i) + \varepsilon_i, \quad i = 1, \ldots, n, \tag{1}$$

where regression function $f_{G_*}(\cdot)$ takes the following form:

$$f_{G_*}(x) := \sum_{i=1}^{k_*} \text{Softmax}\left(\frac{(\beta_{1i}^*)^\top x}{\|\beta_{1i}^*\| \cdot \|x\|} + \beta_{0i}^*\right) \cdot h(x, \eta_i^*). \tag{2}$$

Here, the function $h(x, \eta)$ is known as *the expert function,* which we assumed to be of parametric form. Meanwhile, $(\beta_{0i}^*, \beta_{1i}^*, \eta_i^*)_{i=1}^{k_*}$ are true yet unknown parameters in the parameter space $\Theta \subset \mathbb{R} \times \mathbb{R}^{d_1} \times \mathbb{R}^{d_2}$ and $G_* := \sum_{i=1}^{k_*} \exp(\beta_{0i}^*)\delta_{(\beta_{1i}^*, \eta_i^*)}$ denotes the associated *mixing measure,* i.e. a weighted sum of Dirac measures $\delta$. Additionally, we define for any vector $v = (v_1, v_2, \ldots, v_{k_*})$ in $\mathbb{R}^{k_*}$ that $\text{Softmax}(v_i) := \exp(v_i)/\sum_{j=1}^{k_*} \exp(v_j)$. In the cosine router in equation (2), we omit the step of reducing the dimension of the input token $x$, and assume that it has already been in a low-dimensional space for simplicity. Furthermore, we assume that $X_1, X_2, \ldots, X_n$ are i.i.d. samples from some probability distribution $\mu$. Lastly, $\varepsilon_1, \varepsilon_2, \ldots, \varepsilon_n$ are independent Gaussian noise variables such that $\mathbb{E}[\varepsilon_i | X_i] = 0$ and $\text{Var}(\varepsilon_i | X_i) = \sigma^2$ for all $1 \leq i \leq n$. Notably, the Gaussian assumption is just for the simplicity of the proof argument.

**Least squares estimation (LSE).** To estimate the true parameters $(\beta_{0i}^*, \beta_{1i}^*, \eta_i^*)_{i=1}^{k_*}$ or, equivalently, to estimate the true mixing measure $G_*$, we leverage the popular least squares method (see, e.g., van de Geer, 2000). Formally, the mixing measure $G_*$ is approximated by

$$\widehat{G}_n := \arg\min_G \sum_{i=1}^n \left(Y_i - f_G(X_i)\right)^2. \tag{3}$$

Under the *exact-specified* setting, i.e., when the true number of expert $k_*$ is known, the minimum in the above equation is subject to the set of all mixing measures with $k_*$ atoms, denoted by $\mathcal{E}_{k_*}(\Theta) := \{G = \sum_{i=1}^{k_*} \exp(\beta_{0i})\delta_{(\beta_{1i}, \eta_i)} : (\beta_{0i}, \beta_{1i}, \eta_i) \in \Theta\}$. On the other hand, under the *over-specified* setting, i.e., when $k_*$ is unknown and the true model (2) is over-specified by a mixture

of $k$ experts where $k > k_*$, the minimum is subject to the set of all mixing measures with at most $k$ atoms, i.e., $\mathcal{G}_k(\Theta) := \{G = \sum_{i=1}^{k'} \exp(\beta_{0i})\delta_{(\beta_{1i},\eta_i)} : 1 \le k' \le k, \ (\beta_{0i}, \beta_{1i}, \eta_i) \in \Theta\}$.

**Universal assumptions.** In the sequel, we implicitly impose four following mild assumptions on the model parameters, which were widely used in previous works (Nguyen et al., 2024c), unless stating otherwise:

*(A.1) Convergence of LSE:* The parameter space $\Theta \subseteq \mathbb{R} \times \mathbb{R}^{d_1} \times \mathbb{R}^{d_2}$ is compact, while the input space $\mathcal{X} \subseteq \mathbb{R}^{d_1}$ is bounded. This helps ensure the convergence of least squares estimation.

*(A.2) Distinct experts:* The true parameters $\eta_1^*, \ldots, \eta_{k_*}^*$ are distinct so that the experts $h(\cdot, \eta_1^*), \ldots, h(\cdot, \eta_{k_*}^*)$ are different from each other. Furthermore, the expert function $h(\cdot, \eta)$ is Lipschitz continuous w.r.t its parameters and bounded.

*(A.3) Identifiability of the MoE:* In order that the cosine routing MoE is identifiable, i.e., $f_G(x) = f_{G_*}(x)$ for almost every $x$ implies that $G \equiv G_*$, we let $\beta_{0k_*}^* = 0$.

*(A.4) Input-dependent router:* To ensure that the router is input-dependent, we assume that at least one among the parameters $\beta_{11}^*, \ldots, \beta_{1k_*}^*$ is non-zero.

**Technical challenges.** The normalization of parameters in the cosine router leads to a fundamental challenge in theory. In particular, to establish parameter and expert estimation rates based on the convergence rate of the regression function, we rely on decomposing the discrepancy $f_{\widehat{G}_n}(x) - f_{G_*}(x)$ into a combination of linearly independent terms by applying Taylor expansions to the product of the softmax's numerator and the expert function, i.e. $H(x, \beta_1, \eta) := \exp\left(\frac{\beta_1^\top x}{\|\beta_1\| \cdot \|x\|}\right) h(x, \eta)$. However, the normalization of $\beta_1$ in the cosine router leads to an intrinsic interaction among the elements of the parameter $\beta_1$ via the following PDE:

$$\beta_1^\top \frac{\partial H}{\partial \beta_1}(x, \beta_1, \eta) = 0. \tag{4}$$

Although parameter interactions expressed in the language of PDEs have been observed in Nguyen et al. (2024c), the structure of the above interaction is much more sophisticated (even hold for the first-order derivatives while those in Nguyen et al. (2024c) occurs only when taking the second-order derivatives). Thus, this PDE induces several linearly dependent terms in the Taylor expansion, and we have to aggregate their coefficients in order to form the desired combination of linearly independent terms. Then, the resulting coefficients become complex, thereby negatively affecting the convergence of expert estimation. To the best of our knowledge, such a phenomenon with the cosine router has never been observed in previous works.

**Main contributions.** In this work, we develop a comprehensive theoretical analysis of regression function estimation as well as parameter and expert estimations under the cosine router MoE model (1). Our contributions are two-fold and can be summarized as follows (see also Table 1):

**1. Cosine router:** Equipped with the cosine router, we demonstrate that under both the exact-specified and the over-specified settings, the rates for estimating ground-truth parameters $\beta_{0i}^*, \beta_{1i}^*$ and $\eta_i^*$ are slower than any polynomial rates and, therefore, could be as slow as $\mathcal{O}_P(1/\log^\tau(n))$, where $\tau > 0$ is some constant. These slow rates are attributed to the internal interaction among router parameters expressed by the PDE in equation (4). As a result, the estimation rates for experts $h(\cdot, \eta_i^*)$ are also negatively affected, and could be of order $\mathcal{O}_P(1/\log^\tau(n))$.

**2. Perturbed cosine router:** In response, we propose a novel router called *perturbed cosine router* in which we add noises to the $L^2$ norms of the token representations and the expert embeddings. This not only helps stabilize the router but also eliminates the intrinsic interaction in equation (4). Additionally, we also establish identifiability conditions to characterize expert functions that have faster estimation rates than others under the exact-specified and over-specified settings, respectively. Those conditions indicate that the rates for estimating experts, which are formulated as feed-forward networks (FFNs) with widely used activation functions such as $\mathrm{ReLU}$ and $\mathrm{GeLU}$, are significantly improved, ranging from $\mathcal{O}_P(\sqrt[4]{\log(n)/n})$ to $\mathcal{O}_P(\sqrt{\log(n)/n})$.

**Outline.** In Section 2, we establish the convergence rates of parameter and expert estimations under the setting of the cosine router MoE. Then, we derive these rates when the cosine router is replaced by the perturbed cosine router in Section 3. Based on these theoretical results, we provide a few practical

Table 1: Summary of worst possible estimation rates for linear experts, polynomial experts and FFN experts equipped with the ReLU activation function under the MoE with linear router (Nguyen et al., 2024c), cosine router (ours) and perturbed cosine routers (ours).

| Routers/ Experts | Linear: $a^\top x + b$ | Polynomial: $(a^\top x + b)^p, p \geq 2$ | ReLU FFN |
|---|---|---|---|
| Linear | $1/\log^\tau(n)$ | $1/\log^\tau(n)$ | $n^{-1/4}$ |
| Cosine | $1/\log^\tau(n)$ | $1/\log^\tau(n)$ | $1/\log^\tau(n)$ |
| Perturbed cosine | $1/\log^\tau(n)$ | $n^{-1/4}$ | $n^{-1/4}$ |

implications in Section 4. Next, we empirically verify the (theoretical) benefits of the perturbed cosine router over the cosine router under both the synthetic and real data settings in Section 5 before concluding the paper in Section 6. Finally, proofs and additional details of the experiments are deferred to the Appendices.

**Notations.** We let $[n]$ stand for the set $\{1, 2, \ldots, n\}$ for any $n \in \mathbb{N}$. Next, for any set $S$, we denote $|S|$ as its cardinality. For any vector $v \in \mathbb{R}^d$ and $\alpha := (\alpha_1, \alpha_2, \ldots, \alpha_d) \in \mathbb{N}^d$, we let $v^\alpha = v_1^{\alpha_1} v_2^{\alpha_2} \ldots v_d^{\alpha_d}$, $|v| := v_1 + v_2 + \ldots + v_d$ and $\alpha! := \alpha_1! \alpha_2! \ldots \alpha_d!$, while $\|v\|$ stands for its $L^2$-norm value. Lastly, for any two positive sequences $(a_n)_{n \geq 1}$ and $(b_n)_{n \geq 1}$, we write $a_n = \mathcal{O}(b_n)$ or $a_n \lesssim b_n$ if there exists $C > 0$ such that $a_n \leq C b_n$ for all $n \in \mathbb{N}$. Meanwhile, the notation $a_n = \mathcal{O}_P(b_n)$ indicates that $a_n/b_n$ is stochastically bounded.

## 2 COSINE ROUTER MIXTURE OF EXPERTS

In this section, we characterize the parameter and expert estimation rates under the over-specified setting of the cosine router MoE. We first start with the convergence rate of the regression function estimation $f_{\widehat{G}_n}$ to the true regression function $f_{G_*}$ under the $L^2(\mu)$ norm in the following theorem:

**Theorem 1.** *Given the least-square estimator $\widehat{G}_n$ defined in equation* (3)*, the regression estimator $f_{\widehat{G}_n}(.)$ converges to the true regression function $f_{G_*}(.)$ at the following rate:*

$$\|f_{\widehat{G}_n} - f_{G_*}\|_{L^2(\mu)} = \mathcal{O}_P(\sqrt{\log(n)/n}).$$

The proof of Theorem 1 is in Appendix B.1. The result of Theorem 1 indicates that the regression estimation rate is parametric. Therefore, as long as we can establish the lower bound $\|f_{\widehat{G}_n} - f_{G_*}\|_{L^2(\mu)} \gtrsim \mathcal{L}(\widehat{G}_n, G_*)$ where $\mathcal{L}$ is some loss function among parameters, we arrive at the parameter estimation rate $\mathcal{L}(\widehat{G}_n, G_*) = \mathcal{O}_P(\sqrt{\log(n)/n})$. This approach is the key component of the convergence rates of parameter and expert estimations under the cosine router MoE. In the sequel, we will consider the over-specified setting of the cosine router, while the results for the exact-specified setting will be presented in Appendix A.2.

Recall that under the over-specified setting, the true number of experts $k_*$ is unknown. Then, based on the notion of Voronoi cells (Manole & Ho, 2022), we will construct a Voronoi loss function among parameters tailored to this setting.

**Voronoi loss.** Let $G$ be a mixing measure with $k'$ atoms $\omega_i := (\beta_{1i}, \eta_i)$. Then, we distribute these atoms to the Voronoi cells generated by the atoms $\omega_j^* := (\beta_{1j}^*, \eta_j^*)$ of $G_*$, which are defined as

$$\mathcal{A}_j \equiv \mathcal{A}_j(G) := \{i \in [k'] : \|\omega_i - \omega_j^*\| \leq \|\omega_i - \omega_\ell^*\|, \forall \ell \neq j\}. \tag{5}$$

Then, the Voronoi loss of interest is given by

$$\mathcal{L}_{1,r}(G, G_*) := \sum_{j=1}^{k_*} \Big| \sum_{i \in \mathcal{A}_j} \exp(\beta_{0i}) - \exp(\beta_{0j}^*) \Big| + \sum_{j=1}^{k_*} \sum_{i \in \mathcal{A}_j} \exp(\beta_{0i}) \Big[ \|\Delta\beta_{1ij}\|^r + \|\Delta\eta_{ij}\|^r \Big],$$

where $r \geq 1$ is some constant, $\Delta\beta_{1ij} := \beta_{1i} - \beta_{1j}^*$ and $\Delta\eta_{ij} := \eta_i - \eta_j^*$.

Note that, due to the parameter interaction inside the cosine router captured by the PDE (4), the lower bound $\|f_{\widehat{G}_n} - f_{G_*}\|_{L^2(\mu)} \gtrsim \mathcal{L}_{1,r}(\widehat{G}_n, G_*)$ does not hold true, and thus, we cannot achieve

the desired bound $\mathcal{L}_{1,r}(\widehat{G}_n, G_*) = \mathcal{O}_P(\sqrt{\log(n)/n})$ mentioned at the beginning of Section 2. By contrast, we show in Appendix B.3 an opposed result to the previous lower bound, saying that

$$\lim_{\varepsilon \to 0} \inf_{G \in \mathcal{E}_{k_*}(\Theta):\mathcal{L}_{1,r}(G,G_*) \leq \varepsilon} \|f_G - f_{G_*}\|_{L^2(\mu)}/\mathcal{L}_{1,r}(G, G_*) = 0,$$

for any $r \geq 1$. This result implies the following minimax lower bound of parameter estimation:

**Theorem 2.** *Under the over-specified setting, the following minimax lower bound of estimating $G_*$*

$$\inf_{\overline{G}_n \in \mathcal{G}_k(\Theta)} \sup_{G \in \mathcal{G}_k(\Theta) \setminus \mathcal{G}_{k_*-1}(\Theta)} \mathbb{E}_{f_G}[\mathcal{L}_{1,r}(\overline{G}_n, G)] \gtrsim n^{-1/2},$$

*holds true for any $r \geq 1$, where $\mathbb{E}_{f_G}$ indicates the expectation taken w.r.t the product measure with $f_G^n$ and the infimum is over all estimators taking values in $\mathcal{G}_k(\Theta)$.*

See Appendix B.3 for the proof of Theorem 2. There are two main implications of the above result:

**(i) Parameter estimation rates.** The minimax lower bound together with the formulation of $\mathcal{L}_{1,r}$ indicate that at least one among the rates for estimating parameters $\beta_{1j}^*$, $\beta_{0i}^*$, $\eta_j^*$ is slower than any polynomial rates $\mathcal{O}_P(n^{-1/2r})$ and, thus, could be of order $\mathcal{O}_P(1/\log^\tau(n))$, for some constant $\tau > 0$.

**(ii) Router estimation rates:** When the estimation rate of either $\beta_1^*$ or $\beta_0^*$ is slower than any polynomial rates, since the softmax function is Lipschitz w.r.t the Euclidean norm (Gao & Pavel, 2018), we deduce that the worst possible rate of estimating the cosine router or the mixture weights in equation (2) could also be slower than any polynomial rates. In practice, the router and the expert networks are trained simultaneously (see Section 1.2 in (Shazeer et al., 2017)). Thus, the slow convergence of the router might decelerate the model convergence.

**(iii) Expert estimation rates.** Assume that $\widehat{G}_n := \sum_{i=1}^{k_*} \exp(\widehat{\beta}_{0i})\delta_{(\widehat{\beta}_{1i}^n, \widehat{\eta}_i^n)}$. Since the expert $h(\cdot, \eta)$ is Lipschitz continuous, it follows that

$$\sup_x |h(x, \widehat{\eta}_i^n) - h(x, \eta_j^*)| \lesssim \|\widehat{\eta}_i^n - \eta_j^*\|. \tag{6}$$

Consequently, the worst possible rate for estimating the expert $h(x, \eta_j^*)$ is identical to the worst possible rate for estimating the parameter $\eta_j^*$. For instance, if the expert function takes the polynomial form of $h(x, \eta) := (\eta x)^2$ (considered in (Mendes & Jiang, 2012; Nguyen et al., 2021; 2024c)), where we assume $x \in \mathbb{R}$ for simplicity, then we have $|h(\hat{\eta}, x) - h(x, \eta^*)| = |\hat{\eta} - \eta^*| \cdot |\hat{\eta} + \eta^*| \cdot |x|$. As a result, the expert estimation rate is exactly the parameter estimation rate. Thus, when the estimation rate of $\eta_j^*$ is slower than any polynomial rates, the worst possible estimation rates for the experts $h(\cdot, \eta_j^*)$ could also be slower than any polynomial rates and be as slow as $\mathcal{O}_P(1/\log^\tau(n))$. This indicates that the cosine router is even less sample efficient than the linear router (see also Table 1). Note that by employing techniques of partitioning the input space in (Nguyen et al., 2024b), we can show that these results still hold when using the Top-$K$ sparse softmax gating function. Therefore, in order to improve the sample efficiency while preserving the robustness to the representation collapse, we need to slightly modify the structure of the cosine router in the next section.

## 3 PERTURBED COSINE ROUTER MIXTURE OF EXPERTS

In this section, we demonstrate that the pessimistic non-polynomial convergence rates of parameter and expert estimation under the cosine router can be easily circumvented by the widely used technique in practice to stabilize the cosine router: adding noises to the $L^2$ norm in the cosine router. Although this perturbation technique and the cosine router have been well studied in the literature, we would like to emphasize that to the best of our knowledge, our work is the first to be aware of combining these two methods together which we refer to as the *perturbed cosine router MoE*. We now present the formulation of a MoE with the perturbed cosine router under the regression setting.

**Problem setup for the perturbed cosine router MoE model**. We assume that an i.i.d. sample of size $n$: $(X_1, Y_1), (X_2, Y_2), \ldots, (X_n, Y_n) \in \mathbb{R}^{d_1} \times \mathbb{R}$ is generated according to the model

$$Y_i = g_{G_*}(X_i) + \varepsilon_i, \quad i = 1, \ldots, n, \tag{7}$$

where regression function $g_{G_*}(\cdot)$ takes the following form:

$$g_{G_*}(x) := \sum_{i=1}^{k_*} \text{Softmax}\left(\frac{(\beta_{1i}^*)^\top x}{(\|\beta_{1i}^*\| + \tau_1) \cdot (\|x\| + \tau_2)} + \beta_{0i}^*\right) \cdot h(x, \eta_i^*). \tag{8}$$

Here, $\tau_1, \tau_2 > 0$ are two noise hyper-parameters. The main difference between the two regression functions $f_{G_*}$ and $g_{G_*}$ is the noise hyper-parameters $\tau_1, \tau_2$ that we add to the norms of the expert embeddings $\beta_{1i}^*$ and the token input $x$, which leads to the perturbed cosine router. By doing so, the parameter interaction inside the router as in equation (4) does not occur. More specifically, let us denote $\widetilde{H}(x, \beta_1, \eta) := \exp\left(\frac{\beta_1^\top x}{(\|\beta_1\|+\tau_1)\cdot(\|x\|+\tau_2)}\right)h(x,\eta)$, then it can be verified that $\beta_1^\top \frac{\partial \widetilde{H}}{\partial \beta_1}(x, \beta_1, \eta) \neq 0$.

**Least squares estimation.** Similar to the cosine router setting, we can estimate the unknown ground-truth parameters $(\beta_{0i}^*, \beta_{1i}^*, \eta_i^*)_{i=1}^{k_*}$ using the least-square estimator, which is given by:

$$\widetilde{G}_n := \arg\min_G \sum_{i=1}^n \left(Y_i - g_G(X_i)\right)^2. \tag{9}$$

In the following theory, we provide a convergence rate of regression function estimation under the perturbed cosine router MoE model.

**Theorem 3.** *Given a least squares estimator $\widetilde{G}_n$ defined in equation* (9)*, the regression function estimation $g_{\widetilde{G}_n}(\cdot)$ admits the following convergence rate:*

$$\|g_{\widetilde{G}_n} - g_{G_*}\|_{L^2(\mu)} = \mathcal{O}_P(\sqrt{\log(n)/n}). \tag{10}$$

Proof of Theorem 3 is in Appendix C.1. The result of Theorem 3 proves that the regression function estimation rate $\mathcal{O}_P(\sqrt{\log(n)/n})$ under the perturbed cosine router MoE is of the same order as that with the vanilla cosine router in Theorem 1. Following the similar proof strategy in the cosine router MoE in Section 2 for capturing the convergence rates of parameter and expert estimations under the perturbed cosine router MoE model, it is sufficient to establish the lower bound between the difference of regression functions and the difference of parameters under both the exact-specified and over-specified settings.

In this section, we study the over-specified setting of the perturbed cosine router. The results for the exact-specified setting of the perturbed cosine router is in Appendix A.2.

We now derive a condition called *strong identifiability* on the expert function $h(\cdot, \eta)$ to identify which experts exhibit faster estimation rates than others under the over-specified setting.

**Definition 1** (Strong identifiability). *An expert function $x \mapsto h(x, \eta)$ is called strongly identifiable if it is twice differentiable with respect to its parameter $\eta$, and the set of functions in $x$*

$$\left\{\frac{\partial^{|\alpha_1|+|\alpha_2|}\widetilde{H}}{\partial \beta_1^{\alpha_1} \partial \eta^{\alpha_2}}(x, \beta_{1i}, \eta_i) : \alpha_1 \in \mathbb{N}^{d_1}, \alpha_2 \in \mathbb{N}^{d_2}, 0 \leq |\alpha_1| + |\alpha_2| \leq 2\right\},$$

*is linearly independent for almost every $x$ for any $k \geq 1$ and distinct parameters $\eta_1, \ldots, \eta_k$, where we denote $\widetilde{H}(x, \beta_1, \eta) := \exp(\frac{\beta_1^\top x}{(\|\beta_1\|+\tau_1)\cdot(\|x\|+\tau_2)})h(x, \eta)$.*

**Example.** For experts formulated as neural networks, i.e. $h(x, (a, b)) = \phi(a^\top x + b)$, if the activation $\phi$ is selected as $\phi(z) = \mathrm{ReLU}(z)$, $\phi(z) = \tanh(\cdot)$ or $\phi(z) = z^p$ for $p \geq 2$, then they are strongly identifiable. Conversely, a linear expert $h(x, (a, b)) = a^\top x + b$ fails to meet the strong identifiability the condition.

To capture the convergence behavior of expert estimation rate under the over-specified setting in Theorem 4, we will use the Voronoi loss defined as follows:

$$\mathcal{L}_2(G, G_*) := \sum_{j=1}^{k_*} \left|\sum_{i \in \mathcal{A}_j} \exp(\beta_{0i}) - \exp(\beta_{0j}^*)\right| + \sum_{j \in [k_*]:|\mathcal{A}_j|=1} \sum_{i \in \mathcal{A}_j} \exp(\beta_{0i})\left[\|\Delta\beta_{1ij}\| + \|\Delta\eta_{ij}\|\right]$$

$$+ \sum_{j \in [k_*]:|\mathcal{A}_j|>1} \sum_{i \in \mathcal{A}_j} \exp(\beta_{0i})\left[\|\Delta\beta_{1ij}\|^2 + \|\Delta\eta_{ij}\|^2\right]. \tag{11}$$

**Theorem 4.** *Suppose that the expert function $h(x, \eta)$ satisfies the strong identifiability condition in Definition 1, then the following $L^2$-lower bound holds for any mixing measure $G \in \mathcal{G}_k(\Theta)$:*

$$\|g_G - g_{G_*}\|_{L^2(\mu)} \gtrsim \mathcal{L}_2(G, G_*).$$

*Furthermore, this bound and the result in Theorem 3 imply that $\mathcal{L}_2(\widetilde{G}_n, G_*) = \mathcal{O}_P(\sqrt{\log(n)/n})$.*

The proof of Theorem 4 is in Appendix C.3. A few comments regarding this theorem are in order:

**(i) Parameter estimation rates.** Under the over-specified setting, parameters $\beta_{1j}^*, \eta_j^*$ which are fitted by one atom, i.e. $|\mathcal{A}_j(\widetilde{G}_n)| = 1$, share the same estimation rate of order $\mathcal{O}_P(\sqrt{\log(n)/n})$. Meanwhile, those for parameters fitted by more than one atom, i.e. $|\mathcal{A}_j(\widetilde{G}_n)| > 1$, are slightly slower, standing at order $\mathcal{O}_P(\sqrt[4]{\log(n)/n})$.

**(ii) Expert estimation rates.** Given the above parameter estimation rates and the inequality (6), we observe that the rates for estimating strongly identifiable experts $h(\cdot, \eta_j^*)$ range from $\mathcal{O}_P(\sqrt[4]{\log(n)/n})$ to $\mathcal{O}_P(\sqrt{\log(n)/n})$. Notably, these rates apply even for polynomial experts of degree at least two, i.e. $h(x, (a, b)) = (a^\top x + b)^p$ with $p \geq 2$, as they satisfy the strong identifiability condition. By contrast, the estimation rates for polynomial experts when using the vanilla cosine router (see Theorem 2) and the linear router (see Theorem 4.6, (Nguyen et al., 2024c)) are significantly slower and could be of order $\mathcal{O}_P(1/\log^\tau(n))$, where $\tau > 0$ is some constant (see also Table 1). This observation highlights that our proposed perturbed cosine router is more sample efficient than both the linear router and the cosine router.

## 4 PRACTICAL IMPLICATIONS

We now discuss two important practical implications from the theoretical results of the paper.

**1. Router and expert design:** From the benefits of the perturbed cosine router for the expert estimation of MoE models, our theories suggest that when using the cosine router to avoid the representation collapse, practitioners should add noises to $L^2$ norms of the token hidden representations and the expert embeddings to achieve a favorable performance. Additionally, the strong identifiability condition also verifies the advantages of using non-linear expert networks over linear ones.

**2. Misspecified settings.** Thus far in the paper, we have only considered well-specified settings, namely, the data are assumed to be sampled from the (perturbed) cosine router MoE. Although it may look restrictive, the results under this setting lay an important foundation for a more realistic misspecified setting where the data are not necessarily generated from those models. Under that misspecified setting, we assume that the data are generated from a regression framework as in equation (1) but with an arbitrary regression function $q(\cdot)$, which is not a (perturbed) cosine router MoE. Then, we can demonstrate that the LSE $\widehat{G}_n$ converges to a mixing measure $\overline{G} \in \arg\min_{G \in \mathcal{G}_k(\Theta)} \|q - f_G\|_{L^2(\mu)}$, where $f_G(\cdot)$ is a regression function taking the form of the (perturbed) cosine router MoE. Furthermore, the optimal mixing measure will be in the boundary of the parameter space $\mathcal{G}_k(\Theta)$, namely, $\overline{G}$ has $k$ atoms. Thus, as $n$ becomes sufficiently large, $\widehat{G}_n$ also has $k$ atoms. The insights from our theories for the well-specified setting indicate that the Voronoi losses can be used to obtain the estimation rates of individual parameters of the LSE $\widehat{G}_n$ to those of $\overline{G}$ and therefore, achieve the following expert estimation rates under the misspecified settings, which will be empirically validated via numerical experiments in Appendix F:

*(2.1) Cosine router MoE:* the worst expert estimation rate could be as slow as $\mathcal{O}_P(1/\log^\tau(n))$ for some $\tau > 0$. It indicates that we still need an exponential number of data (roughly $\exp(1/\epsilon^\tau)$ where $\epsilon$ is the desired approximation error) to estimate the experts as well as select important experts.

*(2.2) Perturbed cosine router MoE:* the slowest expert estimation rate is of order $\mathcal{O}_P(n^{-1/4})$. Thus, we only need a polynomial number of data (roughly $\epsilon^{-4}$) to estimate the experts. This explains why the perturbed cosine router is a solution to the parameter estimation problem, or more generally, the expert estimation problem of the MoE models.

However, the convergence analysis under the misspecified setting suffers from the challenges of understanding the universal approximation power of the (perturbed) cosine router, which have remained elusive in the literature. However, since this is beyond the scope of our paper, we leave it for future development.

## 5 EXPERIMENTS

In this section, we first conduct numerical experiments on synthetic data (cf. Section 5.1), and then carry out experiments with real data on language modeling (cf. Section 5.2) and domain generalization (cf. Section 5.3) tasks. Our main goal is to empirically demonstrate the efficacy of the perturbed cosine router over the vanilla cosine router and the linear router in MoE models.

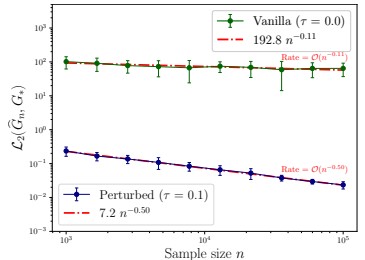 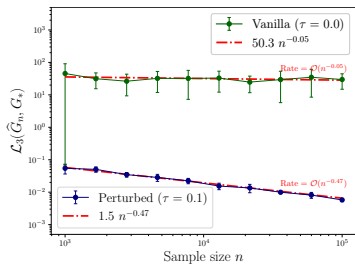

(a) Exact-specified setting with $k = k_* = 8$ experts    (b) Over-specified setting with $k = k_* + 1 = 9$

Figure 1: Logarithmic plots displaying empirical convergence rates. Subfigures 1a and 1b depict the empirical averages of the Voronoi losses $\mathcal{L}_3(\widehat{G}_n, G_*)$ (cf. equation (12)) and $\mathcal{L}_2(\widehat{G}_n, G_*)$ (cf. equation (11)) for the exact and over-specified settings, respectively. The blue lines depict the Voronoi loss associated with the perturbed router, whereas the green lines are indicative of the Voronoi loss associated with the standard cosine router. The red dash-dotted lines are used to illustrate the fitted lines for determining the empirical convergence rate.

## 5.1 NUMERICAL EXPERIMENTS

We first perform numerical experiments on synthetic data to empirically verify the theoretical convergence rates of the least squares estimation for both perturbed and vanilla cosine router MoE models. We generate synthetic data based on the model described in equation (1). Specifically, we generate $\{(X_i, Y_i)\}_{i=1}^n \subset \mathbb{R}^d \times \mathbb{R}$ by first sampling $X_i \sim \mathrm{Uniform}([-1, 1]^d)$ for $i = 1, \ldots, n$. Then, we generate $Y_i$ according to the model: $Y_i = f_{G_*}(X_i) + \varepsilon_i$ for $i \in [n]$, where the regression function $f_{G_*}(\cdot)$ is defined as: $f_{G_*}(x) := \sum_{i=1}^{k_*} \mathrm{Softmax}\left( \frac{(\beta_{1i}^*)^\top x}{(\|\beta_{1i}^*\| + \tau) \cdot (\|x\| + \tau)} + \beta_{0i}^* \right) \cdot \phi\left( (a_i^*)^\top x + b_i^* \right)$. The input data dimension is set at $d = 32$. We employ $k_* = 8$ experts of the form $\phi\left( (a_i^*)^\top x + b_i^* \right)$, where the activation function $\phi$ is set to be the ReLU function. The details of the values of the parameters as well as the training procedure are in Appendix E.1.

**Results.** Two experimental settings are examined: (1) Exact-specified, and (2) Over-specified. In the exact-specified setting, the model is fitted with the same number of experts as the data generation model, specifically $k = k_* = 8$. In the over-specified setting, the model includes one additional expert, totaling $k = k_* + 1 = 9$ experts. In each setting, experiments are conducted using both the standard and the perturbed cosine routers, with $\tau$ set to zero for the standard router and $0.1$ for the perturbed router. For each experiment, we calculate the Voronoi losses for every model and report the mean values for each sample size in Figure 1. Error bars representing two standard deviations are also shown. In Figure 1a, the empirical convergence rates of both the standard and perturbed routers are analyzed under the exact-specified setting. The perturbed router shows a rapid convergence rate of $\mathcal{O}(n^{-0.5})$, while the standard vanilla router has a noticeably slower rate of $\mathcal{O}(n^{-0.11})$. Similarly, in Figure 1b, the convergence rates are assessed for the same routers under the over-specified setting. Here, the perturbed router again shows a faster convergence rate of $\mathcal{O}(n^{-0.47})$, compared to the cosine router's slower rate of $\mathcal{O}(n^{-0.05})$.

**Misspecified settings.** In Appendix F, we will also conduct numerical experiments for comparing the sample efficiency of the cosine router and its perturbed variant under the setting where the data are generated from the regression framework with the same regression function.

## 5.2 LANGUAGE MODELING

In this section, we focus on the language modeling task (Bahl et al., 1983), a fundamental challenge in natural language processing that involves predicting the next word or character in a sequence to evaluate a model's ability to generate and understand text. To assess how different routers influence the model's ability to capture linguistic structures and enhance performance across varying levels of textual granularity, we compare the performance of perturbed and vanilla cosine router Mixture of Experts (MoEs) on both character-level (Graves, 2013) and word-level (Bengio et al., 2000) tasks.

**Datasets.** We evaluate the model's pre-training capabilities on character-level language modeling using Enwik8 and Text8 datasets (Mahoney, 2011), and assess its word-level language modeling performance on Wikitext-103 (Merity et al., 2016).

Table 2: Performance of vanilla and perturbed cosine routers on language modeling tasks.

| Router/Experts | Enwik8 (BPC ↓) | | Text8 (BPC ↓) | | Wikitext-103 (PPL ↓) | |
|---|---|---|---|---|---|---|
| | Small | Medium | Small | Medium | Small | Medium |
| Cosine | 1.213 | 1.161 | 1.310 | 1.271 | 90.070 | 38.018 |
| Perturbed cosine | **1.197** | **1.147** | **1.303** | **1.251** | **89.910** | **37.859** |

**Metrics.** To quantify the performance of our perturbed cosine router relative to the vanilla cosine one, we utilize the Bit Per Character (BPC) metric (Graves, 2013) for character-level language modeling and Perplexity (PPL) (Jelinek et al., 1977) for word-level language modeling tasks.

**Architecture and training procedure.** In order to alleviate the representation collapse issue associated with estimating routing scores in the original space, we follow Chi et al. (2022) to first project input representations on lower-dimensional space and parameterize experts with corresponding lower-dimensional embeddings. Subsequently, we calculate the routing scores of inputs and embeddings in this reduced-dimensional space using our proposed perturbed cosine router. Our experiments adopt the Switch Transformer (Fedus et al., 2021), which is fundamentally a sparse variant of the T5 encoder-decoder (Raffel et al., 2020), with MoE layers replacing the MLPs. Detailed information regarding the datasets, metrics, training setup, and hyperparameters for this task is provided in Appendix E.2.

**Results.** The empirical advantage of our proposed cosine router over the vanilla version when applied to language modeling tasks is demonstrated in Table 2. The results indicate that the perturbed cosine router enhances the performance of the original cosine router in all datasets across both small and medium configurations. It notably improves results for the Enwik8 and Text8 datasets at various scales and slightly outperforms the original cosine router for the Wikitext-103 dataset.

## 5.3 DOMAIN GENERALIZATION

We conduct experiments on the applications of MoE models in domain generalization. Our objective is to empirically demonstrate the efficacy of our proposed perturbed cosine router over the vanilla cosine router in this field. Domain generalization (Zhou et al., 2023a) aims to generalize a model's performance to unseen test domains with distributions different from those encountered during training. Specifically, in domain generalization, a model is expected to leverage multiple training datasets gathered from various domains and exhibit robustness to domain shifts during testing. Such ability of out-of-distribution generalization largely hinges on the model's capability to incorporate invariances across multiple domains (Li et al., 2023). Given that distribution shifts in data correspond to distribution shifts in (visual) attributes (Wiles et al., 2022), capturing these diverse attributes and aligning them with invariant correlations is crucial. Mixture of Experts emerges as a powerful tool for efficiently capturing these visual attributes, and it has been proven effective in enhancing performance in domain generalization (Li et al., 2023). Therefore, we further justify the effectiveness of our perturbed cosine router in domain generalization.

**Datasets.** We followed the experimental setting of Li et al. (2023) and evaluated our method using 5 benchmark datasets in DomainBed: PACS, VLCS, OfficeHome, TerraIncognita, and DomainNet. Each dataset is comprised of images for classification tasks from different domains.

**Architecture.** Following Gulrajani & Lopez-Paz (2021), we conduct experiments on ViT-S/16, which has an input patch size of $16 \times 16$, comprising 6 heads in multi-head attention layers, and a total of 12 transformer blocks. We adopt a *last-two* two-layer configuration, where each MoE block comprises 6 experts. The router selects the top 2 out of 6 experts for each image patch.

**Training procedure and result.** We follow the training-domain validation procedure outlined in (Li et al., 2023; Gulrajani & Lopez-Paz, 2021), where each training domain is split into training and validation subsets. The final overall validation set consists of the validation subsets from all training domains. Subsequently, we select the model with the highest performance on the overall validation set. To ensure fair comparisons, the results are averaged over three runs.

Table 3 summarizes the experimental results. For each dataset, we report the average results across test domains. The results demonstrate that our perturbed cosine router consistently outperforms the linear and vanilla cosine router across all datasets, thereby convincingly justifying the effectiveness of adding noise to cosine routers. Detailed performances for each domain are reported in Tables 4 and 5.

Table 3: Average out-of-distribution test accuracies.

| Router/Experts | PACS | VLCS | OfficeHome | TerraIncognita | DomainNet | Avg. |
|---|---|---|---|---|---|---|
| Linear | 86.33 | 78.15 | 73.02 | 41.30 | 48.19 | 65.40 |
| Cosine | 87.22 | 78.99 | 73.27 | 45.55 | 48.45 | 66.70 |
| Perturbed cosine | **89.36** | **80.01** | **74.09** | **49.87** | **48.51** | **68.37** |

Table 4: Per-domain performance of PACS, VLCS, OfficeHome, TerraIncognita.

| | Router/Experts | A | C | P | S |
|---|---|---|---|---|---|
| PACS | Linear | 87.29 | 81.20 | **98.50** | 78.34 |
| | Cosine | 89.24 | 86.11 | 97.60 | 75.92 |
| | Perturbed cosine | **89.87** | **86.97** | 97.90 | **82.68** |

| | Router/Experts | C | L | S | V |
|---|---|---|---|---|---|
| VLCS | Linear | 97.53 | 63.65 | 74.09 | 77.33 |
| | Cosine | 98.59 | 67.42 | 70.88 | **79.07** |
| | Perturbed cosine | 98.59 | **67.80** | **74.70** | 78.95 |

| | Router/Experts | A | C | P | R |
|---|---|---|---|---|---|
| OfficeHome | Linear | 72.99 | 57.27 | 79.03 | 82.78 |
| | Cosine | 73.40 | 57.27 | 78.69 | 83.70 |
| | Perturbed cosine | **74.64** | **57.85** | **79.59** | **84.27** |

| | Router/Experts | L100 | L30 | L43 | L46 |
|---|---|---|---|---|---|
| TerraIncognita | Linear | 45.99 | 28.51 | 54.66 | 36.05 |
| | Cosine | 50.00 | 37.49 | 53.02 | 41.67 |
| | Perturbed cosine | **57.59** | **43.30** | **56.93** | 41.67 |

Table 5: Per-domain performance of DomainNet.

| | Router/Experts | clipart | infograph | painting | quickdraw | real | sketch |
|---|---|---|---|---|---|---|---|
| DomainNet | Linear | **69.11** | **24.95** | 54.81 | 16.88 | 68.95 | 54.41 |
| | Cosine | 68.05 | 24.48 | **55.75** | 17.39 | 69.41 | 55.59 |
| | Perturbed | 68.31 | 24.52 | 55.03 | **17.90** | **69.46** | **55.83** |

## 6 CONCLUSION

In this paper, we investigate the impacts of the cosine router on the convergence rates of least squares estimation in MoE models. We figure out that owing to the parameter interaction inside the cosine router expressed by a PDE, the rates for estimating parameters and experts are slower than any polynomial rates and therefore, could be as slow as $\mathcal{O}_P(1/\log^\tau(n))$. In response to this issue, we propose using the perturbed cosine router where we add noises to the $L^2$ norms of the token representations and the expert embeddings in the cosine router in order to eliminate the previous parameter interaction. Equipped with this novel router, we demonstrate that if the expert function satisfies the strong identifiability condition, then the parameter and expert estimation rates are significantly improved to be of polynomial orders. Finally, we conduct several experiments on both synthetic and real-world data to empirically justify the theoretical results.

There are a few limitations in our current analysis. First of all, the assumption that the data are sampled from the (perturbed) cosine router MoE is often violated in real-world settings. However, as discussed in Section 4, our theories can totally be extended to a more realistic misspecified setting where the data are not necessarily generated from those models, which we leave for future development. Second, since the ground-truth parameters are implicitly assumed to be independent of the sample size $n$, the parameter and expert estimation rates presented in this work are point-wise rather than uniform. To deal with this problem, we can utilize the techniques for characterizing the uniform parameter estimation rates in traditional mixture models (see (Heinrich & Kahn, 2018; Do et al., 2023; Yan et al., 2025)). Nevertheless, since the adaptation of those techniques to the setting of the (perturbed) cosine router MoE is still challenging due to the complex structures of the (perturbed) cosine router, we believe that further technical tools need to be developed to achieve the desired uniform estimation rates.

ACKNOWLEDGEMENTS

NH acknowledges support from the NSF IFML 2019844 and the NSF AI Institute for Foundations of Machine Learning.

REPRODUCIBILITY STATEMENT

To facilitate the reproduction of our empirical results, we present detailed descriptions of the data and the experimental setup in Section 5 and Appendix E. We will release our code upon the acceptance of our submission. All datasets used in this study are publicly available, enabling full replication of our experiments.

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

# Supplementary Material for "Statistical Advantages of Perturbing Cosine Router in Mixture of Experts"

In this supplementary material, we first explore the exact-specified settings of the (perturbed) cosine router MoE model in Appendix A. Next, we provide proofs of theoretical results associated with the cosine router MoE and its perturbed counterpart in Appendix B and Appendix C, respectively. Those proofs are partially supported by auxiliary results presented in Appendix D. Subsequently, in Appendix E, we specify the details for the experiments performed in Section 5. Finally, we conduct further numerical experiments on the convergence of least squares estimation under the misspecified settings in Appendix F.

## A ADDITIONAL RESULTS

In this appendix, we provide the convergence analysis of parameter and expert estimation under the exact-specified settings of the cosine router MoE and its perturbed variant in Appendix A.1 and Appendix A.2, respectively.

### A.1 EXACT-SPECIFIED SETTING OF THE COSINE ROUTER MOE

Firstly, we start with the exact-specified setting of the cosine router MoE.

Recall that under the exact-specified setting, the true number of experts $k_*$ is known. According to the proof technique for deriving parameter estimation rates under the exact-specified setting in the literature (Nguyen et al., 2023), a key step is to apply the first-order Taylor expansions to the product of the softmax's numerator and the expert function, i.e. $H(x, \beta_1, \eta) := \exp\left(\frac{\beta_1^\top x}{\|\beta_1\| \cdot \|x\|}\right) h(x, \eta)$.
However, since the parameter interaction via the PDE in equation (4), i.e. $\beta_1^\top \frac{\partial H}{\partial \beta_1}(x, \beta_1, \eta) = 0$, holds even for the first derivatives of the function $H$, the convergence of LSE under the exact-specified setting is illustrated by minimax lower bound for estimating $G_*$ in Theorem 5.

**Theorem 5.** *Under the exact-specified setting, the following minimax lower bound of estimating $G_*$*

$$\inf_{\overline{G}_n \in \mathcal{E}_{k_*}(\Theta)} \sup_{G \in \mathcal{E}_{k_*}(\Theta)} \mathbb{E}_{f_G}[\mathcal{L}_{1,r}(\overline{G}_n, G)] \gtrsim n^{-1/2},$$

*holds true for any $r \geq 1$, where $\mathbb{E}_{f_G}$ indicates the expectation taken w.r.t the product measure with $f_G^n$ and the infimum is over all estimators taking values in $\mathcal{E}_{k_*}(\Theta)$.*

Proof of Theorem 5 is deferred to Appendix B.2. It can be seen that the convergence behavior of parameter and expert estimation under the exact-specified setting is analogous to that under the over-specified setting. That is, the rates for estimating parameters $\beta_{1j}^*$ and $\eta_j^*$ as well as experts $h(\cdot, \eta_j^*)$ are slower than any polynomial rates and thus, could be as slow as $\mathcal{O}_P(1/\log^\tau(n))$, where $\tau > 0$ is some constant.

### A.2 EXACT-SPECIFIED SETTING OF THE PERTURBED COSINE ROUTER MOE

We now consider the exact-specified setting of the perturbed cosine router MoE model (7). To begin with, we introduce a condition called *weak identifiability* on the expert function $h(\cdot, \eta)$ to characterize which experts have faster estimation rates than others under this setting.

**Definition 2** (Weak identifiability). *An expert function $x \mapsto h(x, \eta)$ is said to be weakly identifiable if it is differentiable w.r.t its parameter $\eta$ and the set of functions in $x$*

$$\left\{ \frac{\partial^{|\alpha_1| + |\alpha_2|} \widetilde{H}}{\partial \beta_1^{\alpha_1} \partial \eta^{\alpha_2}}(x, \beta_{1i}, \eta_i) : \alpha_1 \in \mathbb{N}^{d_1}, \alpha_2 \in \mathbb{N}^{d_2}, 0 \leq |\alpha_1| + |\alpha_2| \leq 1 \right\},$$

*is linearly independent for almost every $x$, for any $k \geq 1$ and pair-wise distinct parameters $\eta_1, \ldots, \eta_k$, where we denote $\widetilde{H}(x, \beta_1, \eta) := \exp(\frac{\beta_1^\top x}{(\|\beta_1\| + \tau_1) \cdot (\|x\| + \tau_2)}) h(x, \eta)$.*

Recall from the "Technical challenges" paragraph in Section 1 that a key step to establish the expert estimation rates is to decompose the difference $f_{\widetilde{G}_n}(x) - f_{G_*}(x)$ into a combination of linearly independent terms via Taylor expansions to the function $H(\cdot, \beta_1, \eta)$. Therefore, the purpose of the weak identifiability condition is to avoid all potential parameter interactions as in equation (4), which may lead to undesirable linearly dependent terms.

**Example.** For simplicity, we consider experts formulated as neural networks, i.e. $h(x, (a, b)) = \phi(a^\top x + b)$. It can be validated that if the function $\phi(\cdot)$ is either a popular activation such as ReLU$(\cdot)$ and tanh$(\cdot)$ or a polynomial $\phi(z) = z^p$, for any $p \in \mathbb{N}$, then the expert $h(x, (a, b))$ is weakly identifiable. On the other hand, a constant expert $h(\cdot, \eta) = constant$ fails to satisfy the weak identifiability the condition.

Next, we will use the Voronoi loss function $\mathcal{L}_3(G, G_*)$ defined below to determine the estimation rates for weakly identifiable experts in Theorem 6, whose proof can be found in Appendix C.2:

$$\mathcal{L}_3(G, G_*) := \sum_{j=1}^{k_*} \Big| \sum_{i \in \mathcal{A}_j} \exp(\beta_{0i}) - \exp(\beta_{0j}^*) \Big| + \sum_{j=1}^{k_*} \sum_{i \in \mathcal{A}_j} \exp(\beta_{0i}) \Big[ \|\Delta \beta_{1ij}\| + \|\Delta \eta_{ij}\| \Big]. \quad (12)$$

**Theorem 6.** *Assume that $h(\cdot, \eta)$ is a weakly identifiable expert function, then the following lower bound holds true for any $G \in \mathcal{E}_{k_*}(\Theta)$:*

$$\|g_G - g_{G_*}\|_{L^2(\mu)} \gtrsim \mathcal{L}_3(G, G_*).$$

*Furthermore, this bound and the result in Theorem 3 imply that $\mathcal{L}_3(\widetilde{G}_n, G_*) = \mathcal{O}_P(\sqrt{\log(n)/n})$.*

The bound $\mathcal{L}_3(\widetilde{G}_n, G_*) = \mathcal{O}_P(\sqrt{\log(n)/n})$ all the parameters $\beta_{1j}^*, \eta_j^*$ enjoy the same parametric estimation rates, standing at order $\mathcal{O}_P(\sqrt{\log(n)/n})$. Furthermore, by employing the argument in equation (6), we deduce that the rates for estimating experts $h(\cdot, \eta_j^*)$ are also of order $\mathcal{O}_P(\sqrt{\log(n)/n})$. Those rates are substantially faster than their counterparts when using the vanilla cosine router, which could be as slow as $\mathcal{O}_P(1/\log^\tau(n))$ (see Theorem 5). This comparison highlights the benefits of our proposed perturb cosine router over the vanilla cosine router.

# B    PROOF OF RESULTS FOR COSINE ROUTER MOE

In this appendix, we provide proofs for the theoretical results regarding the cosine router in stated in Section 2, including Theorem 1, Theorem 5, and Theorem 2, in that order.

## B.1    PROOF OF THEOREM 1

First of all, let us introduce the definitions of some necessary concepts for the proof, namely an $\varepsilon$-bracket, a bracketing number, a bracketing entropy, an $\varepsilon$-cover and a covering number. In particular, let $(\mathcal{R}, \|\cdot\|)$ be the space of real-valued functions $f : \mathcal{X} \to \mathbb{R}$. Then, the aforementioned concepts are defined as follows:

**Definition 3** ($\varepsilon$-bracket). *Given two functions $L(\cdot)$ and $U(\cdot)$, the bracket $[L, U]$ is the set of all functions $f \in \mathcal{R}$ such that $L(x) \le f(x) \le U(x)$ for all $x \in \mathcal{X}$, and $\|U - L\| \le \varepsilon$.*

**Definition 4** (Bracketing number). *The bracketing number $N_{[]}(\varepsilon, \mathcal{R}, \|\cdot\|)$ is the minimum number of $\varepsilon$-brackets needed to cover $\mathcal{R}$.*

**Definition 5** (Bracketing entropy). *The bracketing entropy $H_B(\varepsilon, \mathcal{R}, \|\cdot\|)$ is the logarithm of the bracketing number $N_{[]}(\varepsilon, \mathcal{R}, \|\cdot\|)$.*

**Definition 6** ($\varepsilon$-cover). *An $\varepsilon$-cover of the set $\mathcal{R}$ under some norm $\|\cdot\|$ is a set $\{\pi_1, \ldots, \pi_N\}$ such that for any $f \in \mathcal{R}$, there exists some $i \in [N]$ such that $\|f - \pi_i\| \le \varepsilon$.*

**Definition 7** (Covering number). *The $\varepsilon$-covering number $N(\varepsilon, \mathcal{R}, \|\cdot\|)$ is the minimum number of balls $B(\pi; \varepsilon) = \{f \in \mathcal{R} : \|f - \pi\| \le \varepsilon\}$ need to cover $\mathcal{R}$.*

Subsequently, we denote by $\mathcal{R}_k(\Theta)$ the set of regression functions w.r.t mixing measures in $\mathcal{G}_k(\Theta)$, that is, $\mathcal{R}_k(\Theta) := \{f_G(x) : G \in \mathcal{G}_k(\Theta)\}$. Additionally, for each $\delta > 0$, the $L^2$ ball centered around the regression function $f_{G_*}$ and intersected with the set $\mathcal{R}_k(\Theta)$ is defined as

$$\mathcal{R}_k(\Theta, \delta) := \left\{ f \in \mathcal{R}_k(\Theta) : \|f - f_{G_*}\|_{L^2(\mu)} \le \delta \right\}.$$

In order to measure the size of the above set, van de Geer (2000) suggest using the following quantity:

$$\mathcal{J}_B(\delta, \mathcal{R}_k(\Theta, \delta)) := \int_{\delta^2/2^{13}}^{\delta} H_B^{1/2}(t, \mathcal{R}_k(\Theta, t), \|\cdot\|_{L^2(\mu)}) \, dt \vee \delta, \tag{13}$$

where $H_B(t, \mathcal{R}_k(\Theta, t), \|\cdot\|_{L^2(\mu)})$ stands for the bracketing entropy (van de Geer, 2000) of $\mathcal{R}_k(\Theta, u)$ under the $L^2$-norm, and $t \vee \delta := \max\{t, \delta\}$. By using the similar proof argument of Theorem 7.4 and Theorem 9.2 in (van de Geer, 2000) with notations being adapted to this work, we obtain the following lemma:

**Lemma 1.** *Take $\Psi(\delta) \geq \mathcal{J}_B(\delta, \mathcal{R}_k(\Theta, \delta))$ that satisfies $\Psi(\delta)/\delta^2$ is a non-increasing function of $\delta$. Then, for some universal constant $c$ and for some sequence $(\delta_n)$ such that $\sqrt{n}\delta_n^2 \geq c\Psi(\delta_n)$, we achieve that*

$$\mathbb{P}\Big(\|f_{\widehat{G}_n} - f_{G_*}\|_{L^2(\mu)} > \delta\Big) \leq c \exp\left(-\frac{n\delta^2}{c^2}\right),$$

*for all $\delta \geq \delta_n$.*

**General picture.** We first show that when the expert functions are Lipschitz continuous, the following bound holds for any $0 < \varepsilon \leq 1/4$:

$$H_B(\varepsilon, \mathcal{R}_k(\Theta, \varepsilon), \|.\|_{L^2(\mu)}) \lesssim \log(1/\varepsilon). \tag{14}$$

Given this bound, it follows that

$$\mathcal{J}_B(\delta, \mathcal{R}_k(\Theta, \delta)) = \int_{\delta^2/2^{13}}^{\delta} H_B^{1/2}(t, \mathcal{R}_k(\Theta, t), \|\cdot\|_{L^2(\mu)}) \, dt \vee \delta \lesssim \int_{\delta^2/2^{13}}^{\delta} \log(1/t) dt \vee \delta. \tag{15}$$

Let $\Psi(\delta) = \delta \cdot [\log(1/\delta)]^{1/2}$, then $\Psi(\delta)/\delta^2$ is a non-increasing function of $\delta$. Furthermore, equation (15) indicates that $\Psi(\delta) \geq \mathcal{J}_B(\delta, \mathcal{R}_k(\Theta, \delta))$. In addition, let $\delta_n = \sqrt{\log(n)/n}$, then we get that $\sqrt{n}\delta_n^2 \geq c\Psi(\delta_n)$ for some universal constant $c$. Finally, by applying Lemma 1, we achieve the desired conclusion of the theorem. As a consequence, it suffices to demonstrate the bound (14).

**Proof for the bound** (14). In order to prove the bracketing entropy bound in equation (14), we leverage the proof arguments for the convergence of regression estimation in (Nguyen et al., 2024c).

Since the expert functions are Lipschitz continuous, then for any function $f_G \in \mathcal{R}_k(\Theta)$, we have that $f_G(x) \leq M$ for all $x$ where $M > 0$ is some constant.

Let $\tau \leq \varepsilon$ and $\{\pi_1, \ldots, \pi_N\}$ be the $\tau$-cover under the $L^\infty$ norm of the set $\mathcal{R}_k(\Theta)$ where $N := N(\tau, \mathcal{R}_k(\Theta), \|\cdot\|_{L^\infty})$ is the $\tau$-covering number of the metric space $(\mathcal{R}_k(\Theta), \|\cdot\|_{L^\infty})$. Then, we construct the brackets of the form $[L_i(x), U_i(x)]$ for all $i \in [N]$ as follows:

$$L_i(x) := \max\{\pi_i(x) - \tau, 0\},$$
$$U_i(x) := \max\{\pi_i(x) + \tau, M\}.$$

From the above formulation, it can be checked that $\mathcal{R}_k(\Theta) \subset \cup_{i=1}^N [L_i(x), U_i(x)]$, and $U_i(x) - L_i(x) \leq \min\{2\tau, M\}$. Additionally, we get that

$$\|U_i - L_i\|_{L^2(\mu)} = \left(\int [U_i(x) - L_i(x)]^2\right)^{1/2} d\mu(x) \leq 2\tau.$$

By definition of the bracketing entropy, we achieve that

$$H_B(2\tau, \mathcal{R}_k(\Theta), \|\cdot\|_{L^2(\mu)}) = \log N_{[]}(2\tau, \mathcal{R}_k(\Theta), \|\cdot\|_{L^2(\mu)})$$
$$\leq \log N = \log N(\tau, \mathcal{R}_k(\Theta), \|\cdot\|_{L^\infty}). \tag{16}$$

Therefore, it is necessary to provide an upper bound for the covering number $N$. Indeed, let us denote $\Delta := \{(\beta_0, \beta_1) \in \mathbb{R} \times \mathbb{R}^{d_1} : (\beta_0, \beta_1, \eta) \in \Theta\}$ and $\Omega := \{\eta \in \mathbb{R}^{d_2} : (\beta_0, \beta_1, \eta) \in \Theta\}$. Since $\Theta$ is a compact set, $\Delta$ and $\Omega$ are also compact. Therefore, we can find $\tau$-covers $\Delta_\tau$ and $\Omega_\tau$ for $\Delta$ and $\Omega$, respectively. Furthermore, it can be validated that

$$|\Delta_\tau| \leq \mathcal{O}_P(\tau^{-(d_1+1)k}), \quad |\Omega_\tau| \leq \mathcal{O}_P(\tau^{-d_2 k}).$$

For each mixing measure $G = \sum_{i=1}^{k} \exp(\beta_{0i})\delta_{(\beta_{1i}, \eta_i)} \in \mathcal{G}_k(\Theta)$, we consider two other mixing measures $G'$ and $\overline{G}$ defined as

$$G' := \sum_{i=1}^{k} \exp(\beta_{0i})\delta_{(\beta_{1i}, \overline{\eta}_i)}, \qquad \overline{G} := \sum_{i=1}^{k} \exp(\overline{\beta}_{0i})\delta_{(\overline{\beta}_{1i}, \overline{\eta}_i)}.$$

Here, $\overline{\eta}_i \in \Omega_\tau$ such that $\overline{\eta}_i$ is the closest to $\eta_i$ in that set, while $(\overline{\beta}_{0i}, \overline{\beta}_{1i}) \in \Delta_\tau$ is the closest to $(\beta_{0i}, \beta_{1i})$ in that set. Now, we aim to upper bound the term $\|f_G - f_{G'}\|_\infty$. In particular, we have

$$
\begin{aligned}
\|f_G - f_{G'}\|_\infty &= \sup_{x \in \mathcal{X}} \left| \sum_{i=1}^{k} \mathrm{Softmax}\left( \frac{(\beta_{1i})^\top x}{\|\beta_{1i}\| \cdot \|x\|} + \beta_{0i} \right) \cdot [h(x, \eta_i) - h(x, \overline{\eta}_i)] \right| \\
&\leq \sum_{i=1}^{k} \sup_{x \in \mathcal{X}} \mathrm{Softmax}\left( \frac{(\beta_{1i})^\top x}{\|\beta_{1i}\| \cdot \|x\|} + \beta_{0i} \right) \cdot |h(x, \eta_i) - h(x, \overline{\eta}_i)| \\
&\leq \sum_{i=1}^{k} \sup_{x \in \mathcal{X}} |h(x, \eta_i) - h(x, \overline{\eta}_i)| \\
&\lesssim \sum_{i=1}^{k} \sup_{x \in \mathcal{X}} \|\eta_i - \overline{\eta}_i\| \cdot \|x\| \lesssim \tau,
\end{aligned}
$$

Above, the second inequality holds as the softmax weight is bounded by 1, and the third inequality is due to the fact that the expert $h(x, \cdot)$ is a Lipschitz function w.r.t $\eta$ and the input space $\mathcal{X}$ is bounded, i.e., $\|x\| \leq B$ for any $x \in \mathcal{X}$ for some constant $B > 0$.

Next, we demonstrate that $\|f_{G'} - f_{\overline{G}}\|_\infty \lesssim \tau$ as follows:

$$
\begin{aligned}
\|f_{G'} - f_{\overline{G}}\|_\infty &= \sup_{x \in \mathcal{X}} \left| \sum_{i=1}^{k} \left[ \mathrm{Softmax}\left( \frac{\beta_{1i}^\top x}{\|\beta_{1i}\| \cdot \|x\|} + \beta_{0i} \right) - \mathrm{Softmax}\left( \frac{\overline{\beta}_{1i}^\top x}{\|\overline{\beta}_{1i}\| \cdot \|x\|} + \overline{\beta}_{0i} \right) \right] \cdot h(x, \overline{\eta}_i) \right| \\
&\leq \sum_{i=1}^{k} \sup_{x \in \mathcal{X}} \left| \mathrm{Softmax}\left( \frac{\beta_{1i}^\top x}{\|\beta_{1i}\| \cdot \|x\|} + \beta_{0i} \right) - \mathrm{Softmax}\left( \frac{\overline{\beta}_{1i}^\top x}{\|\overline{\beta}_{1i}\| \cdot \|x\|} + \overline{\beta}_{0i} \right) \right| \cdot |h(x, \overline{\eta}_{\ell_i})| \\
&\lesssim \sum_{i=1}^{k} \sup_{x \in \mathcal{X}} [\|\beta_{1i} - \overline{\beta}_{1i}\| \cdot \|x\| + |\beta_{0i} - \overline{\beta}_{0i}|] \\
&\leq \sum_{i=1}^{k} \sup_{x \in \mathcal{X}} [\tau \cdot B + \tau] \lesssim \tau,
\end{aligned}
$$

By the triangle inequality, we have

$$\|f_G - f_{\overline{G}}\|_\infty \leq \|f_G - f_{G'}\|_\infty + \|f_{G'} - f_{\overline{G}}\|_\infty \lesssim \tau.$$

By definition of the covering number, we deduce that

$$
\begin{aligned}
N(\tau, \mathcal{R}_k(\Theta), L^\infty) &\leq |\Delta_\tau| \times |\Omega_\tau| \\
&\leq \mathcal{O}_P(n^{-(d_1+1)k}) \times \mathcal{O}(n^{-d_2 k}) \\
&\leq \mathcal{O}(n^{-(d_1+1+d_2)k}).
\end{aligned}
\tag{17}
$$

Putting the results in equations (16) and (17) together, we achieve that

$$H_B(2\tau, \mathcal{R}_k(\Theta), \| \cdot \|_{L^2(\mu)}) \lesssim \log(1/\tau).$$

By setting $\tau = \varepsilon/2$, we achieve that

$$H_B(\varepsilon, \mathcal{R}_k(\Theta), \|.\|_{L^2(\mu)}) \lesssim \log(1/\varepsilon),$$

which completes the proof.

### B.2 Proof of Theorem 5

**Lemma 2.** *If the following holds for any $r \geq 1$:*

$$\lim_{\varepsilon \to 0} \inf_{G \in \mathcal{E}_{k_*}(\Theta): \mathcal{L}_{1,r}(G,G_*) \leq \varepsilon} \frac{\|f_G - f_{G_*}\|_{L^2(\mu)}}{\mathcal{L}_{1,r}(G,G_*)} = 0, \tag{18}$$

*then we obtain that*

$$\inf_{\overline{G}_n \in \mathcal{E}_{k_*}(\Theta)} \sup_{G \in \mathcal{E}_{k_*}(\Theta)} \mathbb{E}_{f_G}[\mathcal{L}_{1,r}(\overline{G}_n, G)] \gtrsim n^{-1/2}. \tag{19}$$

*Proof of Lemma 2.* Indeed, from the Gaussian assumption on the noise variables $\epsilon_i$, we obtain that $Y_i|X_i \sim \mathcal{N}(f_{G_*}(X_i), \sigma^2)$ for all $i \in [n]$. Next, the assumption in equation (18) indicates for sufficiently small $\varepsilon > 0$ and a fixed constant $C_1 > 0$ which we will choose later, we can find a mixing measure $G'_* \in \mathcal{E}_{k_*}(\Theta)$ such that $\mathcal{L}_{1,r}(G'_*, G_*) = 2\varepsilon$ and $\|f_{G'_*} - f_{G_*}\|_{L^2(\mu)} \leq C_1\varepsilon$. From Le Cam's lemma (Yu, 1997), as the Voronoi loss function $\mathcal{L}_{1,r}$ satisfies the weak triangle inequality, we obtain that

$$\inf_{\overline{G}_n \in \mathcal{E}_{k_*}(\Theta)} \sup_{G \in \mathcal{E}_{k_*}(\Theta)} \mathbb{E}_{f_G}[\mathcal{L}_{1,r}(\overline{G}_n, G)]$$

$$\gtrsim \frac{\mathcal{L}_{1,r}(G'_*, G_*)}{8} \exp(-n\mathbb{E}_{X \sim \mu}[\mathrm{KL}(\mathcal{N}(f_{G'_*}(X), \sigma^2), \mathcal{N}(f_{G_*}(X), \sigma^2))])$$

$$\gtrsim \varepsilon \cdot \exp(-n\|f_{G'_*} - f_{G_*}\|^2_{L^2(\mu)}),$$

$$\gtrsim \varepsilon \cdot \exp(-C_1 n\varepsilon^2), \tag{20}$$

where the second inequality is due to the fact that

$$\mathrm{KL}(\mathcal{N}(f_{G'_*}(X), \sigma^2), \mathcal{N}(f_{G_*}(X), \sigma^2)) = \frac{(f_{G'_*}(X) - f_{G_*}(X))^2}{2\sigma^2}.$$

By choosing $\varepsilon = n^{-1/2}$, we obtain that $\varepsilon \cdot \exp(-C_1 n\varepsilon^2) = n^{-1/2} \exp(-C_1)$. As a consequence, we achieve the desired minimax lower bound in equation (19). $\qquad\square$

**Main proof.** It is sufficient to show that the following limit holds true for any $r \geq 1$:

$$\lim_{\varepsilon \to 0} \inf_{G \in \mathcal{E}_{k_*}(\Theta): \mathcal{L}_{1,r}(G,G_*) \leq \varepsilon} \frac{\|f_G - f_{G_*}\|_{L^2(\mu)}}{\mathcal{L}_{1,r}(G,G_*)} = 0. \tag{21}$$

To this end, we need to construct a sequence of mixing measures $G_n \in \mathcal{E}_{k_*}(\Theta)$ that satisfies $\mathcal{L}_{1,r}(G_n, G_*) \to 0$ and

$$\frac{\|f_{G_n} - f_{G_*}\|_{L^2(\mu)}}{\mathcal{L}_{1,r}(G_n, G_*)} \to 0,$$

as $n \to \infty$. Next, let us take into account the sequence $G_n = \sum_{i=1}^{k_*} \exp(\beta_{0i}^n) \delta_{(\beta_{1i}^n, \eta_i^n)}$ in which

- $\exp(\beta_{0i}^n) = \exp(\beta_{0i}^*)$ for any $1 \leq i \leq k_*$;
- $\beta_{11}^n = \left(1 + \frac{1}{n}\right)\beta_{11}^*$ and $\beta_{1i}^n = \beta_{1i}^*$ for any $2 \leq i \leq k_*$;
- $\eta_i^n = \eta_i^*$ for any $1 \leq i \leq k_*$.

Consequently, it can be verified that when $n \to \infty$, we have

$$\mathcal{L}_{1,r}(G_n, G_*) = \exp(\beta_{01}^*)\left[\|\beta_{11}^n - \beta_{11}^*\|^r\right] = \exp(\beta_{01}^*) \cdot \left(\frac{\sqrt{d}}{n}\right)^r \to 0,$$

Next, we demonstrate that $\|f_{G_n} - f_{G_*}\|_{L^2(\mu)}/\mathcal{L}_{1,r}(G_n, G_*) \to 0$. For that purpose, we consider the quantity

$$Q_n(x) := \left[\sum_{j=1}^{k_*} \exp\left(\frac{(\beta_{1j}^*)^\top x}{\|\beta_{1j}^*\| \cdot \|x\|} + \beta_{0j}^*\right)\right] \cdot [f_{G_n}(x) - f_{G_*}(x)], \tag{22}$$

which can be decomposed as follows:

$$Q_n(x) = \sum_{j=1}^{k_*} \sum_{i \in \mathcal{A}_j} \exp(\beta_{0i}^n) \left[ \exp\left( \frac{(\beta_{1i}^n)^\top x}{\|\beta_{1i}^n\| \cdot \|x\|} \right) h(x, \eta_i^n) - \exp\left( \frac{(\beta_{1j}^*)^\top x}{\|\beta_{1j}^*\| \cdot \|x\|} \right) h(x, \eta_j^*) \right]$$

$$- \sum_{j=1}^{k_*} \sum_{i \in \mathcal{A}_j} \exp(\beta_{0i}^n) \left[ \exp\left( \frac{(\beta_{1i}^n)^\top x}{\|\beta_{1i}^n\| \cdot \|x\|} \right) f_{G_n}(x) - \exp\left( \frac{(\beta_{1j}^*)^\top x}{\|\beta_{1j}^*\| \cdot \|x\|} \right) f_{G_n}(x) \right]$$

$$+ \sum_{j=1}^{k_*} \left( \sum_{i \in \mathcal{A}_j} \exp(\beta_{0i}^n) - \exp(\beta_{0j}^*) \right) \exp\left( \frac{(\beta_{1j}^*)^\top x}{\|\beta_{1j}^*\| \cdot \|x\|} \right) \left[ h(x, \eta_j^*) - f_{G_n}(x) \right]$$

$$:= A_n(x) - B_n(x) + C_n(x).$$

Since $\exp(\beta_{0i}^n) = \exp(\beta_{0i}^*)$ for all $i \in [k_*]$, we deduce that $C_n(x) = 0$. Additionally, from the choices of $\beta_{1i}^n$ and $\eta_i^n$, we can rewrite $A_n(x)$ as

$$A_n(x) = \exp(\beta_{01}^*) \left[ \exp\left( \frac{(\beta_{11}^n)^\top x}{\|\beta_{11}^n\| \cdot \|x\|} \right) - \exp\left( \frac{(\beta_{11}^*)^\top x}{\|\beta_{11}^*\| \cdot \|x\|} \right) \right] h(x, \eta_1^*).$$

Let us denote $F(x, \beta_1) := \exp\left( \frac{\beta_1^\top x}{\|\beta_1\| \cdot \|x\|} \right)$. By applying the Taylor expansion of order $r$, we have

$$A_n(x) = \exp(\beta_{01}^*) h(x, \eta_1^*) \sum_{|\alpha|=1}^{r} \frac{1}{\alpha!} \cdot (\beta_{11}^n - \beta_{11}^*)^\alpha \cdot \frac{\partial^{|\alpha|} F}{\partial \beta_1^\alpha}(x, \beta_{11}^*) + R(x)$$

$$= \exp(\beta_{01}^*) h(x, \eta_1^*) \sum_{|\alpha|=1}^{r} \frac{1}{\alpha!} \left( 1 + \frac{1}{n} \right)^{|\alpha|} (\beta_{11}^*)^\alpha \cdot \frac{\partial^{|\alpha|} F}{\partial \beta_1^\alpha}(x, \beta_{11}^*) + R(x),$$

where $R(x)$ is a Taylor remainder such that $R(x)/\mathcal{L}_{1,r}(G_n, G_*) \to 0$ as $n \to \infty$. It is implied from Lemma 3 (see Appendix D) that

$$\sum_{|\alpha|=t} \frac{1}{\alpha!} (\beta_{11}^*)^\alpha \cdot \frac{\partial^{|\alpha|} F}{\partial \beta_1^\alpha}(x, \beta_{11}^*) = 0,$$

for any $1 \le t \le r$, it follows that $A_n(x) = R(x)$. This result indicates that $A_n(x)/\mathcal{L}_{1,r}(G_n, G_*) \to 0$ as $n \to \infty$. By arguing similarly, we also obtain that $B_n(x)/\mathcal{L}_{1,r}(G_n, G_*) \to 0$ as $n \to \infty$. Combine the previous results together, we achieve that

$$Q_n(x)/\mathcal{L}_{1,r}(G_n, G_*) \to 0.$$

Since the input space $\mathcal{X}$ and the parameter space $\Theta$ are both bounded, the term $\sum_{j=1}^{k_*} \exp\left( \frac{(\beta_{1j}^*)^\top x}{\|\beta_{1j}^*\| \cdot \|x\|} + \beta_{0j}^* \right)$ is also bounded. This result together with the formulation of $Q_n(x)$ in equation (22) suggests that $[f_{G_n}(x) - f_{G_*}(x)]/\mathcal{L}_{1,r}(G_n, G_*) \to 0$ for almost every $x \in \mathcal{X}$. As a consequence, we get that

$$\frac{\|f_{G_n} - f_{G_*}\|_{L^2(\mu)}}{\mathcal{L}_{1,r}(G_n, G_*)} \to 0,$$

as $n \to \infty$, and hence, achieve the result in equation (21).

### B.3 PROOF OF THEOREM 2

Similar to the proof of Theorem 5 in Appendix B.2, we only need to demonstrate that the following limit holds true for any $r \ge 1$:

$$\lim_{\varepsilon \to 0} \inf_{G \in \mathcal{G}_k(\Theta): \mathcal{L}_{1,r}(G, G_*) \le \varepsilon} \frac{\|\bar{f}_G - f_{G_*}\|_{L^2(\mu)}}{\mathcal{L}_{1,r}(G, G_*)} = 0. \tag{23}$$

For that purpose, it suffices to build a sequence of mixing measures $G_n \in \mathcal{G}_k(\Theta)$ that satisfies $\mathcal{L}_{1,r}(G_n, G_*) \to 0$ and

$$\frac{\|\bar{f}_{G_n} - f_{G_*}\|_{L^2(\mu)}}{\mathcal{L}_{1,r}(G_n, G_*)} \to 0,$$

as $n \to \infty$. Let us consider the sequence $G_n = \sum_{i=1}^{k_*+1} \exp(\beta_{0i}^n) \delta_{(\beta_{1i}^n, \eta_i^n)}$ in which

- $\exp(\beta_{01}^n) = \exp(\beta_{02}^n) = \frac{1}{2}\exp(\beta_{01}^*)$, and $\exp(\beta_{0i}^n) = \exp(\beta_{0(i-1)}^*)$ for any $3 \leq i \leq k_*+1$;

- $\beta_{11}^n = \left(1 - \frac{1}{n}\right)\beta_{11}^*$, $\beta_{12}^n = \left(1 + \frac{1}{n}\right)\beta_{11}^*$ and $\beta_{1i}^n = \beta_{1(i-1)}^*$ for any $3 \leq i \leq k_*+1$;

- $\eta_1^n = \eta_2^n = \eta_1^*$, and $\eta_i^n = \eta_{i-1}^*$ for any $3 \leq i \leq k_*+1$.

Consequently, it can be verified that when $n \to \infty$, we have

$$\mathcal{L}_{1,r}(G_n, G_*) = \frac{1}{2}\exp(\beta_{01}^*)\Big[\|\beta_{11}^n - \beta_{11}^*\|^r + \|\beta_{12}^n - \beta_{11}^*\|^r\Big] = \exp(\beta_{01}^*) \cdot \left(\frac{\sqrt{d_1}}{n}\right)^r \to 0,$$

Next, we demonstrate that $\|\bar{f}_{G_n} - f_{G_*}\|_{L^2(\mu)}/\mathcal{L}_{1,r}(G_n, G_*) \to 0$.

To this end, we consider the quantity

$$Q_n(x) := \left[\sum_{j=1}^{k_*} \exp\left(\frac{(\beta_{1j}^*)^\top x}{\|\beta_{1j}^*\| \cdot \|x\|} + \beta_{0j}^*\right)\right] \cdot [f_{G_n}(x) - f_{G_*}(x)], \tag{24}$$

which can be decomposed as follows:

$$Q_n(x) = \sum_{j=1}^{k_*}\sum_{i \in \mathcal{A}_j} \exp(\beta_{0i}^n)\left[\exp\left(\frac{(\beta_{1i}^n)^\top x}{\|\beta_{1i}^n\| \cdot \|x\|}\right) h(x, \eta_i^n) - \exp\left(\frac{(\beta_{1j}^*)^\top x}{\|\beta_{1j}^*\| \cdot \|x\|}\right) h(x, \eta_j^*)\right]$$

$$- \sum_{j=1}^{k_*}\sum_{i \in \mathcal{A}_j} \exp(\beta_{0i}^n)\left[\exp\left(\frac{(\beta_{1i}^n)^\top x}{\|\beta_{1i}^n\| \cdot \|x\|}\right) f_{G_n}(x) - \exp\left(\frac{(\beta_{1j}^*)^\top x}{\|\beta_{1j}^*\| \cdot \|x\|}\right) f_{G_n}(x)\right]$$

$$+ \sum_{j=1}^{k_*}\left(\sum_{i \in \mathcal{A}_j} \exp(\beta_{0i}^n) - \exp(\beta_{0j}^*)\right)\exp\left(\frac{(\beta_{1j}^*)^\top x}{\|\beta_{1j}^*\| \cdot \|x\|}\right)\left[h(x, \eta_j^*) - f_{G_n}(x)\right]$$

$$:= A_n(x) - B_n(x) + C_n(x).$$

From the choices of $\beta_{1i}^n$ and $\eta_i^n$, we can rewrite $A_n(x)$ as

$$A_n(x) = \frac{1}{2}\exp(\beta_{01}^*)h(x, \eta_1^*)\sum_{i=1}^{2}\left[\exp\left(\frac{(\beta_{1i}^n)^\top x}{\|\beta_{1i}^n\| \cdot \|x\|}\right) - \exp\left(\frac{(\beta_{11}^*)^\top x}{\|\beta_{11}^*\| \cdot \|x\|}\right)\right].$$

Let us denote $F(x, \beta_1) := \exp\left(\frac{\beta_1^\top x}{\|\beta_1\| \cdot \|x\|}\right)$. By applying the Taylor expansion of order $r$, we have

$$A_n(x) = \frac{1}{2}\exp(\beta_{01}^*)h(x, \eta_1^*)\sum_{i=1}^{2}\sum_{|\alpha|=1}^{r} \frac{1}{\alpha!} \cdot (\beta_{1i}^n - \beta_{11}^*)^\alpha \cdot \frac{\partial^{|\alpha|}F}{\partial\beta_1^\alpha}(x, \beta_{1i}^*) + R(x)$$

$$= \frac{1}{2}\exp(\beta_{01}^*)h(x, \eta_1^*)\sum_{i=1}^{2}\sum_{|\alpha|=1}^{r} \frac{1}{\alpha!}\left(1 + \frac{(-1)^i}{n}\right)^{|\alpha|}(\beta_{11}^*)^\alpha \cdot \frac{\partial^{|\alpha|}F}{\partial\beta_1^\alpha}(x, \beta_{11}^*) + R(x),$$

where $R(x)$ is a Taylor remainder such that $R(x)/\mathcal{L}_{1,r}(G_n, G_*) \to 0$ as $n \to \infty$. It follows from Lemma 3 (see Appendix D) that

$$\sum_{|\alpha|=t} \frac{1}{\alpha!}\left(1 + \frac{(-1)^i}{n}\right)^{|\alpha|}(\beta_{11}^*)^\alpha \cdot \frac{\partial^{|\alpha|}F}{\partial\beta_1^\alpha}(x, \beta_{11}^*)$$

$$= \left(1 + \frac{(-1)^i}{n}\right)^t \sum_{|\alpha|=t} \frac{1}{\alpha!}(\beta_{11}^*)^\alpha \cdot \frac{\partial^{|\alpha|}F}{\partial\beta_1^\alpha}(x, \beta_{11}^*) = 0.$$

for any $1 \leq t \leq r$. Thus, we get that $A_n(x) = R(x)$, which implies that $A_n(x)/\mathcal{L}_{1,r}(G_n, G_*) \to 0$ as $n \to \infty$. By arguing similarly, we also obtain that $B_n(x)/\mathcal{L}_{1,r}(G_n, G_*) \to 0$ as $n \to \infty$.

Furthermore, we have

$$
\begin{aligned}
C_n(x) &= \Big( \sum_{i=1}^{2} \exp(\beta_{0i}^n) - \exp(\beta_{01}^*) \Big) \exp\left( \frac{(\beta_{11}^*)^\top x}{\|\beta_{11}^*\| \cdot \|x\|} \right) \Big[ h(x, \eta_1^*) - \bar{f}_{G_n}(x) \Big] \\
&\quad + \sum_{j=2}^{k_*} \Big( \exp(\beta_{0(j+1)}^n) - \exp(\beta_{0j}^*) \Big) \exp\left( \frac{(\beta_{1j}^*)^\top x}{\|\beta_{1j}^*\| \cdot \|x\|} \right) \Big[ h(x, \eta_j^*) - \bar{f}_{G_n}(x) \Big] \\
&= 0.
\end{aligned}
$$

Putting the previous results together, we achieve that

$$
Q_n(x)/\mathcal{L}_{1,r}(G_n, G_*) \to 0.
$$

Since the input space $\mathcal{X}$ and the parameter space $\Theta$ are both bounded, the term $\sum_{j=1}^{k_*} \exp\left( \frac{(\beta_{1j}^*)^\top x}{\|\beta_{1j}^*\| \cdot \|x\|} + \beta_{0j}^* \right)$ is also bounded. This result together with the formulation of $Q_n(x)$ in equation (24) suggests that $[f_{G_n}(x) - f_{G_*}(x)]/\mathcal{L}_{1,r}(G_n, G_*) \to 0$ for almost every $x \in \mathcal{X}$. As a consequence, we get that

$$
\frac{\|f_{G_n} - f_{G_*}\|_{L^2(\mu)}}{\mathcal{L}_{1,r}(G_n, G_*)} \to 0,
$$

as $n \to \infty$, and hence, achieve the result in equation (23).

## C  PROOF OF RESULTS FOR PERTURBED COSINE ROUTER MOE

In this appendix, we provide proofs for the theoretical results regarding the perturbed cosine router, namely Theorem 3, Theorem 6, and Theorem 4, in that order.

### C.1  PROOF OF THEOREM 3

Since the proof of Theorem 3 can be done in a similar fashion to that of Theorem 1, it is omitted.

### C.2  PROOF OF THEOREM 6

In this proof, we aim to establish the following inequality:

$$
\inf_{G \in \mathcal{E}_{k_*}(\Theta)} \|g_G - g_{G_*}\|_{L^2(\mu)}/\mathcal{L}_3(G, G_*) > 0. \tag{25}
$$

For that purpose, we divide the proof of the above inequality into local and global parts in the sequel.

**Local part:** In this part, we demonstrate that

$$
\lim_{\varepsilon \to 0} \inf_{G \in \mathcal{G}_k(\Theta): \mathcal{L}_3(G, G_*) \leq \varepsilon} \|g_G - g_{G_*}\|_{L^2(\mu)}/\mathcal{L}_3(G, G_*) > 0. \tag{26}
$$

Assume by contrary that the above inequality does not hold true, then there exists a sequence of mixing measures $G_n = \sum_{i=1}^{k_*} \exp(\beta_{0i}^n) \delta_{(\beta_{1i}^n, \eta_i^n)}$ in $\mathcal{G}_k(\Theta)$ such that $\mathcal{L}_{3n} := \mathcal{L}_3(G_n, G_*) \to 0$ and

$$
\|g_{G_n} - g_{G_*}\|_{L^2(\mu)}/\mathcal{L}_{3n} \to 0, \tag{27}
$$

as $n \to \infty$. Let us denote by $\mathcal{A}_j^n := \mathcal{A}_j(G_n)$ a Voronoi cell of $G_n$ generated by the $j$-th components of $G_*$. Since our arguments are asymptotic, we may assume that those Voronoi cells do not depend on the sample size, i.e. $\mathcal{A}_j = \mathcal{A}_j^n$. Moreover, recall that under the exact-specified setting, each Voronoi cell has only one element. Therefore, we may assume WLOG that $\mathcal{A}_j = \{j\}$, and

$$
\mathcal{L}_{3n} := \sum_{i=1}^{k_*} \Big| \exp(\beta_{0i}^n) - \exp(\beta_{0i}^*) \Big| + \sum_{i=1}^{k_*} \exp(\beta_{0i}^n) \Big[ \|\Delta\beta_{1i}^n\| + \|\Delta\eta_i^n\| \Big],
$$

where we denote $\Delta\beta_{1i}^n := \beta_{1i}^n - \beta_{1i}^*$ and $\Delta\eta_i^n := \eta_i^n - \eta_i^*$.

Since $\mathcal{L}_{3n} \to 0$, we get that $(\beta_{1i}^n, \eta_i^n) \to (\beta_{1i}^*, \eta_i^*)$ and $\exp(\beta_{0i}^n) \to \exp(\beta_{0i}^*)$ as $n \to \infty$ for any $i \in [k_*]$. Now, we divide the proof of the local part into three steps as follows:

**Step 1: Taylor expansion.** In this step, we decompose the term

$$Q_n(x) := \left[ \sum_{j=1}^{k_*} \exp\left( \frac{(\beta_{1j}^*)^\top x}{(\|\beta_{1j}^*\| + \tau_1) \cdot (\|x\| + \tau_2)} + \beta_{0j}^* \right) \right] \cdot [g_{G_n}(x) - g_{G_*}(x)]$$

into a combination of linearly independent elements using Taylor expansion. In particular, let us denote $F(x, \beta_1) := \exp\left( \frac{\beta_1^\top x}{(\|\beta_1\| + \tau_1) \cdot (\|x\| + \tau_2)} \right)$, then we have

$$
\begin{aligned}
Q_n(x) = & \sum_{i=1}^{k_*} \exp(\beta_{0i}^n)\Big[ F(x, \beta_{1i}^n)h(x, \eta_i^n) - F(x, \beta_{1i}^*)h(x, \eta_i^*) \Big] \\
& - \sum_{i=1}^{k_*} \exp(\beta_{0i}^n)\Big[ F(x, \beta_{1i}^n) - F(x, \beta_{1i}^*) \Big] g_{G_n}(x) \\
& + \sum_{i=1}^{k_*} \Big( \exp(\beta_{0i}^n) - \exp(\beta_{0i}^*) \Big)\Big[ F(x, \beta_{1i}^*)h(x, \eta_i^*) - F(x, \beta_{1i}^*)g_{G_n}(x) \Big] \\
:= & A_n(x) - B_n(x) + C_n(x).
\end{aligned}
\tag{28}
$$

By means of the first-order Taylor expansion, we have

$$A_n(x) = \sum_{i=1}^{k_*} \sum_{|\alpha|=1} \frac{\exp(\beta_{0i}^n)}{\alpha!}(\Delta\beta_{1i}^n)^{\alpha_1}(\Delta\eta_i^n)^{\alpha_2} \cdot \frac{\partial^{|\alpha_1|}F}{\partial\beta_1^{\alpha_1}}(x, \beta_{1i}^*)\frac{\partial^{|\alpha_2|}h}{\partial\eta^{\alpha_2}}(x, \eta_i^*) + R_1(x), \tag{29}$$

where $R_1(x)$ is a Taylor remainder such that $R_1(x)/\mathcal{L}_{3n} \to 0$ as $n \to \infty$. Similarly, we also get that

$$B_n(x) = \sum_{i=1}^{k_*} \sum_{|\gamma|=1} \frac{\exp(\beta_{0i}^n)}{\gamma!}(\Delta\beta_{1i}^n)^\gamma \cdot \frac{\partial^{|\gamma|}F}{\partial\beta_1^\gamma}(x, \beta_{1i}^*)g_{G_n}(x) + R_2(x),$$

where $R_2(x)$ is a Taylor remainder such that $R_2(x)/\mathcal{L}_{3n} \to 0$ as $n \to \infty$. As a result, we deduce that

$$
\begin{aligned}
Q_n(x) = & \sum_{i=1}^{k_*} \sum_{|\alpha|=0}^{1} T_{i,\alpha_1,\alpha_2}^n \cdot \frac{\partial^{|\alpha_1|}F}{\partial\beta_1^{\alpha_1}}(x, \beta_{1i}^*)\frac{\partial^{|\alpha_2|}h}{\partial\eta^{\alpha_2}}(x, \eta_i^*) + R_1(x) \\
& - \sum_{i=1}^{k_*} \sum_{|\gamma|=0}^{1} S_{i,\gamma}^n \cdot \frac{\partial^{|\gamma|}F}{\partial\beta_1^\gamma}(x, \beta_{1i}^*)g_{G_n}(x) - R_2(x),
\end{aligned}
\tag{30}
$$

where we define

$$T_{i,\alpha_1,\alpha_2}^n := \frac{\exp(\beta_{0i}^n)}{\alpha!}(\Delta\beta_{1i}^n)^{\alpha_1}(\Delta\eta_i^n)^{\alpha_2},$$

$$S_{i,\gamma}^n := \frac{\exp(\beta_{0i}^n)}{\gamma!}(\Delta\beta_{1ij}^n)^\gamma,$$

for any $(\alpha_1, \alpha_2) \neq (\mathbf{0}_d, 0)$ and $\gamma \neq \mathbf{0}_d$. Otherwise, $T_{i,\mathbf{0}_d,0}^n = S_{i,\mathbf{0}_d}^n := \exp(\beta_{0i}^n) - \exp(\beta_{0i}^*)$.

**Step 2: Non-vanishing coefficients.** In this step, we show that not all the ratios $T_{i,\alpha_1,\alpha_2}^n/\mathcal{L}_{3n}$, and $S_{i,\gamma}^n/\mathcal{L}_{3n}$ converge to zero. Indeed, assume by contrary that all of them converge to zero, i.e.

$$\frac{T_{i,\alpha_1,\alpha_2}^n}{\mathcal{L}_{3n}} \to 0, \quad \frac{S_{i,\gamma}^n}{\mathcal{L}_{3n}} \to 0$$

as $n \to \infty$. Then, it follows that

- $\frac{1}{\mathcal{L}_{3n}} \cdot \sum_{i=1}^{k_*} \left| \exp(\beta_{0i}^n) - \exp(\beta_{0i}^*) \right| = \frac{1}{\mathcal{L}_{3n}} \cdot \sum_{i=1}^{k_*} |S_{i,\mathbf{0}_d}^n| \to 0;$

- $\frac{1}{\mathcal{L}_{3n}} \cdot \sum_{i=1}^{k_*} \exp(\beta_{0i}^n) \|\Delta\beta_{1i}^n\|_1 = \frac{1}{\mathcal{L}_{3n}} \sum_{i=1}^{k_*} \sum_{u=1}^{d_1} |T_{i,e_{d_1},u,\mathbf{0}_d}^n| \to 0;$

- $\frac{1}{\mathcal{L}_{3n}} \cdot \sum_{i=1}^{k_*} \exp(\beta_{0i}^n) \|\Delta\eta_i^n\|_1 = \frac{1}{\mathcal{L}_{3n}} \sum_{i=1}^{k_*} \sum_{v=1}^{d_2} |T_{i,\mathbf{0}_d,e_{d_2},v}^n| \to 0.$

Due to the topological equivalence of norm-1 and norm-2, we deduce that

$$\frac{1}{\mathcal{L}_{3n}} \cdot \sum_{i=1}^{k_*} \exp(\beta_{0i}^n) \|\Delta\beta_{1i}^n\| \to 0, \qquad \frac{1}{\mathcal{L}_{3n}} \cdot \sum_{i=1}^{k_*} \exp(\beta_{0i}^n) \|\Delta\eta_i^n\| \to 0.$$

As a result, we obtain that

$$1 = \frac{\mathcal{L}_{3n}}{\mathcal{L}_{3n}} = \frac{1}{\mathcal{L}_{3n}} \left\{ \sum_{i=1}^{k_*} \left| \exp(\beta_{0i}^n) - \exp(\beta_{0i}^*) \right| + \sum_{i=1}^{k_*} \exp(\beta_{0i}^n) \left[ \|\Delta\beta_{1i}^n\| + \|\Delta\eta_i^n\| \right] \right\} \to 0,$$

which is a contradiction. Thus, at least one among the ratios $T_{i,\alpha_1,\alpha_2}^n/\mathcal{L}_{3n}$, and $S_{i,\gamma}^n/\mathcal{L}_{3n}$ must not go to zero as $n \to \infty$.

**Step 3: Application of Fatou's lemma.** In this step, we demonstrate a result opposed to that in Step 2, i.e. the ratios $T_{i,\alpha_1,\alpha_2}^n/\mathcal{L}_{3n}$, and $S_{i,\gamma}^n/\mathcal{L}_{3n}$ all converge to zero.

In particular, let us denote by $m_n$ the maximum of the absolute values of $T_{i,\alpha_1,\alpha_2}^n/\mathcal{L}_{3n}$, and $S_{i,\gamma}^n/\mathcal{L}_{3n}$. Since at least one among those ratios must not approach zero as $n \to \infty$, we get that $1/m_n \not\to \infty$ as $n \to \infty$.

Recall from the hypothesis in equation (27) that $\|g_{G_n} - g_{G_*}\|_{L^2(\mu)}/\mathcal{L}_{3n} \to 0$ as $n \to \infty$, which indicates that $\|g_{G_n} - g_{G_*}\|_{L^1(\mu)}/\mathcal{L}_{3n} \to 0$ due to the equivalence between $L^1(\mu)$-norm and $L^2(\mu)$-norm. By means of the Fatou's lemma, we have

$$0 = \lim_{n\to\infty} \frac{\|g_{G_n} - g_{G_*}\|_{L^1(\mu)}}{m_n\mathcal{L}_{3n}} \geq \int \liminf_{n\to\infty} \frac{|g_{G_n}(x) - g_{G_*}(x)|}{m_n\mathcal{L}_{3n}} d\mu(x) \geq 0.$$

This result implies that $[g_{G_n}(x) - g_{G_*}(x)]/[m_n\mathcal{L}_{3n}] \to 0$ for almost every $x$.

Let us denote

$$T_{i,\alpha_1,\alpha_2}^n/m_n\mathcal{L}_{3n} \to t_{i,\alpha_1,\alpha_2},$$
$$S_{i,\gamma}^n/m_n\mathcal{L}_{3n} \to s_{i,\gamma}$$

with a note that at least one among the limits $t_{i,\alpha_1,\alpha_2}$, $s_{i,\gamma}$ is non-zero. Then, from the decomposition in equation (30), we deduce that

$$\sum_{i=1}^{k_*} \sum_{|\alpha|=0}^{1} t_{i,\alpha_1,\alpha_2} \cdot \frac{\partial^{|\alpha_1|}F}{\partial\beta_1^{\alpha_1}}(x,\beta_{1i}^*)\frac{\partial^{|\alpha_2|}h}{\partial\eta^{\alpha_2}}(x,\eta_i^*) - \sum_{i=1}^{k_*} \sum_{|\gamma|=0}^{1} s_{i,\gamma} \cdot \frac{\partial^{|\gamma|}F}{\partial\beta_1^{\gamma}}(x,\beta_{1i}^*)g_{G_n}(x) = 0,$$

for almost every $x$. Note that the expert function $h(\cdot,\eta)$ satisfies the condition in Definition 2, then the above equation implies that $t_{i,\alpha_1,\alpha_2} = s_{i,\gamma} = 0$, for any $i \in [k_*]$, $\alpha_1 \in \mathbb{N}^{d_1}$, $\alpha_2 \in \mathbb{N}^{d_2}$ and $\gamma \in \mathbb{N}^{d_1}$ such that $0 \leq |\alpha_1| + |\alpha_2|, |\gamma| \leq 2$. This contradicts the fact that at least one among the limits $t_{i,\alpha_1,\alpha_2}$, $s_{i,\gamma}$ is different from zero.

Hence, we obtain the local inequality in equation (26). Thus, we can find an $\varepsilon' > 0$ such that

$$\inf_{G\in\mathcal{E}_{k_*}(\Theta):\mathcal{L}_3(G,G_*)\leq\varepsilon'} \|g_G - g_{G_*}\|_{L^2(\mu)}/\mathcal{L}_3(G,G_*) > 0.$$

**Global part:** Given the above result, it suffices to demonstrate that

$$\inf_{G\in\mathcal{E}_{k_*}(\Theta):\mathcal{L}_3(G,G_*)>\varepsilon'} \|g_G - g_{G_*}\|_{L^2(\mu)}/\mathcal{L}_3(G,G_*) > 0. \tag{31}$$

Assume by contrary that the inequality (31) does not hold true, then we can find a sequence of mixing measures $G_n' \in \mathcal{E}_{k_*}(\Theta)$ such that $\mathcal{L}_3(G_n',G_*) > \varepsilon'$ and

$$\lim_{n\to\infty} \frac{\|g_{G_n'} - g_{G_*}\|_{L^2(\mu)}}{\mathcal{L}_3(G_n',G_*)} = 0,$$

which indicates that $\|g_{G'_n} - g_{G_*}\|_{L^2(\mu)} \to 0$ as $n \to \infty$. Recall that $\Theta$ is a compact set, therefore, we can replace the sequence $G'_n$ by one of its subsequences that converge to a mixing measure $G' \in \mathcal{E}_{k_*}(\Omega)$. Since $\mathcal{L}_3(G'_n, G_*) > \varepsilon'$, we deduce that $\mathcal{L}_3(G', G_*) > \varepsilon'$.

Next, by invoking the Fatou's lemma, we have that

$$0 = \lim_{n \to \infty} \|g_{G'_n} - g_{G_*}\|^2_{L^2(\mu)} \geq \int \liminf_{n \to \infty} \left| g_{G'_n}(x) - g_{G_*}(x) \right|^2 \, \mathrm{d}\mu(x).$$

Thus, we get that $g_{G'}(x) = g_{G_*}(x)$ for almost every $x$. From Proposition 1, we deduce that $G' \equiv G_*$. Consequently, it follows that $\mathcal{L}_3(G', G_*) = 0$, contradicting the fact that $\mathcal{L}_3(G', G_*) > \varepsilon' > 0$.

Hence, the proof is completed.

## C.3   PROOF OF THEOREM 4

In this proof, it is sufficient to demonstrate the following inequality:

$$\inf_{G \in \mathcal{G}_k(\Theta)} \|g_G - g_{G_*}\|_{L^2(\mu)} / \mathcal{L}_2(G, G_*) > 0. \tag{32}$$

This can be done by deriving its local part and the global part as in Appendix C.2. Since the global part can be argued similarly, our main goal is to prove the local part:

$$\lim_{\varepsilon \to 0} \inf_{G \in \mathcal{G}_k(\Theta) : \mathcal{L}_2(G, G_*) \leq \varepsilon} \|g_G - g_{G_*}\|_{L^2(\mu)} / \mathcal{L}_2(G, G_*) > 0. \tag{33}$$

Assume by contrary that the above inequality does not hold true, then there exists a sequence of mixing measures $G_n = \sum_{i=1}^k \exp(\beta_{0i}^n) \delta_{(\beta_{1i}^n, \eta_i^n)}$ in $\mathcal{G}_k(\Theta)$ such that $\mathcal{L}_{2n} := \mathcal{L}_2(G_n, G_*) \to 0$ and

$$\|g_{G_n} - g_{G_*}\|_{L^2(\mu)} / \mathcal{L}_{2n} \to 0, \tag{34}$$

as $n \to \infty$. Let us denote by $\mathcal{A}_j^n := \mathcal{A}_j(G_n)$ a Voronoi cell of $G_n$ generated by the $j$-th components of $G_*$. Since our arguments are asymptotic, we may assume that those Voronoi cells do not depend on the sample size, i.e., $\mathcal{A}_j = \mathcal{A}_j^n$. Therefore, we may assume WLOG that

$$\mathcal{L}_{2n} := \sum_{j=1}^{k_*} \Big| \sum_{i \in \mathcal{A}_j} \exp(\beta_{0i}^n) - \exp(\beta_{0j}^*) \Big| + \sum_{j \in [k_*] : |\mathcal{A}_j| > 1} \sum_{i \in \mathcal{A}_j} \exp(\beta_{0i}^n) \Big[ \|\Delta \beta_{1ij}^n\|^2 + \|\Delta \eta_{ij}^n\|^2 \Big]$$

$$+ \sum_{j \in [k_*] : |\mathcal{A}_j| = 1} \sum_{i \in \mathcal{A}_j} \exp(\beta_{0i}^n) \Big[ \|\Delta \beta_{1ij}^n\| + \|\Delta \eta_{ij}^n\| \Big],$$

where we denote $\Delta \beta_{1ij}^n := \beta_{1i}^n - \beta_{1j}^*$ and $\Delta \eta_{ij}^n := \eta_i^n - \eta_j^*$.

Now, we divide the proof of the local part into three steps as follows:

**Step 1: Taylor expansion.** In this step, we decompose the term

$$Q_n(x) := \left[ \sum_{j=1}^{k_*} \exp \Big( \frac{(\beta_{1j}^*)^\top x}{(\|\beta_{1j}^*\| + \tau_1) \cdot (\|x\| + \tau_2)} + \beta_{0j}^* \Big) \right] \cdot [g_{G_n}(x) - g_{G_*}(x)]$$

into a combination of linearly independent elements using Taylor expansion. In particular, let us denote $F(x, \beta_1) := \exp \Big( \frac{\beta_1^\top x}{(\|\beta_1\| + \tau_1) \cdot (\|x\| + \tau_2)} \Big)$, then we have

$$Q_n(x) = \sum_{j=1}^{k_*} \sum_{i \in \mathcal{A}_j} \exp(\beta_{0i}^n) \Big[ F(x, \beta_{1i}^n) h(x, \eta_i^n) - F(x, \beta_{1j}^*) h(x, \eta_j^*) \Big]$$

$$- \sum_{j=1}^{k_*} \sum_{i \in \mathcal{A}_j} \exp(\beta_{0i}^n) \Big[ F(x, \beta_{1i}^n) - F(x, \beta_{1j}^*) \Big] g_{G_n}(x)$$

$$+ \sum_{j=1}^{k_*} \Big( \sum_{i \in \mathcal{A}_j} \exp(\beta_{0i}^n) - \exp(\beta_{0j}^*) \Big) \Big[ F(x, \beta_{1j}^*) h(x, \eta_j^*) - F(x, \beta_{1j}^*) g_{G_n}(x) \Big]$$

$$:= A_n(x) - B_n(x) + C_n(x). \tag{35}$$

Next, we continue to separate the term $A_n(x)$ into two parts as

$$
\begin{aligned}
A_n(x) &:= \sum_{j \in [k_*]:|\mathcal{A}_j|=1} \sum_{i \in \mathcal{A}_j} \exp(\beta_{0i}^n) \Big[ F(x, \beta_{1i}^n) h(x, \eta_i^n) - F(x, \beta_{1j}^*) h(x, \eta_j^*) \Big] \\
&\quad + \sum_{j \in [k_*]:|\mathcal{A}_j|>1} \sum_{i \in \mathcal{A}_j} \exp(\beta_{0i}^n) \Big[ F(x, \beta_{1i}^n) h(x, \eta_i^n) - F(x, \beta_{1j}^*) h(x, \eta_j^*) \Big] \\
&:= A_{n,1}(x) + A_{n,2}(x)
\end{aligned}
$$

Similar to equation (29), by applying the first-order and the second-order Taylor expansions to $A_{n,1}(x)$ and $A_{n,2}(x)$, respectively, we have

$$
A_{n,1}(x) = \sum_{j \in [k_*]:|\mathcal{A}_j|=1} \sum_{|\alpha|=1} \frac{\exp(\beta_{0i}^n)}{\alpha!} (\Delta\beta_{1ij}^n)^{\alpha_1} (\Delta\eta_{ij}^n)^{\alpha_2} \cdot \frac{\partial^{|\alpha_1|} F}{\partial \beta_1^{\alpha_1}}(x, \beta_{1j}^*) \frac{\partial^{|\alpha_2|} h}{\partial \eta^{\alpha_2}}(x, \eta_j^*) + R_1(x),
$$

$$
A_{n,2}(x) = \sum_{j \in [k_*]:|\mathcal{A}_j|>1} \sum_{|\alpha|=1}^{2} \frac{\exp(\beta_{0i}^n)}{\alpha!} (\Delta\beta_{1ij}^n)^{\alpha_1} (\Delta\eta_{ij}^n)^{\alpha_2} \cdot \frac{\partial^{|\alpha_1|} F}{\partial \beta_1^{\alpha_1}}(x, \beta_{1j}^*) \frac{\partial^{|\alpha_2|} h}{\partial \eta^{\alpha_2}}(x, \eta_j^*) + R_2(x),
$$

where $R_i(x)$ is a Taylor remainder such that $R_i(x)/\mathcal{L}_{2n} \to 0$ as $n \to \infty$, for $i \in \{1, 2\}$. Analogously, we also get that $B_n(x) = B_{n,1}(x) + B_{n,2}(x)$ where

$$
B_{n,1}(x) = \sum_{j \in [k_*]:|\mathcal{A}_j|=1} \sum_{|\gamma|=1} \frac{\exp(\beta_{0i}^n)}{\gamma!} (\Delta\beta_{1i}^n)^{\gamma} \cdot \frac{\partial^{|\gamma|} F}{\partial \beta_1^{\gamma}}(x, \beta_{1j}^*) g_{G_n}(x) + R_3(x),
$$

$$
B_{n,2}(x) = \sum_{j \in [k_*]:|\mathcal{A}_j|>1} \sum_{|\gamma|=1}^{2} \frac{\exp(\beta_{0i}^n)}{\gamma!} (\Delta\beta_{1i}^n)^{\gamma} \cdot \frac{\partial^{|\gamma|} F}{\partial \beta_1^{\gamma}}(x, \beta_{1j}^*) g_{G_n}(x) + R_4(x),
$$

in which $R_i(x)$ is a Taylor remainder such that $R_i(x)/\mathcal{L}_{2n} \to 0$ as $n \to \infty$, for $i \in \{3, 4\}$.

As a result, we deduce that

$$
\begin{aligned}
Q_n(x) &= \sum_{j=1}^{k_*} \sum_{|\alpha|=0}^{2} T_{j,\alpha_1,\alpha_2}^n \cdot \frac{\partial^{|\alpha_1|} F}{\partial \beta_1^{\alpha_1}}(x, \beta_{1j}^*) \frac{\partial^{|\alpha_2|} h}{\partial \eta^{\alpha_2}}(x, \eta_j^*) + R_1(x) + R_2(x) \\
&\quad - \sum_{j=1}^{k_*} \sum_{|\gamma|=0}^{2} S_{j,\gamma}^n \cdot \frac{\partial^{|\gamma|} F}{\partial \beta_1^{\gamma}}(x, \beta_{1j}^*) g_{G_n}(x) - R_3(x) - R_4(x),
\end{aligned} \tag{36}
$$

where we define

$$
T_{j,\alpha_1,\alpha_2}^n := \sum_{i \in \mathcal{A}_j} \frac{\exp(\beta_{0i}^n)}{\alpha!} (\Delta\beta_{1ij}^n)^{\alpha_1} (\Delta\eta_{ij}^n)^{\alpha_2},
$$

$$
S_{j,\gamma}^n := \sum_{i \in \mathcal{A}_j} \frac{\exp(\beta_{0i}^n)}{\gamma!} (\Delta\beta_{1ij}^n)^{\gamma},
$$

for any $(\alpha_1, \alpha_2) \neq (\mathbf{0}_d, 0)$ and $\gamma \neq \mathbf{0}_d$. Otherwise, $T_{j,\mathbf{0}_d,0}^n = S_{j,\mathbf{0}_d}^n := \sum_{i \in \mathcal{A}_j} \exp(\beta_{0i}^n) - \exp(\beta_{0j}^*)$.

**Step 2: Non-vanishing coefficients.** In this step, we show that not all the ratios $T_{j,\alpha_1,\alpha_2}^n/\mathcal{L}_{2n}$, and $S_{j,\gamma}^n/\mathcal{L}_{2n}$ converge to zero. Indeed, assume by contrary that all of them converge to zero, i.e.

$$
\frac{T_{j,\alpha_1,\alpha_2}^n}{\mathcal{L}_{2n}} \to 0, \quad \frac{S_{j,\gamma}^n}{\mathcal{L}_{2n}} \to 0
$$

as $n \to \infty$. Then, it follows that

- $\frac{1}{\mathcal{L}_{2n}} \cdot \sum_{j=1}^{k_*} \Big| \sum_{i \in \mathcal{A}_j} \exp(\beta_{0i}^n) - \exp(\beta_{0j}^*) \Big| = \frac{1}{\mathcal{L}_{2n}} \cdot \sum_{j=1}^{k_*} |S_{j,\mathbf{0}_d}^n| \to 0;$
- $\frac{1}{\mathcal{L}_{2n}} \cdot \sum_{j=1}^{k_*} \sum_{i \in \mathcal{A}_j} \exp(\beta_{0i}^n) \|\Delta\beta_{1ij}^n\|_1 = \frac{1}{\mathcal{L}_{2n}} \sum_{j=1}^{k_*} \sum_{u=1}^{d_1} |T_{j,e_{d_1,u},\mathbf{0}_d}^n| \to 0;$

- $\frac{1}{\mathcal{L}_{2n}} \cdot \sum_{j=1}^{k_*} \sum_{i \in \mathcal{A}_j} \exp(\beta_{0i}^n) \|\Delta \eta_{ij}^n\|_1 = \frac{1}{\mathcal{L}_{2n}} \sum_{j=1}^{k_*} \sum_{v=1}^{d_2} |T_{j,\mathbf{0}_d,e_{d_2,v}}^n| \to 0;$

- $\frac{1}{\mathcal{L}_{2n}} \cdot \sum_{j=1}^{k_*} \sum_{i \in \mathcal{A}_j} \exp(\beta_{0i}^n) \|\Delta \beta_{1ij}^n\|^2 = \frac{1}{\mathcal{L}_{2n}} \sum_{j=1}^{k_*} \sum_{u=1}^{d_1} |T_{j,2e_{d_1,u},\mathbf{0}_d}^n| \to 0;$

- $\frac{1}{\mathcal{L}_{2n}} \cdot \sum_{j=1}^{k_*} \sum_{i \in \mathcal{A}_j} \exp(\beta_{0i}^n) \|\Delta \eta_{ij}^n\|^2 = \frac{1}{\mathcal{L}_{2n}} \sum_{j=1}^{k_*} \sum_{v=1}^{d_2} |T_{j,\mathbf{0}_d,2e_{d_2,v}}^n| \to 0.$

Due to the topological equivalence of the norm-1 and norm-2, we deduce that

$$\frac{1}{\mathcal{L}_{2n}} \cdot \sum_{j=1}^{k_*} \sum_{i \in \mathcal{A}_j} \exp(\beta_{0i}^n) \|\Delta \beta_{1ij}^n\| \to 0, \qquad \frac{1}{\mathcal{L}_{2n}} \cdot \sum_{j=1}^{k_*} \sum_{i \in \mathcal{A}_j} \exp(\beta_{0i}^n) \|\Delta \eta_{ij}^n\| \to 0.$$

Thus, by taking the summation of the above limits, we obtain that $1 = \mathcal{L}_{2n}/\mathcal{L}_{2n} \to 0$ as $n \to \infty$, which is a contradiction. Consequently, at least one among the ratios $T_{j,\alpha_1,\alpha_2}^n/\mathcal{L}_{2n}$, and $S_{j,\gamma}^n/\mathcal{L}_{2n}$ must not go to zero as $n \to \infty$.

**Step 3: Application of Fatou's lemma.** In this step, we demonstrate a result opposed to that in Step 2, i.e. the ratios $T_{j,\alpha_1,\alpha_2}^n/\mathcal{L}_{2n}$, and $S_{j,\gamma}^n/\mathcal{L}_{2n}$ all converge to zero.

In particular, let us denote by $m_n$ the maximum of the absolute values of $T_{j,\alpha_1,\alpha_2}^n/\mathcal{L}_{2n}$, and $S_{j,\gamma}^n/\mathcal{L}_{2n}$. Since at least one among those ratios must not approach zero as $n \to \infty$, we get that $1/m_n \not\to \infty$ as $n \to \infty$.

Recall from the hypothesis in equation (34) that $\|g_{G_n} - g_{G_*}\|_{L^2(\mu)}/\mathcal{L}_{2n} \to 0$ as $n \to \infty$, which indicates that $\|g_{G_n} - g_{G_*}\|_{L^1(\mu)}/\mathcal{L}_{2n} \to 0$ due to the equivalence between $L^1(\mu)$-norm and $L^2(\mu)$-norm. By means of the Fatou's lemma, we have

$$0 = \lim_{n \to \infty} \frac{\|g_{G_n} - g_{G_*}\|_{L^1(\mu)}}{m_n \mathcal{L}_{2n}} \geq \int \liminf_{n \to \infty} \frac{|g_{G_n}(x) - g_{G_*}(x)|}{m_n \mathcal{L}_{2n}} \mathrm{d}\mu(x) \geq 0.$$

This result implies that $[g_{G_n}(x) - g_{G_*}(x)]/[m_n \mathcal{L}_{2n}] \to 0$ for almost every $x$.

Let us denote

$$T_{j,\alpha_1,\alpha_2}^n/m_n \mathcal{L}_{2n} \to t_{j,\alpha_1,\alpha_2},$$
$$S_{j,\gamma}^n/m_n \mathcal{L}_{2n} \to s_{j,\gamma}$$

with a note that at least one among the limits $t_{j,\alpha_1,\alpha_2}, s_{j,\gamma}$ is non-zero. Then, from the decomposition in equation (36), we deduce that

$$\sum_{j=1}^{k_*} \sum_{|\alpha|=0}^{2} t_{j,\alpha_1,\alpha_2} \cdot \frac{\partial^{|\alpha_1|} F}{\partial \beta_1^{\alpha_1}}(x, \beta_{1j}^*) \frac{\partial^{|\alpha_2|} h}{\partial \eta^{\alpha_2}}(x, \eta_j^*) - \sum_{j=1}^{k_*} \sum_{|\gamma|=0}^{2} s_{j,\gamma} \cdot \frac{\partial^{|\gamma|} F}{\partial \beta_1^{\gamma}}(x, \beta_{1j}^*) g_{G_n}(x) = 0,$$

for almost every $x$. Note that the expert function $h(\cdot, \eta)$ satisfies the condition in Definition 1, then the above equation implies that $t_{j,\alpha_1,\alpha_2} = s_{j,\gamma} = 0$, for any $j \in [k_*]$, $\alpha_1 \in \mathbb{N}^{d_1}$, $\alpha_2 \in \mathbb{N}^{d_2}$ and $\gamma \in \mathbb{N}^{d_1}$ such that $0 \leq |\alpha_1| + |\alpha_2|, |\gamma| \leq 2$. This contradicts the fact that at least one among the limits $t_{j,\alpha_1,\alpha_2}, s_{j,\gamma}$ is different from zero.

Hence, we obtain the local inequality in equation (33) and complete the proof.

# D AUXILIARY RESULTS

In this appendix, we present three additional results to facilitate the proofs in Appendix B and Appendix C.

**Proposition 1** (Identifiability). *If $g_G(x) = g_{G_*}(x)$ holds true for almost every $x$, then it follows that $G \equiv G'$.*

*Proof of Proposition 1.* For almost every $x$, since $g_G(x) = g_{G_*}(x)$, then we have

$$\sum_{i=1}^{k} \mathrm{Softmax}\Big(\frac{(\beta_{1i})^\top x}{(\|\beta_{1i}\| + \tau_1) \cdot (\|x\| + \tau_2)} + \beta_{0i}\Big) \cdot h(x, \eta_i)$$

$$= \sum_{i=1}^{k_*} \mathrm{Softmax}\Big(\frac{(\beta_{1i}^*)^\top x}{(\|\beta_{1i}^*\| + \tau_1) \cdot (\|x\| + \tau_2)} + \beta_{0i}^*\Big) \cdot h(x, \eta_i^*). \qquad (37)$$

Due to the identifiability of the expert function $h(\cdot, \eta)$, the set $\{h(x, \eta_i') : i \in [k']\}$, where $\eta_1', \ldots, \eta_{k'}'$ are pair-wise different vectors for some $k' \in \mathbb{N}$, is linearly independent for almost every $x$.

Additionally, note that if $k \neq k_*$, then there exists some index $i \in [k_*]$ such that $\zeta_i \neq \eta_j^*$ for any $j \in [k_*]$. This result implies that $\text{Softmax}\left(\frac{(\beta_{1i})^\top x}{(\|\beta_{1i}\| + \tau_1) \cdot (\|x\| + \tau_2)} + \beta_{0i}\right) = 0$ for almost every $x$, which is a contradiction. Thus, we must have $k = k_*$. As a result, it follows that

$$\left\{\text{Softmax}\left((\beta_{1i})^\top x + \beta_{0i}\right) : i \in [k]\right\} = \left\{\text{Softmax}\left((\beta_{1i}^*)^\top x + \beta_{0i}^*\right) : i \in [k_*]\right\},$$

for almost every $x$. WLOG, we may assume that

$$\text{Softmax}\left(\frac{(\beta_{1i})^\top x}{(\|\beta_{1i}\| + \tau_1) \cdot (\|x\| + \tau_2)} + \beta_{0i}\right) = \text{Softmax}\left(\frac{(\beta_{1i}^*)^\top x}{(\|\beta_{1i}^*\| + \tau_1) \cdot (\|x\| + \tau_2)} + \beta_{0i}^*\right), \tag{38}$$

for almost every $x$, for any $i \in [k_*]$. As the $\text{Softmax}$ function is invariant to translations, then the equation (38) indicates that

$$\frac{(\beta_{1i})^\top x}{(\|\beta_{1i}\| + \tau_1) \cdot (\|x\| + \tau_2)} = \frac{(\beta_{1i}^*)^\top x}{(\|\beta_{1i}^*\| + \tau_1) \cdot (\|x\| + \tau_2)},$$
$$\beta_{0i} = \beta_{0i}^* + v_0,$$

for some $v_0 \in \mathbb{R}$. The first equation implies that $\beta_{1i} = \beta_{1i}^*$, while the second equation together with the assumption $\beta_{0k} = \beta_{0k}^* = 0$ lead to $\beta_{0i} = \beta_{0i}^*$ for any $i \in [k_*]$.

Let us consider $x \in \mathcal{X}_\ell$, where $\ell \in [q]$ such that $\{\ell_1, \ldots, \ell_K\} = \{1, \ldots, K\}$. Then, the equation (37) can be rewritten as

$$\sum_{i=1}^{k_*} \exp(\beta_{0i}) \exp\left(\frac{(\beta_{1i}^*)^\top x}{(\|\beta_{1i}^*\| + \tau_1) \cdot (\|x\| + \tau_2)}\right) h(x, \zeta_i)$$
$$= \sum_{i=1}^{k_*} \exp(\beta_{0i}^*) \exp\left(\frac{(\beta_{1i}^*)^\top x}{(\|\beta_{1i}^*\| + \tau_1) \cdot (\|x\| + \tau_2)}\right) h(x, \eta_i^*), \tag{39}$$

for almost every $x \in \mathcal{X}_\ell$. Next, we denote $P_1, P_2, \ldots, P_m$ as a partition of the index set $[k_*]$, where $m \leq k$, such that $\exp(\beta_{0i}) = \exp(\beta_{0i'}^*)$ for any $i, i' \in P_j$ and $j \in [k_*]$. On the other hand, when $i$ and $i'$ do not belong to the same set $P_j$, we let $\exp(\beta_{0i}) \neq \exp(\beta_{0i'})$. Thus, we can represent equation (39) as

$$\sum_{j=1}^{m} \sum_{i \in P_j} \exp(\beta_{0i}) \exp\left(\frac{(\beta_{1i}^*)^\top x}{(\|\beta_{1i}^*\| + \tau_1) \cdot (\|x\| + \tau_2)}\right) h(x, \zeta_i)$$
$$= \sum_{j=1}^{m} \sum_{i \in P_j} \exp(\beta_{0i}^*) \exp\left(\frac{(\beta_{1i}^*)^\top x}{(\|\beta_{1i}^*\| + \tau_1) \cdot (\|x\| + \tau_2)}\right) h(x, \eta_i^*),$$

for almost every $x \in \mathcal{X}_\ell$. Recall that we have $\beta_{1i} = \beta_{1i}^*$ and $\beta_{0i} = \beta_{0i}^*$, for any $i \in [k_*]$, then the above result leads to

$$\{\eta_i : i \in P_j\} \equiv \{\eta_i^* : i \in P_j\},$$

for any $j \in [m]$. As a consequence, we obtain that

$$G = \sum_{j=1}^{m} \sum_{i \in P_j} \exp(\beta_{0i}) \delta_{(\beta_{1i}, \eta_i)} = \sum_{j=1}^{m} \sum_{i \in P_j} \exp(\beta_{0i}) \delta_{(\beta_{1i}^*, \eta_i^*)} = G_*.$$

Hence, we reach the conclusion of this proposition. □

**Lemma 3.** *Let* $F(x, \beta_1) := \exp\left(\frac{\beta_1^\top x}{\|\beta_1\| \cdot \|x\|}\right)$. *For any vector* $\beta_1 \in \mathbb{R}^{d_1}$ *and* $t \in \mathbb{N}$, *we have*

$$\sum_{u_1, \ldots, u_t = 1}^{d_1} \beta_1^{(u_1)} \ldots \beta_1^{(u_t)} \cdot \frac{\partial^t F}{\partial \beta_1^{(u_1)} \ldots \partial \beta_1^{(u_t)}}(x, \beta_1) = 0. \tag{40}$$

*Proof of Lemma 3.* We will prove the above result by using the induction method. In particular, we first show that it holds for $t = 1$. By taking the first derivative of $F$ w.r.t $\beta_1$, we have

$$\frac{\partial F}{\partial \beta_1}(x, \beta_1) = \frac{x \cdot \|\beta_1\| \cdot \|x\| - \frac{\beta_1}{\|\beta_1\|} \cdot \|x\| \cdot \beta_1^\top x}{\|\beta_1\|^2 \|x\|^2} \cdot F(x, \beta_1).$$

Then, it follows that

$$\beta_1^\top \frac{\partial F}{\partial \beta_1}(x, \beta_1) = \frac{\beta_1^\top x \cdot \|\beta_1\| \cdot \|x\| - \frac{\beta_1^\top \beta_1}{\|\beta_1\|} \cdot \|x\| \cdot \beta_1^\top x}{\|\beta_1\|^2 \|x\|^2} = 0,$$

or equivalently,

$$\sum_{u_1=1}^{d_1} \beta_1^{(u_1)} \cdot \frac{\partial F}{\partial \beta_1^{(u_1)}}(x, \beta_1) = 0.$$

Subsequently, assume that the equation (40) holds for $t - 1$, i.e.

$$\sum_{u_1,\dots,u_{t-1}=1}^{d_1} \beta_1^{(u_1)} \dots \beta_1^{(u_{t-1})} \cdot \frac{\partial^{t-1} F}{\partial \beta_1^{(u_1)} \dots \partial \beta_1^{(u_{t-1})}}(x, \beta_1) = 0,$$

we will demonstrate that it also holds for $t$. Note that the above left-hand side can be decomposed as

$$\sum_{u_1,\dots,u_{t-1}=1}^{d_1} \beta_1^{(u_1)} \dots \beta_1^{(u_{t-1})} \cdot \frac{\partial^{t-1} F}{\partial \beta_1^{(u_1)} \dots \partial \beta_1^{(u_{t-1})}}(x, \beta_1) = \sum_{u_1,\dots,u_{t-1} \neq u_t} \beta_1^{(u_1)} \dots \beta_1^{(u_{t-1})} \cdot \frac{\partial^{t-1} F}{\partial \beta_1^{(u_1)} \dots \partial \beta_1^{(u_{t-1})}}$$

$$+ \binom{t-1}{1} \sum_{u_2,\dots,u_{t-1} \neq u_t} \beta_1^{(u_2)} \dots \beta_1^{(u_{t-1})} \beta_1^{(u_t)} \cdot \frac{\partial^{t-1} F}{\partial \beta_1^{(u_2)} \dots \partial \beta_1^{(u_{t-1})} \partial \beta_1^{(u_t)}}(x, \beta_1)$$

$$+ \dots$$

$$+ \binom{t-1}{t-2} \sum_{u_{t-1} \neq u_t} \beta_1^{(u_{t-1})} (\beta_1^{(u_t)})^{t-2} \cdot \frac{\partial^{t-1} F}{\partial \beta_1^{(u_{t-1})} \partial (\beta_1^{(u_t)})^{t-2}}(x, \beta_1) + (\beta_1^{(u_t)})^{t-1} \cdot \frac{\partial^{t-1} F}{\partial (\beta_1^{(u_t)})^{t-1}}(x, \beta_1),$$

where $u_t$ is some index in $[d]$. By taking the derivatives of both sides w.r.t $\beta_1^{(u_t)}$, we get

$$0 = \sum_{u_1,\dots,u_{t-1} \neq u_t} \beta_1^{(u_1)} \dots \beta_1^{(u_{t-1})} \cdot \frac{\partial^t F}{\partial \beta_1^{(u_1)} \dots \partial \beta_1^{(u_t)}}$$

$$+ \binom{t-1}{1} \sum_{u_2,\dots,u_{t-1} \neq u_t} \beta_1^{(u_2)} \dots \beta_1^{(u_{t-1})} \cdot \frac{\partial^{t-1} F}{\partial \beta_1^{(u_2)} \dots \partial \beta_1^{(u_{t-1})} \partial \beta_1^{(u_t)}}(x, \beta_1)$$

$$+ \binom{t-1}{1} \sum_{u_2,\dots,u_{t-1} \neq u_t} \beta_1^{(u_2)} \dots \beta_1^{(u_{t-1})} \beta_1^{(u_t)} \cdot \frac{\partial^t F}{\partial \beta_1^{(u_2)} \dots \partial \beta_1^{(u_{t-1})} \partial (\beta_1^{(u_t)})^2}(x, \beta_1)$$

$$+ \dots$$

$$+ \binom{t-1}{t-2}(t-2) \sum_{u_{t-1} \neq u_t} \beta_1^{(u_{t-1})} (\beta_1^{(u_t)})^{t-3} \cdot \frac{\partial^{t-1} F}{\partial \beta_1^{(u_{t-1})} \partial (\beta_1^{(u_t)})^{t-2}}(x, \beta_1)$$

$$+ \binom{t-1}{t-2} \sum_{u_{t-1} \neq u_t} \beta_1^{(u_{t-1})} (\beta_1^{(u_t)})^{t-2} \cdot \frac{\partial^t F}{\partial \beta_1^{(u_{t-1})} \partial (\beta_1^{(u_t)})^{t-1}}(x, \beta_1)$$

$$+ (t-1)(\beta_1^{(u_t)})^{t-2} \cdot \frac{\partial^{t-1} F}{\partial (\beta_1^{(u_t)})^{t-1}}(x, \beta_1) + (\beta_1^{(u_t)})^{t-1} \cdot \frac{\partial^t F}{\partial (\beta_1^{(u_t)})^t}(x, \beta_1)$$

$$= \sum_{u_1,\dots,u_{t-1}=1}^{d_1} \beta_1^{(u_1)} \dots \beta_1^{(u_{t-1})} \cdot \frac{\partial^t F}{\partial \beta_1^{(u_1)} \dots \partial \beta_1^{(u_t)}}(x, \beta_1)$$

$$+ (t-1) \sum_{u_2,\dots,u_{t-1}=1}^{d_1} \beta_1^{(u_2)} \dots \beta_1^{(u_{t-1})} \cdot \frac{\partial^{t-1} F}{\partial \beta_1^{(u_2)} \dots \partial \beta_1^{(u_t)}}(x, \beta_1).$$

Therefore, it follows that

$$\sum_{u_t=1}^{d_1} \beta_1^{(u_t)} \cdot 0 = \sum_{u_1,\ldots,u_t=1}^{d_1} \beta_1^{(u_1)} \ldots \beta_1^{(u_{t-1})} \beta_1^{(u_t)} \cdot \frac{\partial^t F}{\partial \beta_1^{(u_1)} \ldots \partial \beta_1^{(u_t)}}(x, \beta_1)$$

$$+ (t-1) \sum_{u_2,\ldots,u_t=1}^{d_1} \beta_1^{(u_2)} \ldots \beta_1^{(u_{t-1})} \beta_1^{(u_t)} \cdot \frac{\partial^{t-1} F}{\partial \beta_1^{(u_2)} \ldots \partial \beta_1^{(u_t)}}(x, \beta_1).$$

It is worth noting that

$$\sum_{u_2,\ldots,u_t=1}^{d_1} \beta_1^{(u_2)} \ldots \beta_1^{(u_{t-1})} \beta_1^{(u_t)} \cdot \frac{\partial^{t-1} F}{\partial \beta_1^{(u_2)} \ldots \partial \beta_1^{(u_t)}}(x, \beta_1)$$

$$= \sum_{u_1,\ldots,u_{t-1}=1}^{d_1} \beta_1^{(u_1)} \ldots \beta_1^{(u_{t-1})} \cdot \frac{\partial^{t-1} F}{\partial \beta_1^{(u_1)} \ldots \partial \beta_1^{(u_{t-1})}}(x, \beta_1) = 0.$$

Consequently, we deduce that

$$\sum_{u_1,\ldots,u_t=1}^{d_1} \beta_1^{(u_1)} \ldots \beta_1^{(u_{t-1})} \beta_1^{(u_t)} \cdot \frac{\partial^t F}{\partial \beta_1^{(u_1)} \ldots \partial \beta_1^{(u_t)}}(x, \beta_1) = 0.$$

Hence, we reach the conclusion in equation (40). □

# E  EXPERIMENTAL DETAILS

In this appendix, we provide the details for the numerical experiments on synthetic data, and the experiments with real data on language modeling conducted in Section 5.

## E.1  EXPERIMENTAL DETAILS FOR SYNTHETIC DATA

**Model details.** We now provide the details for the model parameters in model

$$f_{G_*}(x) := \sum_{i=1}^{k_*} \text{Softmax}\left(\frac{(\beta_{1i}^*)^\top x}{(\|\beta_{1i}^*\| + \tau) \cdot (\|x\| + \tau)} + \beta_{0i}^*\right) \cdot \phi\left((a_i^*)^\top x + b_i^*\right). \tag{41}$$

The variance of Gaussian noise is specified as $\sigma^2 = 0.01$. For simplicity, the perturbations for both $\|x\|$ and $\|\beta_{1i}^*\|$ are considered identical, denoted by $\tau_1 = \tau_2 = \tau$. The true parameters for the router, $(\beta_{1i}^*, \beta_{0i}^*) \in \mathbb{R}^d \times \mathbb{R}$, are drawn independently from an isotropic Gaussian distribution with zero mean and variance $\sigma_r^2 = 0.01/d$ for $1 \le i \le 6$, and otherwise are set to zero. Similarly, the true parameters of the experts, $(a_i^*, b_i^*) \in \mathbb{R}^d \times \mathbb{R}$, are drawn independently of an isotropic Gaussian distribution with zero mean and variance $\sigma_e^2 = 1/d$ for all experts. These parameters remain unchanged for all experiments.

**Training procedure.** For each sample size $n$, spanning from $10^3$ to $10^5$, we perform 20 experiments. In every experiment, the parameters initialization for the router's and experts' parameters are adjusted to be near the true parameters, minimizing potential instabilities from the optimization process. Subsequently, we execute SGD across 10 epochs, employing a learning rate of $\eta = 0.1$ to fit a model to the synthetic data. All the numerical experiments are conducted on a MacBook Air equipped with an M1 chip CPU.

## E.2  EXPERIMENTAL DETAILS FOR LANGUAGE MODELING TASK

**Datasets.** We use the Enwik8 and Text8 datasets (Mahoney, 2011) for our character-level language modeling task. The Enwik8 dataset comprises 100 million bytes of unprocessed Wikipedia text, while the Text8 dataset contains 100 million processed Wikipedia characters. We further evaluate the word-level language modeling task on the Wikitext-103 dataset (Merity et al., 2016), which is the largest available word-level language modeling benchmark with long-term dependency. It contains

103M training tokens from 28K articles, with an average length of 3.6K tokens per article, which allows us to test the ability of long-term dependency modeling.

**Metrics.** In the main paper, we employ the Bit per character (BPC) (Graves, 2013) metric to assess the performance of character-level language modeling tasks. This metric measures the average number of bits needed to encode each character in the dataset. It is calculated as follows:

$$\text{BPC(X)} = -\frac{1}{T} \sum_{t=1}^{T} \log_2 \hat{P}_t(x_t)$$

where $T$ is the length of the input string $X$, $\hat{P}_t$ is the approximate distribution and $x_t$ is the character in the input string at location $t$.

Essentially, BPC quantifies the average number of bits required to encode each character in the text using the probability distribution predicted by the model. This concept works as follows: characters with high probability get a short bit sequence, while characters with low probability get a longer sequence. Then, the next character is read from the input and encoded using the bit sequence performance determined from the probability distribution. If the language model is good, the target character will have been predicted with high probability, so the bit sequence will be short. This means that a lower BPC indicates better compression, which in turn demonstrates the model's superior ability to predict the next character accurately.

For the word-level language modeling task on the Wikitext-103 dataset, we utilize Perplexity (PPL) (Jelinek et al., 1977) as our evaluation metric. It represents the exponentiated average negative log-likelihood of a sequence and demonstrates how well the model predicts the next word in a sequence. More specifically, if we have a tokenized sequence $X = (x_0, x_1, ..., x_t)$, the perplexity of $X$ is:

$$\text{PPL(X)} = \exp\left\{ -\frac{1}{t} \sum_{i=1}^{t} \log p_\theta(x_i|x_{<i}) \right\}$$

where $p_\theta(x_i|x_{<i})$ is the log-likelihood of the $i^{th}$ token conditioned on the preceding tokens $x_{<i}$ according to our model. A lower perplexity score indicates better generalization performance.

We follow the implementation of (Pham et al., 2024) to use experts consisting of two linear layers: the first with weights of shape input_dim $\times$ hidden_dim_experts and the second with weights of shape hidden_dim_experts $\times$ input_dim, followed by ReLU activations and Dropout layer with drop rate $p = 0.1$. This architecture ensures the output has a shape of input_dim $\times$ input_dim while can flexibly reduce parameters compared to a single linear layer with weights of shape input_dim $\times$ input_dim when the number of hidden dimensions of experts chosen is smaller than dimensions of the input. We consistently apply this architecture with our perturbed and vanilla cosine router across all datasets.

**Training setup and hyperparameters.** We consider two model configurations: the *small* and *medium* setups. The *small* setup has a total of 15 million parameters with 6 SMoE layers (Dryden & Hoefler, 2022), and each layer can learn spatial structure in the input domain and routing experts at a fine-grained level to utilize it. Similarly, the *medium* setup consists of 36 million parameters with 8 SMoE layers (Dryden & Hoefler, 2022). For SMoE layers in both configurations, we employ 16 experts with *top-2* gating in both cosine and perturbed cosine routers. To mitigate the representation collapse issue, we adopt the method proposed by Chi et al. (2022). Specifically, this approach involves parameterizing the experts using lower-dimensional embeddings $e_i \in \mathbb{R}^{d_e}$ and applying a linear projection to map token vectors into this compact space, rather than the original high-dimensional hidden space $d$. Furthermore, we follow Chi et al. (2022) to set $d_e = 8$, which is half the number of experts in our implementation for the language modeling task.

During training, we use Adam optimizer (Kingma & Ba, 2017) with default parameters. We set the number of training steps to 60000 and 80000 for small and medium configurations, respectively. The results are averaged over three runs for fair comparisons.

All language modeling experiments are conducted on NVIDIA A100 GPUs. Training the small configuration of the Text8 and Enwik8 datasets takes 11 hours, whereas Wikitext-103 requires 5 hours. Training Text8 and Enwik8 for medium configurations takes 17 hours, while Wikitext-103 training takes 8 hours.

### E.3 Experimental Details for Domain Generalization

**Expert types.** Similarly, following (Li et al., 2023), the experts utilized in the domain generalization experiments are two-layer feedforward networks, featuring a GELU activation and a Dropout layer with a drop rate of 0.1 positioned between the two linear layers. The dimensions of the two linear layers are input_dim × hidden_dim_experts and hidden_dim_experts × input_dim, respectively, where hidden_dim_experts = input_dim × 4.

**Training procedure.** All DG experiments are run on NVIDIA A100 GPUs with 15,000 iterations. The training time on PACS, VLCS, OfficeHome, and TerraIncognita is 2 hours, while the training time for DomainNet is 7 hours.

## F    Additional Experiments

In this appendix, we carry out additional experiments on the synthetic data to compare the sample efficiency of the perturbed cosine router to that of the cosine router and the linear router (Nguyen et al., 2024c) under the settings where the data are generated from the same regression framework.

### F.1    Cosine Router vs. Perturbed Cosine Router

**Experimental setup.** We generate the data by first sampling $X_i \sim \text{Uniform}([-1, 1]^d)$ for $i = 1, \ldots, n$. Then, we generate $Y_i$ according to the following model:

$$Y_i = s_{G_*}(X_i) + \epsilon_i,$$

where the regression function $s_{G_*}(\cdot)$ takes the form of a linear router MoE (Nguyen et al., 2024c):

$$\sum_{i=1}^{k_*} \text{Softmax}((\beta_{1i}^*)^\top x + \beta_{0i}^*) \cdot \text{ReLU}\left((a_i^*)^\top x + b_i^*\right).$$

All other experimental details remain the same as specified in Section 5.1 and Appendix E.1.

**Results.** We calculate the Voronoi losses and report the mean values for each sample size in Figure 2a. Error bars representing two standard deviations are also shown. In Figure 2a, the Voronoi losses associated with the cosine router vanish at the rate of $O(n^{-0.10})$, which is very slow and roughly matches our theoretical results. Meanwhile, those associated with the perturbed cosine router converge to zero at significantly faster rate of $O(n^{-0.45})$. This empirically shows that the perturbed cosine router is more sample efficient than the vanilla cosine router.

### F.2    Linear Router vs. Perturbed Cosine Router

**Experimental setup.** From Table 1, we see that the perturbed cosine router outperforms the linear router when the experts are of polynomial forms $(a^\top x + b)^2$. Thus, we will incorporate polynomial experts in both fitted models. Additionally, we generate the data by first sampling $X_i \sim \text{Uniform}([-1, 1]^d)$ for $i = 1, \ldots, n$. Then, we generate $Y_i$ according to the following model:

$$Y_i = s_{G_*}(X_i) + \epsilon_i,$$

where the regression function $s_{G_*}(\cdot)$ takes the form of a linear router MoE with polynomial experts (Nguyen et al., 2024c):

$$\sum_{i=1}^{k_*} \text{Softmax}((\beta_{1i}^*)^\top x + \beta_{0i}^*) \cdot \left((a_i^*)^\top x + b_i^*\right)^2.$$

All other experimental details remain the same as specified in Section 5.1 and Appendix E.1.

**Results.** We calculate the Voronoi losses and report the mean values for each sample size in Figure 2b. Error bars representing two standard deviations are also shown. In Figure 2b, the Voronoi losses associated with the linear router vanish at a very slow rate of $O(n^{-0.13})$. By contrast, the convergence rate of those associated with the perturbed cosine router are of order $O(n^{-0.47})$, which are substantially faster. Thus, we can claim that the perturbed cosine router is more sample efficient than the linear router both theoretically and empirically.

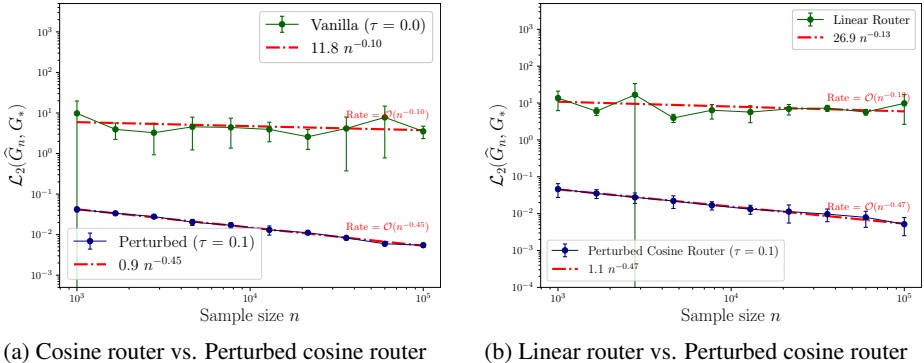

(a) Cosine router vs. Perturbed cosine router     (b) Linear router vs. Perturbed cosine router

Figure 2: Log-log scaled plots displaying the empirical convergence rates. Figure 2a depicts the empirical averages of the Voronoi losses when using the cosine router (green line) versus when using the perturbed cosine router (blue line). The red dash-dotted lines illustrate the fitted lines for determining the empirical convergence rates. Similarly, Figure 2b depicts the empirical averages of the Voronoi losses when using the linear router (green line) versus when using the perturbed cosine router (blue line). We use the same data samples for those experiments.

