# OpenReview forum: "Statistical Advantages of Perturbing Cosine Router in Mixture of Experts"
_ICLR.cc/2025/Conference — ICLR 2025 Poster_

### Official Review · Reviewer_VrcC · 2024-11-01

**Soundness:** 3
**Presentation:** 3
**Contribution:** 3
**Rating:** 6
**Confidence:** 2

**Summary:**

The paper analyzes the generalization bounds of the vanilla cosine router used in Mixture of Experts (MoE) architectures and proposes the addition of noise to enhance these bounds when the experts are Feed-Forward Networks (FFNs) or polynomial functions. Experimental results validate the effectiveness of the proposed method and the theoretical findings.

**Strengths:**

The paper is clearly and logically presented, making it easy to follow the methodology and findings. The experimental results validate the theoretical analysis and demonstrate the effectiveness of the proposed method, providing solid support for the authors' claims.

**Weaknesses:**

As I am not familiar with the statistical techniques used in this field, I found it challenging to fully assess the novelty of this paper compared to prior work. Specifically, in my literature review, I noted that [1] appears closely related to this study. Could the authors clarify any key differences or advancements offered by their approach compared to [1]?
Additionally, as noted by [2], the sparse Mixture of Experts (MoE) approach appears to offer improved generalization capabilities.

[1] Nguyen, Huy, et al. "Statistical Advantages of Perturbing Cosine Router in Sparse Mixture of Experts." arXiv preprint arXiv:2405.14131 (2024).

[2] Li, Bo, et al. "Sparse Mixture-of-Experts are Domain Generalizable Learners." The Eleventh International Conference on Learning Representations.

**Questions:**

1. The intuition behind Equation (4) is unclear. Could the authors elaborate on how the partial differential equation (PDE) influences generalization performance? Additionally, the benefit of the condition $\beta_1\top \frac{\partial \tilde{H}}{\partial \beta_1}(x, \beta_1m \eta)\neq 0$ is not clearly explained.
2. The types of experts used in the experiments should be specified more clearly in the Experimental section. For example, are the experts linear models, ReLU-based feedforward networks (FFNs), or other types?
3. Given that multi-layer perceptrons (MLPs) are commonly used in MoE architectures, how does the perturbed cosine router compare to a linear router with the same generalization bound? What specific advantages does it offer?

---

> ### Author Response · Authors · 2024-11-17
> **Response to Reviewer VrcC (Part 1)**
>
> Dear Reviewer VrcC,
>
> We would like to thank you for your valuable review and for giving **good grades (3)** to the **contribution, soundness and presentation** of our paper. We hope that we can address your concerns with the responses below, and eventually convince you to raise your final rating.
>
> ### **Q1: I noted that [1] appears closely related to this study. Could the authors clarify any key differences or advancements offered by their approach compared to [1]? As noted by [2], the sparse Mixture of Experts (MoE) approach appears to offer improved generalization capabilities.**
>
> Thanks for your question. In this response, we will first highlight the key difference between MoE and sparse MoE. Then, we show that our proof techniques can be directly generalized to the sparse MoE setting.
>
> **1. The key difference between MoE and sparse MoE:** The sparse MoE was first introduced in [3]. Under the sparse MoE setting, each input will only be routed to a few experts which have the highest routing score (the inside part of the softmax function in the mixture weights). This routing mechanism was demonstrated to help scale up the model capacity (the number of model parameters) with a nearly constant computation overhead. Therefore, the sparse MoE has been widely used in large-scale models such as large language models [4, 5] and domain generalization [2].
>
> **2. Generalization of the results to the sparse MoE setting:** We would like to emphasize that although the theoretical results presented in our paper are for the dense MoE, they can be directly generalized for the sparse MoE by employing the technique of partitioning the input space introduced in [6]. In particular, let us recall that the regression function under the perturbed cosine router sparse MoE is given by
>
> $$g_{G_*}(x):=\sum_{i=1}^{k_*} \mathrm{softmax}\left(\mathrm{TopK}\left(\frac{(\beta^*_{1i})^{\top}x}{(||\beta^*_{1i}||+\tau_1)\cdot(||x||+\tau_2)}+\beta^*_{0i}\right)\right)\cdot h(x,\eta^*_i),$$
>
> where $\mathrm{TopK}(u_i)=u_i$ if $u_i$ is in the top K elements of $u=(u_i)$, otherwise, $\mathrm{TopK}(u_i)=-\infty$. It can be seen that the main difference between this regression function and the one considered in our paper is the $\mathrm{TopK}$ function. A key technique to deal with this function is to partition the input space $\mathcal{X}$ into $q:=\binom{k_*}{K}$ regions corresponding to $\binom{k_*}{K}$ different subsets of $k_*$ experts. Assume that $\mathcal{X}_1\subset\mathcal{X}$ is the region
>
> where $\frac{(\beta^*_{1i})^{\top}x}{(\|\beta^*_{1i}\|+\tau_1)\cdot(\|x\|+\tau_2)}+\beta^*_{0i}$ for $i=1,2,\ldots,K$ have the highest values, then we get
>
> $$g_{G_*}(x)=\sum_{i=1}^{K} \mathrm{softmax}\left(\frac{(\beta^*_{1i})^{\top}x}{(||\beta^*_{1i}||+\tau_1)\cdot(||x||+\tau_2)}+\beta^*_{0i}\right)\cdot h(x,\eta^*_i),$$
>
> for $x\in\mathcal{X}_1$. Therefore, we can decompose the $L^2$ distance
>
> $$||g_{G}-g_{G_*}||^2_{L^2(\mu)}=\sum_{\ell=1}^{q}\int_{X_\ell}|g_{G}(x)-g_{G_*}(x)|^2d\mu(x).$$
>
> Then, we can apply the proof techniques in our manuscript to achieve the regression estimation rates in Theorem 1 and Theorem 3. Regarding the parameter and expert estimation rates, it is necessary to modify the Voronoi loss functions, e.g. the loss $L_{2}(G,G_*)$, to capture the convergence behavior of parameters corresponding to the $K$ activated experts as follows:
>
> $$L_2(G,G_*):=\max_{set(\ell_1,\ldots,\ell_K)\subset[k_*]}\Bigg[\sum_{j=1}^{K}\Big|\sum_{i\in A_{\ell_j}}\exp(\beta_{0i})-\exp(\beta^*_{0\ell_j})\Big|+\sum_{j: |A_{\ell_j}|=1}\sum_{i\in A_{\ell_j}}\exp(\beta_{0i})\Big[\|\Delta\beta_{1i\ell_j}\|+\|\Delta \eta_{i\ell_j}\|\Big]$$
> $$+\sum_{j:|A_{\ell_j}|>1}\sum_{i\in A_{\ell_j}}\exp(\beta_{0i})\Big[\|\Delta\beta_{1i\ell_j}\|^2+\|\Delta \eta_{i\ell_j}\|^2\Big]\Bigg].$$
>
> In summary, we achieve the following results:
>
> *(2.1) Regression estimation rate:* we get that $||g_{G_n}-g_{G_*}||_{L^2(\mu)}=\mathcal{O}_P(\sqrt{\log(n)/n})$.
>
> *(2.2) Parameter and Expert estimation rates:* We are also able to demonstrate that
> $L_{2}(G_n,G_*)=\mathcal{O}_P(\sqrt{\log(n)/n})$.
>
>
> After applying this technique, we observe that the convergence behaviors of parameter estimation and expert estimation under the sparse MoE remain unchanged compared to those under the dense MoE. However, the presentation of some objects, namely the Voronoi loss functions, would become unnecessarily complicated for adapting to the sparse MoE setting. For those reasons, we display in our paper only the theoretical results for the dense MoE.
>
> **3. Sparse MoE in real experiments:** Regarding the empirical results, we have leveraged the sparse MoE approach for the language modeling and domain generalization tasks in Section 5.2 and Section 5.3, respectively.

---

> > ### Author Response · Authors · 2024-11-17
> > **Response to Reviewer VrcC (Part 2)**
> >
> > ### **Q2: The intuition behind Equation (4) is unclear. Could the authors elaborate on how the partial differential equation (PDE) influences generalization performance? Additionally, the benefit of the condition $\beta_1^{\top}\frac{\partial\widetilde{H}}{\partial\beta_1}(x,\beta_1,\eta)\neq 0$ is not clearly explained.**
> >
> > Thanks for your suggestion. We will explain the PDE in equation (4) both intuitively and technically, and then explain the benefits of the result $\beta_1^{\top}\frac{\partial\widetilde{H}}{\partial\beta_1}(x,\beta_1,\eta)$.
> >
> > **1. Explanation for the equation (4) under the cosine router MoE:**
> > $$\beta_1^{\top}\frac{\partial H}{\partial\beta_1}(x,\beta_1,\eta)=0,$$
> > where $H(x,\beta_1,\eta):=\exp\Big(\frac{\beta_1^{\top}x}{||\beta_1||\cdot||x||}\Big)h(x,\eta)$.
> >
> > *Intuitively*, the above partial differential equation (PDE) reveals that there is an intrinsic interaction between the parameter $\beta_1$ and itself. Although parameter interactions expressed in the language of PDEs have been observed in [7], the structure of the above interaction are much more sophisticated (even hold for the first-order derivatives while those in [7] occurs only when taking the second-order derivatives). Consequently, the rates for estimating parameters become significantly slow, and could be of order $1/\log^{\tau}(n)$ for some $\tau>0$, whereas those in [7] are also slow but still of polynomial orders.
> >
> > *Technically*, a key step in our proof techniques relies on the decomposition of the regression
> > function discrepancy $f_{G_n}(x)-f_{G_*}(x)$ into a combination of linearly independent terms. This can be done by applying Taylor expansions to the function $H$, which is the product of softmax’s numerator and the expert function $h$. Due to linearly dependent terms resulting from the above PDE, we have to aggregate their coefficients in order to form the desired combination of linearly independent terms. Thus, the rates for estimating parameters are negatively affected.
> >
> > **2. The benefits of the perturbed cosine router MoE under which we have:**
> > $$\beta_1^{\top}\frac{\partial\widetilde{H}}{\partial\beta_1}(x,\beta_1,\eta)\neq 0,$$
> > where $\widetilde{H}(x,\beta_1,\eta):=\exp\Big(\frac{\beta_1^{\top}x}{(||\beta_1||+\tau_1)\cdot(||x||+\tau_2)}\Big)h(x,\eta)$.
> >
> > *Intuitively,* due to the above result, the interaction between the parameter $\beta_1$ and itself mentioned in equation (4) disappears. Therefore, the parameter estimation rates are significantly improved from possibly being $\mathcal{O}_P(1/\log^{\tau}(n))$ to be of polynomial order $\mathcal{O}_P(n^{-1/2})$, which will be technically explained below.
> >
> > *Techincally,* given the regression estimation rate $||g_{G_n}-g_{G_*}||=O_P(n^{-1/2})$ in Theorem 3, a common approach to derive parameter estimation rates is to establish the lower bound $||g_{G_n}-g_{G_*}||\gtrsim L_2(G_n,G_*)$. If that lower bound holds, then we will obtain the polynomial parameter estimation rates via the bound $L_2(G_n,G_*)=O_P(n^{-1/2})$.
> >
> > When $\beta_1^{\top}\frac{\partial\widetilde{H}}{\partial\beta_1}(x,\beta_1,\eta)\neq 0$, we can decompose the difference $g_{G_n}(x)-g_{G_*}(x)$ into a combination of **linearly independent terms** by applying Taylor expansion to the function $\exp\Big(\frac{\beta_1^{\top}x}{(||\beta_1||+\tau_1)(||x||+\tau_2)}\Big)h(x,\eta)$ (see line 1281 and equation (38)). This is a key step to achieve the previous lower bound. Thus, the parameter estimation rates are of polynomial orders.
> >
> > On the other hand, if $\beta_1^{\top}\frac{\partial\widetilde{H}}{\partial\beta_1}(x,\beta_1,\eta)=0$, components in the aforementioned decomposition are linearly dependent as some of them become zero, which is undesirable. Therefore, we cannot achieve our desired lower bound $||g_{G_n}-g_{G_*}||\gtrsim L_2(G_n,G_*)$ as in the case of equation (4).

---

> ### Author Response · Authors · 2024-11-17
> **Response to Reviewer VrcC (Part 3)**
>
> ### **Q3: The types of experts used in the experiments should be specified more clearly in the Experimental section. For example, are the experts linear models, ReLU-based feedforward networks (FFNs), or other types?**
>
> Thanks for your question. We will include the following details on the choices of experts in Appendix E in the revision of our manuscript:
>
> **In the language modeling task**, we follow the implementation of [8] to use experts consisting of two linear layers: the first with weights of shape $input-dim \times hidden-dim-experts$ and the second with weights of shape $hidden-dim-experts \times input-dim$, followed by ReLU activations and Dropout layer with drop rate $p = 0.1$. This architecture ensures the output has a shape of $input-dim \times input-dim$ while can flexibly reduce parameters compared to a single linear layer with weights of shape $input-dim \times input-dim$ when the number of hidden dimensions of experts chosen is smaller than dimensions of the input. We consistently apply this architecture with our perturbed and vanilla cosine router across all datasets.
>
> **In the domain generalization task:**
> Similarly, following [2], the experts utilized in the domain generalization experiments are two-layer feedforward networks, featuring a GELU activation and a Dropout layer with a drop rate of $0.1$ positioned between the two linear layers. The dimensions of the two linear layers are
>
> $input-dim \times hidden-dim-experts$ and $hidden-dim-experts \times input-dim$, respectively,
>
> where $hidden-dim-experts = input-dim \times 4$.
>
> ### **Q4: How does the perturbed cosine router compare to a linear router with the same generalization bound? What specific advantages does it offer?**
>
> Thanks for your questions. We will present the benefits of the perturbed cosine router over the conventional linear router as follows:
>
> **1. The perturbed cosine router addresses the representation collapse issue caused by the linear router:** Chi
> et al. [9] discovered that utilizing the linear router might push hidden representations clustering around expert centroids, implying a trend toward representation collapse. This phenomenon occurs when a fraction of experts govern the decision-making process, leading to the redundancy of other experts and harming the model performance. In response, Chi et al. [9] proposed an alternative known as a cosine router. In particular, the cosine router began with projecting the token hidden representation into a low-dimensional space, followed by applying $L^2$ normalization to both the token representations and expert embeddings. This helped circumvent the dominance of certain experts.
>
> However, we demonstrate in Theorem 2 that using the cosine router might flatten the loss landscape, leading to the slow convergence rates of parameter estimation and expert estimation. Thus, we propose a novel perturbed cosine router
> in which we add noises to the $L^2$ norms of the token representations and the expert embeddings.
> This not only helps stabilize the router training but also improve the convergence rates of parameter estimation and expert estimation significantly as shown in Theorem 4 and Table 1.
>
> **2. The perturbed cosine router is compatible with a broader range of expert functions than the linear router:** We show in Theorem 4 that the parameter and expert estimation rates of order $\mathcal{O}_P(n^{-1/4})$ hold true when we use the perturbed cosine router to aggregate strongly identifiable experts, including experts formulated as neural networks with ReLU activation and polynomial experts, etc. However, Theorem 4.6 in [10] indicates that if we combine polynomial experts using the linear router, then the parameter and expert estimation rates could become as slow as $\mathcal{O}_P(1/\log^{\tau}(n))$, for some constant $\tau>0$.
>
> **3. Experimental results:** We empirically demonstrate in Tables 3, 4 and 5 that the perturbed cosine router MoE outperforms the linear router MoE in most of the tasks in the domain generalization application. Additionally, we also conduct a numerical experiment in Appendix F.2 to show that the convergence rates of parameter estimation and expert estimation when using the perturbed cosine router are faster than those when using the linear router.

---

> ### Author Response · Authors · 2024-11-17
> **Response to Reviewer VrcC (Part 4)**
>
> ### **References**
>
> [1] Nguyen, Huy, et al. "Statistical Advantages of Perturbing Cosine Router in Sparse Mixture of Experts." arXiv preprint arXiv:2405.14131 (2024).
>
> [2] Li, Bo, et al. "Sparse Mixture-of-Experts are Domain Generalizable Learners." The Eleventh International Conference on Learning Representations.
>
> [3] N. Shazeer, A. Mirhoseini, K. Maziarz, A. Davis, Q. Le, G. Hinton, and J. Dean. Outrageously large neural networks: The sparsely-gated mixture-of-experts layer. In ICLR, 2017.
>
> [4] A. Q. Jiang et al. Mixtral of experts. arxiv preprint arxiv 2401.04088, 2024.
>
> [5] W. Fedus, B. Zoph, and N. M. Shazeer. Switch transformers: Scaling to trillion parameter models with simple and efficient sparsity. In JMLR, 2021.
>
> [6] Nguyen, H., Akbarian, P., Yan, F., and Ho, N. Statistical perspective of top-k sparse softmax gating mixture of experts.
> In ICLR, 2024.
>
> [7] H. Nguyen, TT. Nguyen, and N. Ho. Demystifying softmax gating function in Gaussian mixture of experts. Advances in NeurIPS, 2023.
>
> [8] Q. Pham, G. Do, H. Nguyen, T. Nguyen, C. Liu, M. Sartipi, B. T. Nguyen, S. Ramasamy, X. Li, S. Hoi, and N. Ho. Competesmoe - effective training of sparse mixture of experts via competition. arXiv preprint arXiv:2402.02526, 2024.
>
> [9] Chi et al. On the representation collapse of sparse mixture of experts. In NeurIPS, 2022.
>
> [10] H. Nguyen, N. Ho, A. Rinaldo. On least square estimation in softmax gating mixture of experts. In ICML, 2024.

---

> > ### Author Response · Authors · 2024-11-20
> > **Looking forward to your response**
> >
> > Dear Reviewer VrcC,
> >
> > We would like to thank you very much for valuable review, and we hope that our response addresses your previous concerns regarding our paper. However, as the discussion period is expected to end in the next few days, please feel free to let us know if you have any further comments on our work. We would be willing to address any additional concerns from you. Otherwise, we hope that you will consider increasing your scores.
> >
> > Thank you again for spending time on the paper, we really appreciate it!
> >
> > Best regards,
> >
> > The Authors

---

> > > ### Author Response · Authors · 2024-11-25
> > >
> > > Dear Reviewer VrcC,
> > >
> > > We would like to thank you again for spending your precious time on evaluating our manuscript, we really appreciate it. Since the discussion period is expected to end tomorrow, please let us know if you have any additional concerns regarding our work. We are willing to address any concerns from the reviewer. However, if the reviewer is happy with our reponses, we hope that the reviewer will consider adjusting the rating of our work accordingly.
> > >
> > > Thank you,
> > >
> > > The Authors

---

### Official Review · Reviewer_hEMT · 2024-11-02

**Soundness:** 2
**Presentation:** 3
**Contribution:** 3
**Rating:** 6
**Confidence:** 2

**Summary:**

This paper studies the cosine router from a statistical perspective. The main theoretical result is that the cosine router cannot be learned with any polynomial rate, whereas a slight modification (the perturbed cosine router) can. This is then backed up with several experiments showing that the perturbed cosine router outperforms the cosine router.

**Strengths:**

Routing schemes such as the cosine router are very important components in modern machine learning models, but there has been little theoretical study. This paper presents a compelling theoretical picture that a perturbed cosine router should have better rates than the standard cosine router. The experiments are also extensive and seem quite convincing. Finally, the paper is generally quite well-written.

**Weaknesses:**

My main concern is that some of the most important theoretical conclusions of the paper are not formally stated, let alone proven: the claims directly after Theorem 2 about how the theorem implies slow rates for parameter estimation and expert estimation. I (roughly speaking) believe the intuition for (i), though I have some questions -- see below -- but I completely do not understand the intuition for (ii). The claim is that since the parameters \eta are hard to estimate, and h(\eta) is Lipschitz in \eta, then h(\eta) should be hard to estimate as well. But doesn't the inequality go in the wrong direction? e.g. if h(\eta) is constant in \eta, then it's trivial to estimate, and also optimally Lipschitz?

A more minor point -- see questions below -- is that the claim of 1/polylog(n) rate lower bound seems unsubstantiated, since all that is proven is that the rate cannot be any polynomial.

It's possible I am missing something, so I will be happy to increase my score if these concerns are addressed.

**Questions:**

- In e.g. Theorem 2, it seems like an abuse of notation to write \bar{G}_n as in G_k(\Theta) when actually it's an estimator, i.e. a mapping from n samples to G_k(\Theta)? Or am I misunderstanding the statement?
- In Lemma 2, why isn't it enough for the limit to be finite? Why does it have to be 0? It seems like if it's a constant then the application on line 931 would still work, since it seems like C_1 doesn't have to be super small.
- I don't understand the claimed implications (i) and (ii) of Theorem 2. For (i), what if the rate of estimating \beta*_0 is bad? What if only one of {\beta*_1, \eta*} has bad rate and the other has good rate? Finally, just because the rate is slower than any polynomial doesn't mean that it could be 1/polylog(n)?

---

> ### Author Response · Authors · 2024-11-16
> **Response to Reviewer hEMT (Part 1)**
>
> Dear Reviewer hEMT,
>
> We would like to express our gratitude for your constructive feedback and **giving good grades (3)** to the **contribution and presentation** of our paper. We hope that we can address your concerns with the responses below, and eventually convince you to raise your final rating.
>
> ### **Q1: For the implication (ii) of Theorem 2, the claim is that since the parameters $\eta$ are hard to estimate, and $h(\eta)$ is Lipschitz in $\eta$, then $h(\eta)$ should be hard to estimate as well. But doesn't the inequality go in the wrong direction? e.g. if $h(\eta)$ is constant in $\eta$, then it's trivial to estimate, and also optimally Lipschitz?**
>
> Thanks for your questions. We have included the following responses in the revision of our manuscript to address your concerns:
>
> The inequality in equation (6), i.e. $|h(x,\hat{\eta})-h(x,\eta^*)|\leq||\hat{\eta}-\eta^*||$, indicates that the worst possible rate for estimating the expert is identical to the worst possible rate for estimating the parameter $\eta$. Therefore, we only conjecture that the expert estimation rate could be of order $\mathcal{O}_P(1/\log^{\tau}(n))$, for some constant $\tau>0$, with a note that this rate does not necessarily hold true for any expert functions. A counterexample is the case when the expert function $h(x,\eta)$ is constant in $\eta$ as mentioned by the reviewer. However, we would like to emphasize that such expert function is not learnable and thus, it has been rarely studied in previous works.
>
> On the other hand, if the expert function takes the polynomial form of $h(x,\eta):=(\eta x)^2$ (previously considered in [1, 2, 3]), where we assume $x\in\mathbb{R}$ for simplicity, then we have $|h(x,\hat{\eta})-h(x,\eta^*)|=|\hat{\eta}-\eta^*|\cdot|\hat{\eta}+\eta^*|\cdot|x|$. As a result, the expert estimation rate is exactly the parameter estimation rate. In other words, it requires the same number of data points for estimating the parameters to estimate the experts with a given approximation error.
>
> Finally, since the rates for estimating commonly used experts such as the polynomial experts could be extremely slow when using the cosine router, we propose using a novel perturbed cosine router. As shown in Table 1, the worst possible estimation rate for polynomial experts when using the perturbed cosine router is of order $\mathcal{O}_P(n^{-1/4})$, which is significantly faster than that when using cosine router.
>
> [1] E. F. Mendes and W. Jiang. On Convergence Rates of Mixtures of Polynomial Experts. Neural Computation, 2012.
>
> [2] TT. Nguyen, F. Chamroukhi, H. D. Nguyen, F. Forbes. Non-asymptotic model selection in block-diagonal mixture of polynomial experts models. Arxiv preprint.
>
> [3] H. Nguyen, N. Ho, A. Rinaldo. On Least Square Estimation in Softmax Gating Mixture of Experts. In ICML, 2024.
>
> ### **Q2: The claim of $1/polylog(n)$ rate lower bound seems unsubstantiated, since all that is proven is that the rate cannot be any polynomial.**
>
> Thanks for your comment. In Theorem 2, we show that the convergence rates of parameter estimations are slower than polynomial rates $\mathcal{O}_P(n^{-1/2r})$. Then, based on the inequality $\log(n)\leq n$, we conjecture (but not claim) that those rates could be as slow as $\mathcal{O}_P(1/\log^{\tau}(n))$, for some positive constant $\tau$.
>
> ### **Q3: For the implication (i) of Theorem 2, what if the rate of estimating $\beta^{\ast}_0$ is bad? What if only one of $\beta^{\ast}_1$, $\eta^{\ast}$ has bad rate and the other has good rate?**
>
> Thanks for your questions. We address your concerns as follows:
>
> **(i) When the estimation rate of $\eta^{\ast}$ is slower than any polynomial rates:** from the inequality in equation (6), it follows that the worst possible rate of estimating the experts could be slower than any polynomial rates as explained in our response to your Question 1.
>
> **(ii) When the estimation rate of either $\beta^{\ast}_1$ or $\beta^{\ast}_0$ is slower than any polynomial rate:** since the softmax function is 1-Lipschitz with respect to the Euclidean norm, we deduce that the worst possible rate of estimating the cosine router or the mixture weights in equation (2) could also be slower than any polynomial rates. In practice, the router and the expert networks are trained simultaneously (see Section 1.2 in [4]). Thus, the slow convergence of the router might decelerate the model convergence.
>
> Putting the results of these two scenarios, we observe that the slow estimation rates of any parameters might lead to the slow convergence of the model. Therefore, we propose using the perturbed cosine router in mixture of experts, which guarantees that the worst possible parameter estimation rate is $\mathcal{O}_P(n^{-1/4})$, which is significantly faster than that when using the cosine router.
>
> [4] N. Shazeer, A. Mirhoseini, K. Maziarz, A. Davis, Q. Le, G. Hinton, and J. Dean. Outrageously large neural networks: The sparsely-gated mixture-of-experts layer. In ICLR, 2017.

---

> ### Author Response · Authors · 2024-11-16
> **Response to Reviewer hEMT (Part 2)**
>
> ### **Q4: In Theorem 2, it seems like an abuse of notation to write $\bar{G}_n$ as in $\mathcal{G}_k(\Theta)$ when actually it's an estimator, i.e. a mapping from n samples to $\mathcal{G}_k(\Theta)$.**
>
> Thanks for your comment. You are right that $\bar{G}_n$ is an estimator of the true mixing measure and therefore, $\bar{G}_n$ is also a mixing measure. In particular, as mentioned in lines 094-105, since the argmin operator is subject to the set of mixing measures with at most $k$ under the over-specified setting, we write $\bar{G}_n\in\mathcal{G}_k(\Theta)$ in Theorem 2. We will clarify this notation in the revision of our paper.
>
> ### **Q5: In Lemma 2, why isn't it enough for the limit to be finite? Why does it have to be 0? It seems like if it's a constant then the application on line 931 would still work, since it seems like $C_1$ doesn't have to be super small.**
>
> Thanks for your question. Let us explain why the limit in Lemma 2 needs to be zero by contradiction. In particular, if the limit in equation (20) were equal to $C$, where $C>0$, then we could only find a mixing measure $G^1_*$ such that $||f_{G^1_*}-f_{G_*}||\geq C\cdot\mathcal{L}(G^1_*,G_*)$ and $\mathcal{L}(G^1_*,G_*)=2\varepsilon$. Given such information, we would be unable to find a positive constant $C_1$ such that $||f_{G^1_*}-f_{G_*}||\leq C_1\varepsilon$ as in line 931. Hence, we need the limit to be zero. Finally, we would like to note that the proof of this zero limit result has been presented in Appendix B.3.

---

> > ### Author Response · Authors · 2024-11-20
> > **Looking forward to your response**
> >
> > Dear Reviewer hEMT,
> >
> > We would like to thank you very much for your feedback, and we hope that our response addresses your previous concerns regarding our paper. However, as the discussion period is expected to conclude in the next few days, please feel free to let us know if you have any further comments on our work. We would be more than happy to address any additional concerns from you. Otherwise, we hope that you will consider increasing your rating.
> >
> > Thank you again for spending time on the paper, we really appreciate it!
> >
> > Best regards,
> >
> > The Authors

---

> > > ### Comment · Reviewer_hEMT · 2024-11-23
> > >
> > > Thanks for the detailed response. For lemma 2, I still don't understand. If the limit were $C>0$, we would get that for any $\delta>0$, there is some small $\varepsilon>0$ so that for some $G^1_*$, $||f_{G^1_*}-f_{G_*}|| \leq (C+\delta)\cdot\mathcal{L}(G^1_*,G_*)$ and $\mathcal{L}(G^1_*,G_*) \leq \varepsilon$. Doesn't this give what you need?

---

> > > > ### Author Response · Authors · 2024-11-23
> > > >
> > > > Dear Reviewer hEMT,
> > > >
> > > > Thank you for your response. We agree that if the limit were equal to some positive value, then the inequality $||f_{G^1_*}-f_{G_*}||\leq C_1\varepsilon$ might still hold true. However, we demonstrate in equation (22) and in Appendix B.3 that the limit is indeed equal to zero under the exact-specified setting and the over-specified setting, respectively.
> > > >
> > > > Please let us know if you have any additional concerns regarding our manuscript.
> > > >
> > > > Thank you,
> > > >
> > > > The Authors

---

> > > > > ### Author Response · Authors · 2024-11-25
> > > > >
> > > > > Dear Reviewer hEMT,
> > > > >
> > > > > We would like to thank you again for spending your precious time on evaluating our manuscript, we really appreciate it. Since the discussion period is expected to end tomorrow, please let us know if you have any additional concerns regarding our work. We are willing to address any concerns from the reviewer. However, if the reviewer is happy with our reponses, we hope that the reviewer will consider adjusting the rating of our work accordingly.
> > > > >
> > > > > Thank you,
> > > > >
> > > > > The Authors

---

> > > > > > ### Comment · Reviewer_hEMT · 2024-11-25
> > > > > >
> > > > > > Thanks for the clarification, I understand now. I have increased my score due to the revisions in the paper.

---

> > > > > > > ### Author Response · Authors · 2024-11-25
> > > > > > > **Thank you!**
> > > > > > >
> > > > > > > Dear Reviewer hEMT,
> > > > > > >
> > > > > > > We would like to thank you for increasing the rating to 6, we really appreciate it. Please feel free to let us know if you have any further concerns. We will keep revising the manuscript based on the feedback from you and other reviewers.
> > > > > > >
> > > > > > > Thank you,
> > > > > > >
> > > > > > > The Authors

---

### Official Review · Reviewer_idQ2 · 2024-11-04

**Soundness:** 3
**Presentation:** 3
**Contribution:** 2
**Rating:** 6
**Confidence:** 4

**Summary:**

This paper provides a theoretical understanding of the advantage of perturbing cosine router in Mixture-of-Experts (MoE) compared to the vanilla cosine router in terms of sample efficiency. In regression setting where the ground truth itself is generated by an MoE with cosine router and if the expert function satisfies a certain condition ("strong identifiability"), the paper shows that, by adding noise to the L^2 norm of the router weights and the tokens, the intrinsic interactions between the elements of the router parameters can be avoided and thus a better sample complexity can be achieved

**Strengths:**

Originality: The paper provides the first theoretical study of the cosine router MoE and it's perturbed version and confirms the theoretical advantage of the later one, which can be considered as novel.

Quality: The author addresses some technical challenges to complete the theoretical study. For example, the normalization of the cosine router introduces more sophisticated parameter interactions among the elements of the router parameters which needed to be addressed.

Clarity: The paper is well-written and the implication of the results is easily understandable

Significance: The results are significant from the sense that, it provides the theoretical validity of using perturbation in cosine router

**Weaknesses:**

1. The scope of the theoretical result is limited as it required that, the ground truth is itself generated by a cosine routing MoE. It is not clear if practical datasets meet this assumption. Could you please discuss the implications of your results for real-world datasets that may not perfectly match the theoretical assumptions? Any intuition for how their findings might generalize to more realistic settings.

2. In the numerical experiments at section 5.1, the ground truth is generated by an MoE with perturbed cosine router rather than a clean cosine router. Also, the weights are initialized for SGD near the ground truth parameters (which are from the perturbed router). Therefore, for a fair comparison, the ground truth should be generated by the clean router to see whether the perturbed version converge faster (in terms of number of samples required) compared to the clean one. Is it possible to run an additional experiment with the ground truth is generated by a clean cosine router, to more directly test the theoretical claims about improved convergence of the perturbed version?

3. In practical MoE models, all the experts are FFN with ReLU activation with at least one hidden layer (i.e. two-layer FFN). However, in the paper, the exampled FFN version is one-layer. Can the analysis be extended to two-layer case? In other words, does the two-layer FFN with ReLU activation satisfy the strong identifiable condition? Could you please add some discussion on whether and how the analysis might extend to multi-layer FFNs, which are more common in practice?

**Questions:**

The theoretical results are training algorithm agnostic. I'm curious about the analysis under SGD. Here, it has been claimed that adding perturbation (i.e. noise at the router) remove the parameter interaction among the elements of the router parameter. As SGD also add noises, can the SGD based algorithm also or remove the interaction (even for a linear router)? Can you comment on this?


---------------------------
After the response.
Thank you for your response to my questions. However, I decided to decrease my score to 5. My decision is based on the following reasons (related to weakness 1 and weakness 3 of my official review):

1.	The strong identifiability condition is not satisfied by a two-layer ReLU expert function. However, two-layer ReLU is the most common expert function used in practice. Moreover, there exists theoretical works analyzed the two-layer ReLU as expert functions of MoE (e.g., [1], [2])
2.	The authors claimed that the theoretical results can be extended to the more general setting where the ground truth is generated by an arbitrary regression function instead of the perturbed cosine router. A formal discussion has been provided about the generalization at section 4 of the paper. In that case, the formal results of the paper (i.e. Theorem 1-4) should be presented in the more general setting instead the current restrictive setting.

[1] Chen, Z., Deng, Y., Wu, Y., Gu, Q., & Li, Y. (2022). Towards understanding mixture of experts in deep learning. arXiv preprint arXiv:2208.02813.

[2] Chowdhury, M. N. R., Zhang, S., Wang, M., Liu, S., & Chen, P. Y. (2023, July). Patch-level routing in mixture-of-experts is provably sample-efficient for convolutional neural networks. In International Conference on Machine Learning (pp. 6074-6114). PMLR.

After the author response again, I decided to change the score back to 6. The reason is in the comment.

---

> ### Author Response · Authors · 2024-11-17
> **Response to Reviewer idQ2 (Part 1)**
>
> Dear Reviewer idQ2,
>
> We would like to thank you for your insightful review and for giving **good grades (3)** to the **soundness and presentation** of our paper. We hope that we can address your concerns with the responses below, and eventually convince you to raise your final rating.
>
> ### **Q1: The ground truth is itself generated by a cosine routing MoE. It is not clear if practical datasets meet this assumption. Could you please discuss the implications of your results for real-world datasets that may not perfectly match the theoretical assumptions? Any intuition for how their findings might generalize to more realistic settings.**
>
> Thanks for your questions. Actually, we have discussed the generalization of the theoretical results as well as their practical implications in Section 4 of our paper, which will be summarized as follows:
>
> **1. Generalization of the theoretical results to the real-world datasets.** In this paper, we have only considered well-specified settings, namely, the data are assumed to be sampled from the (perturbed) cosine router MoE. Although it may look restrictive, the results under this setting lay an important foundation for a more realistic misspecified setting where the data are not necessarily generated from those models.
>
> Under that misspecified setting, we assume that the data are generated from a regression framework as in equation (1) but with an arbitrary regression function $q(\cdot)$, which is not a (perturbed) cosine router MoE. Then, we can demonstrate that the least square estimator (LSE) $\widehat{G}_n$ converges to a mixing measure
>
> $\overline{G} \in \arg\min_{G \in G_{k}(\Theta)} ||q-f_{G}||_{L^2(\mu)}$,
>
> where $f_{G}(\cdot)$ is a regression function taking the form of the (perturbed) cosine router MoE. Furthermore, the optimal mixing measure will be in the boundary of the parameter space $G_k(\Theta)$, namely, $\overline{G}$ has $k$ atoms. Thus, as $n$ becomes sufficiently large, $\widehat{G}_{n}$ also has $k$ atoms.
>
> The insights from our theories for the well-specified setting indicate that the Voronoi losses can be used to obtain the estimation rates of individual parameters of the LSE $\widehat{G}_n$ to those of $\overline{G}$ and therefore, achieve the following expert estimation rates under the misspecified settings, which will be empirically validated via numerical experiments in Appendix F:
>
> *(1.1) Cosine router MoE:* the worst expert estimation rate could be as slow as $\mathcal{O}_P(1/\log^{\tau}(n))$ for some $\tau > 0$. It indicates that we still need an exponential number of data (roughly $\exp(1/\epsilon^{\tau})$ where $\epsilon$ is the desired approximation error) to estimate the experts as well as select important experts.
>
> *(1.2) Perturbed cosine router MoE:* the slowest expert estimation rate is of order $\mathcal{O}_P(n^{-1/4})$. Thus, we only need a polynomial number of data (roughly $\epsilon^{-4}$) to estimate the experts.
> This explains why the perturbed cosine router is a solution to the parameter estimation problem, or more generally, the expert estimation problem of the MoE models.
>
> **2. Practical implications of the theoretical results:**
>
> *(2.1) Router design:* From the benefits of the perturbed cosine router for the expert estimation of MoE models, our theories suggest that when using the cosine router to avoid the representation collapse issue, practitioners should add noises to $L^2$ norms of the token hidden representations and the expert embeddings to achieve a favorable performance.
>
> *(2.2) Expert design:* We provide in Definition 1 the strong identifiability condition to characterize which expert structures would help improve the model efficiency. We can verify that several commonly used experts, namely those formulated as neural networks with ReLU activation and polynomial experts, are strongly identifiable. This indicates that our theory is potentially useful for designing experts in practical applications.

---

> > ### Author Response · Authors · 2024-11-17
> > **Response to Reviewer idQ2 (Part 2)**
> >
> > ### **Q2: In the numerical experiments, the ground truth is generated by an MoE with perturbed cosine router rather than a clean cosine router. Also, the weights are initialized for SGD near the ground truth parameters (which are from the perturbed router). For a fair comparison, the ground truth should be generated by the clean router to see whether the perturbed version converge faster (in terms of number of samples required) compared to the clean one.**
> >
> > Thanks for your suggestion. We will address your concerns as follows:
> >
> > **Purpose of Experiments in Section 5.1.**
> > The main goal of the experiments in Section 5.1 is to verify the theoretical results rather than to directly compare the performance of the clean (vanilla) router and the perturbed router under an identical setup. Specifically, the key takeaway from Figure 1 is the estimation rate, which aligns with our theoretical predictions. Please see Section 5.2, Section 5.3, and Appendix F.1 for direct comparison of perturbed and vanilla cosine MoE.
> >
> > **Clarification on Figure 1 Experimental Setup.**
> > In Figure 1, we depict two separate experimental cases, each designed solely to verify theoretical results, not to compare the models on an identical dataset. The green curve represents the estimation rate of a vanilla (clean) router MoE fitted on synthetic data generated from a vanilla cosine MoE model, while the blue curve shows the estimation rate of a perturbed cosine MoE fitted on synthetic data generated from a perturbed cosine MoE model. Importantly, we emphasize that the ground truth parameters $(\beta^*_{1i}, \beta^*_{0i}, a^*_i, b^*_i)$ remained the same in both scenarios. This consistent parameterization ensures that the theoretical results can be verified under both clean and perturbed conditions without altering the fundamental model structure.
> >
> > **Possible Confusion with Equation (12).**
> > We believe that some of the confusion may stem from the general formulation presented in Equation (12), where we describe synthetic data generation using a perturbed cosine router model. In this context, Equation (12) allows for a flexible formulation, with the parameter $\tau$ representing the level of perturbation. For the clean (vanilla) router, setting $\tau = 0$ reduces the formulation to a clean router model, effectively removing any perturbation. We acknowledge that this distinction may not have been fully clear and will revise the final manuscript to make the role of $\tau$ and its relationship to clean versus perturbed routers more explicit.
> >
> > **Additional Experiments in Appendix F.1.**
> > To further assist in understanding the comparative performance, we have included additional experiments in Appendix F.1. These experiments directly compare the estimation rates for perturbed and clean router models on data generated from a linear MoE model, providing further insight into their relative behavior under a unified experimental setup.
> >
> > **Comment on Parameter Initialization.**
> > In our experiments, the parameters $(\beta_{1i}, \beta_{0i}, a_i, b_i)$ were initialized near the ground truth values $(\beta^*_{1i}, \beta^*_{0i}, a^*_i, b^*_i)$. This close initialization ensures that any influence of the optimization algorithm on parameter estimation is minimized, allowing us to focus solely on the parameter estimation rate.

---

> > > ### Author Response · Authors · 2024-11-17
> > > **Response to Reviewer idQ2 (Part 3)**
> > >
> > > ### **Q3: Does the two-layer FFN with ReLU activation satisfy the strong identifiability condition? Could you please add some discussion on whether and how the analysis might extend to multi-layer FFNs, which are more common in practice?**
> > >
> > > Thanks for your questions. We will address your concerns as follows:
> > >
> > > **1. Two-layer feed-forward networks (FFNs) with ReLU activation:** We first consider the two-layer FFN $h_1(x,(W_1,b_1,W_2,b_2)):=\mathrm{ReLU}(xW_1+b_1)W_2+b_2$ where $x,b_1,b_2\in\mathbb{R}^{1\times d}$ and $W_1,W_2\in\mathbb{R}^{d\times d}$. Additionally, let us recall the function $\widetilde{H}$ in the strong identifiability condition $\widetilde{H}(x,\beta_1,W_1,b_1,W_2,b_2):=\exp(\frac{\beta_1^{\top}x}{(||\beta_1||+\tau_1)(||x||+\tau_2)})h_1(x,W_1,b_1,W_2,b_2)$. It can be verified that $\frac{\partial^2\widetilde{H}}{\partial b_2^{\top}\partial b_2}=0$. Thus, the set of functions in $x$ in Definition 1 is not linearly independent for almost every $x$, meaning that the expert function $h_1$ is not strongly identifiable.
> > >
> > > However, if we apply an additional activation function to the two-layer FFN $h_1$, that is, $h_2(x,(W_1,b_1,W_2,b_2)):=\sigma(\mathrm{ReLU}(xW_1+b_1)W_2+b_2)$, where the activation function $\sigma$ can be ReLU, GELU or a polynomial function $\sigma(z)=z^p$ for $p\geq 2$, then we can validate that the expert $h_2$ is strongly identifiable by taking the derivatives of the function $\widetilde{H}$ and using the definition of linear independence.
> > >
> > > **2. Theory extension to multi-layer FFNs:** Based on the above result, we conjecture that the $L$-layer neural network $h_L(x,W,b):=\sigma_L(\sigma_{L-1}(\ldots\sigma_1(xW_1+b_1))W_L+b_L)$, where the activation functions $\sigma_{\ell}$ is among the ReLU, GELU and the polynomial function $\sigma_{\ell}(z)=z^p$ for $p\geq 2$, is strongly identifiable. Providing a rigorous proof for this conjecture would require great technical effort as we need to control the linear independence of several derivative terms when the number of layers $L$ becomes increasingly large. Therefore, we believe that further technical tools need to be developed to extend the current theory to the setting of the multi-layer FFNs. Since this is beyond the scope of our work, we leave it for future development.
> > >
> > > ### **Q4: The theoretical results are training algorithm agnostic. I'm curious about the analysis under SGD. Here, it has been claimed that adding perturbation (i.e. noise at the router) remove the parameter interaction among the elements of the router parameter. As SGD also add noises, can the SGD based algorithm also or remove the interaction (even for a linear router)?**
> > >
> > > Thanks for your insightful question. While the theoretical results are training algorithm agnostic, they also shed light on the optimization landscape of the loss function and the behaviors of first-order optimization algorithms, such as SGD. For instance, the intrinsic interaction among parameters under the cosine router setting implies that the optimization landscape tend to be very flat around the global optima, namely, the determinant of the Hessian matrix around the global optima is singular. It leads to extremely slow convergence rates of parameter estimation. Therefore, even with good initialization around the global optima, first-order optimization algorithms, such as SGD, may converge extremely slow towards to the global optima. On the other hand, by adding noise to the cosine router, which eliminates the interaction among parameters, the optimization landscape becomes much sharper and less flat around the global optima, as evidenced by the (much) faster convergence rates of parameter estimation compared to the cosine router setting. It indicates that first-order optimization algorithms, such as SGD, may converge much faster to the global optima under the perturbed cosine router setting compared to those under the standard cosine router setting. We leave the rigorous investigation of the behaviors of first-order optimization algorithms, such as SGD, under both cosine and perturbed cosine router settings for the future work.
> > >
> > > Finally, we would like to clarify that the noise in SGD is corresponding to the noise from the data, which may not be directly related to the interaction among the parameters. Therefore, we conjecture that even with SGD, we may still need an extra step of adding noise to the cosine router to not only stabilize the training of the SGD but also to guarantee that the SGD converge faster to the optimum.

---

> > > > ### Author Response · Authors · 2024-11-20
> > > > **Looking forward to your response**
> > > >
> > > > Dear Reviewer idQ2,
> > > >
> > > > We would like to thank you very much for insightful review, and we hope that our response addresses your previous concerns regarding our paper. However, as the discussion period is expected to end in the next few days, please feel free to let us know if you have any further comments on our work. We would be willing to address any additional concerns from you. Otherwise, we hope that you will consider increasing your rating.
> > > >
> > > > Thank you again for spending time on the paper, we really appreciate it!
> > > >
> > > > Best regards,
> > > >
> > > > The Authors

---

> > > > > ### Author Response · Authors · 2024-11-23
> > > > >
> > > > > Dear Reviewer idQ2,
> > > > >
> > > > > Thank you for your response. Let us address your two concerns as follows:
> > > > >
> > > > > **1. The strong identifiability condition:** we would like to emphasize that the strong identifiability condition is a sufficient condition for experts to have polynomial estimation rates. Therefore, the fact that the two-layer feed-forward networks are not strongly identifiable does not mean that they admit slow estimation rates. On the other hand, this condition provides some new insights into designing expert structure. For example, in the last layer of the FFN, if we remove the bias parameter, that is, $h_1(x,(W_1,b_1,W_2,b_2)):=\mathrm{ReLU}(xW_1+b_1)W_2$, or we apply the non-linear activation function, that is, $h_1(x,(W_1,b_1,W_2,b_2)):=\sigma(\mathrm{ReLU}(xW_1+b_1)W_2+b_2)$, then, the convergence rates of expert estimation might be improved, at least from the statistical perspective.
> > > > >
> > > > > **2. The more general setting:** We would like to emphasize that there are still several challenges in generalizing the current theoretical results to the setting where the data are sampled from an arbitrary distribution. One of the main issue is to study the universal approximation power of the gating function (the mixture weight function), which has remained elusive so far. This challenge has been acknowledged in previous work, e.g. [1]. Therefore, to the best of our knowledge, there have not been any papers investigating the convergence rate of expert estimation under this general setting. Since we also believe that further results on the universal approximation power of the gating function need to be developed, we leave the generalization of the current results to future development.
> > > > >
> > > > > Finally, we would like to emphasize that our main contribution in this work is the novel perturbed cosine router for MoE, which is theoretically and empirically demonstrated to outperform the linear router and the cosine router. We hope that the reviewer will evaluate our work based on this contribution and consider adjusting your score accordingly.
> > > > >
> > > > > [1] H. Nguyen, N. Ho, A. Rinaldo. Sigmoid Gating is More Sample Efficient than Softmax Gating in Mixture of Experts.

---

> ### Author Response · Authors · 2024-11-26
> **Regarding Reviewer idQ2's two new concerns (Part 1)**
>
> Dear Reviewer idQ2,
>
> We would like to thank you again for spending your precious time evaluating our manuscript, which we really appreciate. Following our previous post, we continue to further address your two concerns regarding our paper as follows:
>
> ### **1. Beyond the two-layer ReLU expert function:**
> Although the two-layer ReLU expert function is not strongly identifiable, several other popular expert functions still satisfy the strong identifiability condition in Definition 1, namely the two-layer GELU expert function which have been extensively leveraged in computer vision [1] and language modeling [2]. In our work, we have demonstrated the efficacy of the two-layer GELU expert function in domain generalization (see Appendix E.3). In particular, we show in Tables 3, 4, and 5 that with the two-layer GELU experts, using the perturbed cosine router helps achieve higher performance than using the linear router and the cosine router.
>
> Subsequently, we would like to emphasize that the strong identifiability condition is a sufficient condition for experts to have fast estimation rates (polynomial order). Therefore, the fact that the two-layer feed-forward networks are not strongly identifiable does not imply that they admit slow estimation rates. We will clarify this point via the comparison to the two previous work you mentioned:
>
> **Compared to Chen et al. [4]:** Chen et al. [4] consider a sparse mixture of experts with one expert activated per input in their analysis. Since only one expert is activated per input, the weight associated with the activated expert remains constant at one. Thus, if we considered this setting in our work, then the interaction between the router parameter $\beta_1$ and itself via the partial differential equation (4)
> $$\beta_1^{\top}\frac{\partial H}{\partial\beta_1}(x,\beta_1,\eta)=0,$$
> where $H(x,\beta_1,\eta):=\exp\Big(\frac{\beta_1^{\top}x}{||\beta_1||\cdot||x||}\Big)h(x,\eta)$, would not occur. As explained at the end of this response, such parameter interaction is the main reason for slow expert estimation rates, and the strong identifiability for experts is derived to avoid this interaction. Consequently, if we activate only one expert per input as in [4], the estimation rate of the two-layer ReLU expert function should be fast of polynomial order. We have empirically validated this claim by conducting the experiments on language modeling where we adopt the Switch Transformer [6] (activating one expert per token) and employ the two-layer ReLU expert function (See Section 5.2 and Appendix E.2 in our manuscript).
>
> **Compared to Chowdhury et al. [5]:** It can be seen from equation (1) in Chowdhury et al. [5] that they employ the expert function of the form $h_1(x,(W_1,b_1,W_2,b_2)):=\mathrm{ReLU}(xW_1+b_1)W_2$, which is a two-layer feed-forward network without the bias in the last layer. We can verify that this expert function is strongly identifiable and, therefore, has fast estimation rate as per our Theorem 4.
>
> **Explanation for the parameter interaction in equation (4)**:
> $$\beta_1^{\top}\frac{\partial H}{\partial\beta_1}(x,\beta_1,\eta)=0,$$
> where $H(x,\beta_1,\eta):=\exp\Big(\frac{\beta_1^{\top}x}{||\beta_1||\cdot||x||}\Big)h(x,\eta)$.
>
> *Intuitively*, the above partial differential equation (PDE) reveals that there is an intrinsic interaction between the parameter $\beta_1$ and itself. Although parameter interactions expressed in the language of PDEs have been observed in [7], the structure of the above interaction are much more sophisticated (even hold for the first-order derivatives while those in [7] occurs only when taking the second-order derivatives). Consequently, the rates for estimating parameters become significantly slow, and could be of order $1/\log^{\tau}(n)$ for some $\tau>0$, whereas those in [7] are also slow but still of polynomial orders.
>
> *Technically*, a key step in our proof techniques relies on the decomposition of the regression
> function discrepancy $f_{G_n}(x)-f_{G_*}(x)$ into a combination of linearly independent terms. This can be done by applying Taylor expansions to the function $H$, which is the product of softmax’s numerator and the expert function $h$. Due to linearly dependent terms resulting from the above PDE, we have to aggregate their coefficients in order to form the desired combination of linearly independent terms. Thus, the rates for estimating parameters are negatively affected.

---

> ### Author Response · Authors · 2024-11-26
> **Regarding Reviewer idQ2's two new concerns (Part 2)**
>
> ### **2. The setting where the ground truth is generated by an arbitrary regression function:**
> In our response to your Question 1, we introduced the framework when the data were generated from an arbitrary regression function, and presented our expected theoretical results under that setting. However, the main challenge of establishing the convergence rate of expert estimation under that setting comes from the underdeveloped universal approximation power of the (perturbed) cosine router. In particular, assume that $q:X\to\mathbb{R}$ is the regression function from which the data are sampled, and has the following Fourier transform:
> $$q(x)=\int_{X}e^{i\omega\cdot x}\tilde{q}(\omega)d\omega,$$
>
> for some complex-valued function $\tilde{q}(x)$ for which $\omega\tilde{q}(\omega)$ is integrable and the term
>
> $$C_q=\int_{X}||\omega||_2|\tilde{q}(\omega)|d\omega$$
>
> is finite. Then, the existence of a linear combination of $\tilde{k}$ (perturbed) cosine routers, denoted by $f_{\widetilde{G}}(x)$, such that
>
> $$\int_{X}[f_{\widetilde{G}}(x)-q(x)]^2d\mu(x)\leq\frac{(2C_q)^2}{\tilde{k}}.$$
>
> has remained elusive in the literature. This is a key yet challenging step to determine the convergence rate of expert estimation under the general setting. Such approximation property has been established for the sigmoidal function (see [3]) but there have not been any efforts to verify this property of the (perturbed) cosine router (to the best of our knowledge). Therefore, we believe that further technical tools need to be developed to investigate the universal approximation of the (perturbed) cosine router. Since this is beyond the scope of our work, we leave this research direction for future development.
>
> **Numerical experiments:** Despite the challenges of the theoretical study of the general setting, we carry out several numerical experiments where the data are generated from an arbitrary regression function in Appendix F. Consequently, Figure 2 indicates that the proposed perturbed cosine router still admits a faster estimation rate than those of the linear router and the cosine router.
>
> ### **3. The main contributions of our work:**
>
> Finally, we would like to emphasize that the main contribution of our work is the proposed perturbed cosine router in MoE.
>
> Theoretically, we show that using the perturbed cosine router helps achieve faster expert estimation rates than using the linear router or the cosine router. Therefore, it requires fewer data points to estimate the experts with the same approximation error.
>
> Empirically, we conduct several experiments on language modeling and domain generalization applications to demonstrate the benefits of using the proposed perturbed cosine router over the linear router and the cosine router in Section 5.
>
> ### **We hope that the reviewer will re-evaluate our manuscript and adjust your rating accordingly if our responses address your concerns.**

---

> > ### Comment · Reviewer_idQ2 · 2024-11-26
> > **Increasing score and more questions**
> >
> > Thank you for your response. As you specified that at least two-layer ReLU without bias satisfy the strong identifiability condition, I have increased the score again. However, I have some follow-up questions regarding the explanation around chen et. al [1]:
> >
> > 1.        The authors stated that, when one expert is activated by one input (e.g. chen et. al.[1]), the weight associated with the activated expert remains constant at one. Are you referring to the gating weight? If it is the gating weight, my understanding is that it is not constant at one. Rather, it is the SoftMax value of the expert which is activated by the input calculated over all the experts for the input (i.e. the gating value is the function of router’s parameters).
> > 2.        The authors stated that, if we activate only one expert per input, the estimation rate of the two-layer ReLU expert function should be fast of polynomial order. Is it also true for clean cosine router or even for linear router? (i.e. does one activated expert per input setting avoid parameter interaction for all variant of the routers discussed in the paper?) The authors referred to section 5.2 (switch transformer case) for experimental validation of their statement. However, in the results at Table 2 of the paper, there is no comparison with linear router. Also, perturbed cosine router is providing better results compare to clean cosine router in that case.
> >
> > Finally, I would suggest the authors to briefly discuss the technical challenges of extending the results to the more general setting (can be at the discussion of mis-specified setting in section 4).
> >
> > [1] Chen, Z., Deng, Y., Wu, Y., Gu, Q., & Li, Y. (2022). Towards understanding mixture of experts in deep learning. arXiv preprint arXiv:2208.02813.

---

> > > ### Author Response · Authors · 2024-11-26
> > > **Regarding Reviewer idQ2's follow-up questions (Part 1)**
> > >
> > > Dear Reviewer idQ2,
> > >
> > > We would like to thank you for increasing the rating to 6, which we really appreciate. In the sequel, we will answer your follow-up questions regarding our previous response. Please feel free to let us know if you have any additional concerns, we are willing to address all of them.
> > >
> > > **Q1: The authors stated that, when one expert is activated by one input (e.g. Chen et al.[1]), the weight associated with the activated expert remains constant at one. Are you referring to the gating weight? If it is the gating weight, my understanding is that it is not constant at one. Rather, it is the SoftMax value of the expert which is activated by the input calculated over all the experts for the input (i.e. the gating value is the function of router’s parameters).**
> > >
> > > Thanks for your question. Let us clarify why we said that the weight associated with the activated expert remained constant at one as follows:
> > >
> > > Firstly, let the regression function take the following form of a sparse mixture of experts (proposed in [2]) with (perturbed) cosine router:
> > >
> > > $$g_{G_*}(x):=\sum_{i=1}^{k_*} \mathrm{softmax}\left(\mathrm{TopK}\left(\frac{(\beta^*_{1i})^{\top}x}{(||\beta^*_{1i}||+\tau_1)\cdot(||x||+\tau_2)}+\beta^*_{0i}\right)\right)\cdot h(x,\eta^*_i),$$
> > >
> > > where $\mathrm{TopK}(u_i)=u_i$ if $u_i$ is in the top $K$ elements of $u=(u_1,\ldots,u_{k_*})$,
> > >
> > > otherwise, $\mathrm{TopK}(u_i)=-\infty$. Additionally, we define $\mathrm{softmax}(u_i):=\dfrac{\exp(u_i)}{\sum_{j=1}^{k_*}\exp(u_j)}$.
> > >
> > > To activate one expert per input, we set $K=1$. Next, we assume that given an input $x_1$, the first expert has the highest router score. Then, we get that
> > > $$\mathrm{TopK}\left(\frac{(\beta^*_{11})^{\top}x_1}{(||\beta^*_{11}||+\tau_1)\cdot(||x_1||+\tau_2)}+\beta^*_{01}\right)=\frac{(\beta^*_{11})^{\top}x_1}{(||\beta^*_{11}||+\tau_1)\cdot(||x_1||+\tau_2)}+\beta^*_{01},$$
> > > and
> > > $$\mathrm{TopK}\left(\frac{(\beta^*_{1i})^{\top}x_1}{(||\beta^*_{1i}||+\tau_1)\cdot(||x_1||+\tau_2)}+\beta^*_{0i}\right)=-\infty,$$
> > > for $2\leq i\leq k_*$. As a result, only the first expert is activated and the output of the sparse MoE is given by
> > > $$g_{G_*}(x_1)=h(x_1,\eta^*_{1}).$$
> > > This equation explains why we said that the weight associated with the activated expert remained constant at one.
> > >
> > > [1] Chen et al. Towards understanding mixture of experts in deep learning.
> > >
> > > [2] Shazeer et al. Outrageously large neural networks: The sparsely-gated mixture-of-experts layer.

---

> ### Author Response · Authors · 2024-11-26
> **Regarding Reviewer idQ2's follow-up questions (Part 2)**
>
> **Q2: The authors stated that, if we activate only one expert per input, the estimation rate of the two-layer ReLU expert function should be fast of polynomial order. Is it also true for clean cosine router or even for linear router? (i.e. does one activated expert per input setting avoid parameter interaction for all variant of the routers discussed in the paper?) The authors referred to section 5.2 (switch transformer case) for experimental validation of their statement. However, in the results at Table 2 of the paper, there is no comparison with linear router. Also, perturbed cosine router is providing better results compare to clean cosine router in that case.**
>
> Thanks for your questions. Firstly, we would like to confirm the claim that the two-layer ReLU expert function should have fast estimation rate if only one expert is activated per input also holds for the clean cosine router and the linear router. Secondly, let us address your concerns regarding the empirical results presented in Table 2 as follows:
>
> **1. No comparison with the linear router in language modeling:** Chi et al. [3] have theoretically shown that using the linear router might cause the representation collapse issue, that is, all the experts tend to learn the same representation which harms the model performance. Therefore, they propose using the clean cosine router to address this issue. Then, they conduct several extensive experiments on the language modeling to demonstrate the benefits of using the clean cosine router over the linear router. On the other hand, the main focus of our work is to show that the perturbation technique might help accelerate the expert convergence and stabilize the cosine router. Thus, for the language modeling tasks, we concentrate only on comparing the performance when using the perturbed cosine router versus when using the clean cosine router.
>
> **2. Our purpose of referring to Table 2:**  We refer to the empirical results in Table 2 in order to emphasize that although the two-layer ReLU expert function (with bias parameter in the last layer) is not strongly identifiable, using this expert function with both the perturbed cosine router and the clean cosine router still induces good performances under the scenario when only one expert is activated per input. And the strong identifiability condition is merely a sufficient condition for characterizing experts having fast estimation rates.
>
> **3. The perturbed cosine router achieves better results than the clean cosine router in Table 2:** Under the scenario when only one expert is activated per input, although the two-layer ReLU expert function should have faster estimation rate when using either the perturbed cosine router or the cosine router, the performance of the perturbed cosine router is still higher than that of the clean cosine router as the perturbation technique helps stabilize the cosine router. In particular, let us recall that the cosine router is given by
>
> $$\dfrac{\beta_{1i}^{\top}x}{||\beta_{1i}||\cdot||x||}.$$
>
> Therefore, since the norm of either the router parameter $\beta_1$ or the input $x$ might sometimes become close to zero during the training, the cosine router convergence and the model performance might be negatively affected. On the other hand, thanks to the noises added to the denominator of the perturbed cosine router
>
> $$\dfrac{\beta_{1i}^{\top}x}{(||\beta_{1i}||+\tau_1)(||x||+\tau_2)},$$
>
> the router convergence and the model performance are not negatively affected when the norms of the router parameter $\beta_1$ and the input $x$ approach zero.
>
> [3] Chi et al. On the representation collapse of sparse mixture of experts.
>
> **Q3: I would suggest the authors to briefly discuss the technical challenges of extending the results to the more general setting (can be at the discussion of mis-specified setting in section 4).**
>
> Thanks for your suggestion. We have just included the challenge discussion of extending the results to the mis-specified setting (presented in part 2 of our response to your new concerns) in the revision of our manuscript. Since we have reached the page limit for the main text, we decide to put the discussion in Appendix G.

---

> > ### Comment · Reviewer_idQ2 · 2024-11-26
> >
> > Thank you for your response. However, I want to draw attention to the authors about the two different definitions of gating function in Shazeer et. al. [1] and Fedus et.al. [2] (the paper introduced switch transformer). Shazeer et. al. defined the gating function as you described in your response but consider routing input to more than one expert. On the other hand, in Fedus et. al. [2] where only one expert is activated, the softmax is calculated over all the experts (please see equation (1) of Fedus et. al. [2]). Therefore, the gating weight is not constant and one.
> >
> > [1] Shazeer et al. Outrageously large neural networks: The sparsely-gated mixture-of-experts layer.
> >
> > [2] Fedus, William, Barret Zoph, and Noam Shazeer. "Switch transformers: Scaling to trillion parameter models with simple and efficient sparsity." Journal of Machine Learning Research 23.120 (2022): 1-39.

---

> > > ### Author Response · Authors · 2024-11-27
> > >
> > > Dear Reviewer idQ2,
> > >
> > > Thank you for pointing out the difference between the definitions of the sparse gating function given in Shazeer et al. [1] and Fedus et al. [2]. We will definitely revise our manuscript based on this difference. However, we would like to mention that this difference does not affect the main results of our paper. Finally, we'd like to express our gratitude for spending your precious time on evaluating our manuscript and giving constructive feedback to help strengthen the paper significantly, which we really appreciate.
> > >
> > > Best regards,
> > >
> > > The Authors

---

### Author Response · Authors · 2024-11-17
**General Response**

Dear Area Chairs and Reviewers,

We would like to express our gratitude for your
valuable feedback and constructive questions, which have greatly contributed to the improvement of our manuscript. Additionally, we really appreciate the following endorsement from the reviewers for our work:

**(i) Contributions:** The theoretical study of the cosine router MoE and its perturbed version is **novel** *(Reviewer idQ2)*. This paper presents a **compelling** theoretical picture that a perturbed cosine router should have better rates than the standard cosine router *(Reviewer hEMT)*.


**(ii) Soundness:**  The experiments are also **extensive** and seem quite **convincing** *(Reviewer hEMT)*. The experimental results validate the theoretical analysis and demonstrate the effectiveness of the proposed method, providing **solid** support for the authors' claims *(Reviewer VrcC)*.

**(iii) Presentation:** The paper is **well-written** *(Reviewer idQ2, Reviewer hEMT)* and the implication of the results is **easily understandable** *(Reviewer idQ2)*. The paper is **clearly and logically presented**, making it **easy to follow** the methodology and findings *(Reviewer VrcC)*.

Subsequently, we will address the concerns of the reviewers separately in our responses to their reviews and include according changes in the revision of our paper.

---

### Meta-Review · Area_Chair_YwVM · 2024-12-11

**Metareview:**

This paper introduces the perturbed cosine router in Mixture of Experts (MoE) models, showing that it improves sample efficiency by adding noise to router parameters. The authors provide theoretical guarantees and empirical validation, demonstrating that the perturbed router outperforms the standard cosine router.

Reviewers appreciated the novelty and theoretical contribution of the paper, noting the solid empirical validation and the clear explanation of the method’s advantages. The results were seen as significant, addressing an important problem in MoE models.

**Additional Comments On Reviewer Discussion:**

Reviewers raised concerns about the generalizability of the theoretical results and requested more clarity on certain claims. The authors responded by extending their analysis and clarifying the broader applicability of their approach. Overall, the reviewers expressed positive feedback after the revisions.

---

### Decision · Program_Chairs · 2025-01-22

Accept (Poster)